# Distributionally Robust Performative Optimization

**Zhuangzhuang Jia[1], Yijie Wang[2], Roy Dong[1], Grani A. Hanasusanto[1]**

[1]Department of Industrial and Enterprise Systems Engineering
University of Illinois Urbana–Champaign
[2]School of Economics and Management, Tongji University

{zj12,roydong,gah}@illinois.edu, yijiewang@tongji.edu.cn

## Abstract

In performative stochastic optimization, decisions can influence the distribution of random parameters, rendering the data-generating process itself decision-dependent. In practice, decision-makers rarely have access to the true distribution map and must instead rely on imperfect surrogate models, which can lead to severely suboptimal solutions under misspecification. Data scarcity or costly collection further exacerbates these challenges in real-world settings. To address these challenges, we propose a distributionally robust framework for performative optimization that explicitly accounts for ambiguity in the decision-dependent distribution. Our framework introduces three modeling paradigms that capture a broad range of applications in machine learning and decision-making under uncertainty. This latter setting has not previously been explored in the performative optimization literature. To tackle the intractability of the resulting nonconvex objectives, we develop an iterative algorithm named repeated robust risk minimization, which alternates between solving a decision-independent distributionally robust optimization problem and updating the ambiguity set based on the previous decision. This decoupling ensures computational tractability at each iteration while enhancing robustness to model uncertainty. We provide reformulations compatible with off-the-shelf solvers and establish theoretical guarantees on convergence and suboptimality. Extensive numerical experiments in strategic classification, revenue management, and portfolio optimization demonstrate significant performance gains over state-of-the-art baselines, highlighting the practical value of our approach.

## 1 Introduction

Decision-makers' actions often have a ripple effect on the external environment, which can lead to changes in the distribution of uncertain parameters. For instance, in the realm of portfolio management, institutional investors' decisions can have a profound effect on stock prices. This is partly due to their substantial capital, which can propel stock prices to rise (or fall) when they are buying (or selling) [57], and partly because their actions shape market sentiment towards those particular stocks [2]. Similarly, in revenue management, airlines often make forecasts and design pricing strategies based on the historical patterns of passenger behavior. However, passengers are not static; they often modify their behaviors in reaction to the new pricing strategies of airlines, which in turn shifts the overall pattern of consumer behavior [11].

When the solution of a stochastic optimization problem affects the distribution of the uncertain parameters, we call such a problem *performative* [40, 15]. The primary goal of the decision maker within such a dynamic environment is to find a decision that minimizes the expected risk after the environment has reacted to its deployment. A natural way of introducing this decision-dependent uncertainty is to construct a distributional map from the set of decisions to the space of distributions.

39th Conference on Neural Information Processing Systems (NeurIPS 2025).

However, the true underlying map is typically unknown in reality. Consequently, practitioners usually rely on a nominal or reference distribution map constructed from historical observations or domain expertise. While methods grounded in the reference distribution map might perform satisfactorily on the observed samples, they often fail to achieve acceptable performance in out-of-sample circumstances.

This paper aims to tackle this core deficiency by leveraging the ideas of *Distributionally Robust Optimization*. Unlike traditional approaches that assume a single distribution map, distributionally robust performative optimization (DRPO) adopts a more flexible strategy: it establishes an ambiguity set of plausible distributions centered at the reference distribution. Next, the objective of the decision maker is to derive an optimal decision that minimizes the worst-case expected risk, where the worst case is taken over all distribution maps from within this ambiguity set. By optimizing against this adversarial perspective within a neighborhood of the reference map, it mitigates the overfitting issue and improves the out-of-sample performance.

The difficulty in solving DRPO stems from the dependence of the distribution map on the decision, which prohibits the direct use of existing solution schemes from the robust and distributionally robust optimization literature. For instance, even under the simplest setting where the loss function is linear and the distribution map is linear in decisions, the resultant problem is a nonconvex bilinear program. We address this challenge by designing a repeated robust risk minimization algorithm. Specifically, the decision maker repeatedly obtains an optimal decision that minimizes DRPR risk using the reference distribution from the previous iteration. Finally, we conduct a theoretical analysis of the algorithm, providing convergence and sub-optimality guarantees. Our main contributions can be summarized as follows:

1. **General DRPO Framework**: We propose a distributionally robust framework based on Wasserstein distance for performative optimization. This approach enables safe decision-making when we lack full information about the underlying decision-dependent distributions and only have access to some reference distributions. It encompasses a wide class of loss functions and can be applied to numerous machine learning and decision-making problems. As a byproduct of our reformulations, we identify a relatively general setting where the distributionally robust model is equivalent to Tikhonov regularization.

2. **Repeated Robust Risk Minimization Algorithm**: We develop a repeated robust risk minimization algorithm for the problem that effectively mitigates the intractability of decision-dependent uncertainty. Our approach decouples the decisions associated with the ambiguity set from the expected loss, optimizing the latter while fixing the former to the previous decision in each iteration. This transforms the challenging DRPO problem into a sequence of tractable conic programs, rendering the framework computationally feasible and amenable to solutions via off-the-shelf solvers.

3. **Rigorous Theoretical Guarantees:** We provide convergence results of the repeated robust risk minimization algorithm to the stable solutions for our proposed models. To our knowledge, the convergence of such an algorithm has not previously been established in the distributionally robust settings. Our results show that, in general, the distributionally robust models lead to faster convergence than non-robust schemes. We further prove that our stable solutions are near to the robust performatively optimal ones.

## 1.1 Related Work

**Stochastic Optimization:** Our study closely relates to a body of research delving into endogenous uncertainty in stochastic optimization. Early examples include production planning problems with decision-dependent production costs [24] and oil planning problem with decision-dependent information discovery [18]. Their solution entails constructing a scenario-tree-based stochastic programming model and implementing a decomposition algorithm for resolution. Building on this, a subsequent study [19] extends the methodology to the multi-stage setting, focusing on production scheduling that minimizes costs while fulfilling diverse product demands. More broadly, [54] explore general multi-stage stochastic optimization problems beset with endogenous uncertainty, devising a conservative solution framework by leveraging piecewise linear decision rule approximations.

**Robust Optimization:** In robust optimization, decision-dependent uncertainty is usually imposed directly on the uncertainty set. Specific applications include customized endogenous uncertainty

sets for software partitioning [50] and robust scheduling, where the uncertainty set is constructed as a decision-dependent combination of simpler sets [56]. Decision-dependent sets have also been designed for primitive uncertainties in control systems [63]. The complexity analysis of the robust selection problem with decision-dependent information discovery is studied by [33], where they present polynomial complexity results for two special cases. For general settings, [37] show the problem is NP-complete and demonstrate the benefit of using endogenous uncertainty sets via a shortest-path problem. Algorithmic approaches have been developed, including exact nested decomposition schemes for two-stage problems [38] and approximation methods based on decision rules for the multistage setting [62].

**Distributionally Robust Optimization:** Our research expands upon the conventional framework of distributionally robust optimization by incorporating the influence of decision-making on the probability distribution. However, due to the problem difficulty, decision-dependent ambiguity sets generally lead to intractable reformulations, which could be computationally intensive for large instances. [61] study a broad class of distributionally robust optimization problems with decision-dependent moment ambiguity sets and conduct stability analysis. For the two stage setting, [31] analyze a wide range of decision-dependent ambiguity sets and establish non-convex semi-infinite reformulations and [23] explore decision-dependent information discovery using the K-adaptability approximation scheme. In multistage settings, [59] focus on moment-based ambiguity sets and derive a mixed-integer semidefinite programming reformulation. This decision-dependent DRO framework has been applied to a wide range of operations management problems, including nurse staffing [47], facility location [5], and retrofitting planning [14]. In contrast to the existing approaches that require an explicit specification of the ambiguity set, our scheme relies only on reference distributions at finitely many decisions.

**Performative Supervised Learning:** Our study also belongs to an emerging class of research on performative supervised learning, where algorithmic predictions can actively mold the surrounding environment and alter the underlying distributions of uncertain parameters [15, 20, 34, 40, 32]. The origins of performative learning can be traced back to studies on supervised learning under distribution drifts [3, 4, 16]. The foundational framework of performative prediction was introduced by [40], who designed a retraining algorithm and analyze the convergence behaviors to stochastic performatively stable points. Unfortunately, the convergence results do not extend to the robust settings, as taking worst-case expectations introduces non-smoothness into the objective function. Gradient-based methods have been developed in [40, 32, 15] for non-robust formulations, but they are inapplicable in robust settings due to the intractability of computing gradients of worst-case expectations. [28] has extended the scope of performative prediction to decision-making via performative omniprediction. We refer readers who are interested in performative learning to a comprehensive review [21].

Finally, we would like to highlight the difference between our paper and two related works on distributionally robust performative prediction. [39] propose a distributionally robust performative model to promote machine learning fairness. However, their robust smoothness assumption on the objective and robust sensitivity assumption on the worst-case distributions are unrealistic. Additionally, unlike the Wasserstein ambiguity set adopted in our paper, their phi-divergence ambiguity set precludes any continuous distribution, so it cannot ensure true probability distribution coverage guarantee. [58] study distributionally robust performative prediction based on a KL-divergence ambiguity set. Similar to [39], such an ambiguity set may fail to provide sufficient protections against distributions whose scenarios do not coincide with those in the reference distribution. In the paper, the authors develop an alternating minimization (coordinate descent) algorithm to solve the distributionally robust model. Unfortunately, such an algorithm may fail to converge, and the paper also does not provide any convergence guarantee. The algorithm also assumes the existence of a solver that can solve performative risk-sensitive minimization problems; however, to our knowledge, there is no existing literature that studies such problems and provides convergent algorithms.

## 1.2 Notation and Terminology

We use $\mathbb{R}_+$ and $\mathbb{R}_{++}$ to denote the sets of nonnegative and strictly positive real numbers, respectively. The identity matrix is denoted by $\mathbb{I}$. For any $n \in \mathbb{N}$, we define $[n]$ as the index set $\{1, \ldots, n\}$. The Dirac measure concentrating unit mass at $\boldsymbol{\xi} \in \Xi$ is denoted by $\delta_{\boldsymbol{\xi}}$. For any real-valued matrix $\boldsymbol{A}$, its Schatten-$q$ norm is defined as $\|\boldsymbol{A}\|_q = (\mathrm{tr}(\boldsymbol{A}^\top \boldsymbol{A})^{q/2})^q$. Random variables are designated with tilde signs (e.g., $\tilde{\boldsymbol{z}}$), while their realizations are denoted by the same symbols without tildes (e.g., $\boldsymbol{z}$).

## 2 Preliminaries

In stochastic optimization, the goal of the decision maker is to find a decision $\boldsymbol{\theta} \in \boldsymbol{\Theta} \subseteq \mathbb{R}^d$ that minimizes the risk

$$R(\boldsymbol{\theta}) = \mathbb{E}_{\mathbb{P}}[\ell(\tilde{\boldsymbol{z}}, \boldsymbol{\theta})], \tag{1}$$

where the vector $\tilde{\boldsymbol{z}} \in \mathcal{Z} \subseteq \mathbb{R}^m$ comprises the random parameters, and $\mathbb{P}$ denotes the underlying distribution. We assume that the loss function is convex in $\boldsymbol{\theta}$ for any fixed $\boldsymbol{z} \in \mathcal{Z}$. When the solution of a stochastic optimization problem affects the distribution of the uncertain parameters, we call such problem *performative* [40, 15]. A natural way of introducing this decision-dependent uncertainty is through a map $\mathbb{P}(\cdot)$ from the set of decisions to the space of distributions. Hence, in performative stochastic optimization problems, the objective of the decision maker is to obtain a decision $\boldsymbol{\theta}$ that minimizes the *performative risk*

$$PR(\boldsymbol{\theta}) = \mathbb{E}_{\mathbb{P}(\boldsymbol{\theta})}[\ell(\tilde{\boldsymbol{z}}, \boldsymbol{\theta})], \tag{2}$$

where $\mathbb{P}(\boldsymbol{\theta})$ is the distribution of uncertain parameters $\tilde{\boldsymbol{z}}$ given the choice of the decision $\boldsymbol{\theta}$. Unfortunately, the true underlying distribution map $\mathbb{P}(\boldsymbol{\theta})$ is unknown to the decision makers and usually approximated using the reference distribution $\hat{\mathbb{P}}(\boldsymbol{\theta}) = \sum_{s \in [S]} \hat{p}_s(\boldsymbol{\theta}) \delta_{\hat{z}_s(\boldsymbol{\theta})}$, where $\{\hat{z}_s(\boldsymbol{\theta})\}_{s \in [S]}$ represent plausible scenarios and $\{\hat{p}_s(\boldsymbol{\theta})\}_{s \in [S]}$ are their respective probabilities. This reference distribution could be derived from observations or expert knowledge elicitation.

Although methods based on a reference distribution may perform well when the true distribution closely aligns with it, their performance often deteriorates when this assumption is violated. *Distributionally robust optimization* has proven effective in addressing such distributional ambiguity in non-performative stochastic optimization settings. Unlike traditional approaches that assume a single distribution map, distributionally robust optimization adopts a more flexible strategy: it establishes an ambiguity set $\mathbb{B}(\hat{\mathbb{P}}(\boldsymbol{\theta}))$ of distributions that are close to the reference distribution. In this paper, we construct this ambiguity set based on the Wasserstein distance. A formal definition of the Wasserstein distance is as follows.

**Definition 1** (Wasserstein metric). For any $r \geq 1$, let $\mathcal{M}(\mathcal{Z})$ be the space of all probability distributions $\mathbb{Q}$ supported on $\mathcal{Z}$ satisfying $\mathbb{E}_{\mathbb{Q}}[c(\tilde{\boldsymbol{z}}, \boldsymbol{z}_0)^r] = \int_{\Xi} c(\boldsymbol{z}, \boldsymbol{z}_0)^r \mathbb{Q}(dz) < \infty$, where $\boldsymbol{z}_0 \in \mathcal{Z}$ is some reference point and $c(\boldsymbol{z}, \boldsymbol{z}_0)$ is a non-negative, continuous and thus lower semi-continuous [55] reference metric on $\mathcal{Z}$. The type-$r$ $(1 \leq r)$ Wasserstein distance between two distributions $\mathbb{Q}_1$ and $\mathbb{Q}_2$ is defined as

$$\mathbb{W}_r(\mathbb{Q}_1, \mathbb{Q}_2) = \inf_{\pi \in \Pi(\mathbb{Q}_1, \mathbb{Q}_2)} \left( \int_{\mathcal{Z} \times \mathcal{Z}} c(\boldsymbol{z}_1, \boldsymbol{z}_2)^r \, \pi(d\boldsymbol{z}_1, d\boldsymbol{z}_2) \right)^{\frac{1}{r}},$$

where $\Pi(\mathbb{Q}_1 \times \mathbb{Q}_2)$ is the set of all joint probability distributions of random vectors $\boldsymbol{z}_1$ and $\boldsymbol{z}_2$ with marginals $\mathbb{Q}_1$ and $\mathbb{Q}_2$, respectively.

The Wasserstein metric offers a natural way of comparing two distributions when one is derived from the other by small perturbations. The decision variable $\pi$ can be interpreted as a transportation plan for moving a mass distribution denoted by $\mathbb{Q}_1$ to another one denoted by $\mathbb{Q}_2$, where the transportation cost between two points $\boldsymbol{z}_1$ and $\boldsymbol{z}_2$ is given by $c(\boldsymbol{z}_1, \boldsymbol{z}_2)$. Therefore, the $r$-Wasserstein distance can be viewed as the $r$-th root of the minimum transportation cost between $\mathbb{Q}_1$ and $\mathbb{Q}_2$. An important advantage of the Wasserstein distance is its ability to handle distributions with non-overlapping supports, i.e., even when $\mathbb{Q}_1$ and $\mathbb{Q}_2$ have different supports, the Wasserstein distance provides a finite, meaningful value. By contrast, measures like KL-divergence become undefined under such scenarios. We now consider the following Wasserstein ambiguity set

$$\mathbb{B}(\hat{\mathbb{P}}(\boldsymbol{\theta})) = \left\{ \mathbb{Q} \in \mathcal{M}(\Xi) \; : \quad \mathbb{W}_r(\mathbb{Q}, \hat{\mathbb{P}}(\boldsymbol{\theta})) \leq \rho \; \right\}, \tag{3}$$

which is a neighborhood around the reference distribution $\hat{\mathbb{P}}(\boldsymbol{\theta})$. The Wasserstein ambiguity set contains all the distributions whose $r$-Wasserstein distance from $\hat{\mathbb{P}}(\boldsymbol{\theta})$ is less than or equal to $\rho$.

Equipped with the Wasserstein ambiguity set, the decision maker minimizes the *distributionally robust performative risk*

$$DRPR(\boldsymbol{\theta}) := \sup_{\mathbb{Q} \in \mathbb{B}(\hat{\mathbb{P}}(\boldsymbol{\theta}))} \mathbb{E}_{\mathbb{Q}} \, \ell(\tilde{\boldsymbol{z}}, \boldsymbol{\theta}), \tag{4}$$

where $\mathbb{Q}$ is a distribution from within the prescribed ambiguity set $\mathbb{B}(\hat{\mathbb{P}}(\boldsymbol{\theta}))$. In other words, the model optimizes the expected risk over the worst-case distribution map, thereby mitigating overfitting to the reference distribution and improving generalization performance to other plausible distributions. We now introduce the following concepts regarding the optimality and stability of the solutions.

**Definition 2** (Robust performative optimality). A decision $\boldsymbol{\theta}_{\mathrm{RPO}}$ is *robust performatively optimal* if the following relationship holds:

$$\boldsymbol{\theta}_{\mathrm{RPO}} \in \arg\min_{\boldsymbol{\theta}\in\boldsymbol{\Theta}} \sup_{\mathbb{Q}\in\mathbb{B}(\hat{\mathbb{P}}(\boldsymbol{\theta}))} \mathbb{E}_{\mathbb{Q}}\,\ell(\tilde{\boldsymbol{z}},\boldsymbol{\theta}).$$

**Definition 3** (Robust performative stability). A decision $\boldsymbol{\theta}_{\mathrm{RPS}}$ is *robust performatively stable* if the following relationship holds:

$$\boldsymbol{\theta}_{\mathrm{RPS}} \in \arg\min_{\boldsymbol{\theta}\in\boldsymbol{\Theta}} \sup_{\mathbb{Q}\in\mathbb{B}(\hat{\mathbb{P}}(\boldsymbol{\theta}_{\mathrm{RPS}}))} \mathbb{E}_{\mathbb{Q}}\,\ell(\tilde{\boldsymbol{z}},\boldsymbol{\theta}).$$

While distinct from the robust performatively optimal solution, a robust performatively stable solution constitutes a fixed point of the problem and is optimal with respect to the worst-case expected loss over the ambiguity set it induces.

**Definition 4** (Robust decoupled performative risk). We define

$$\mathbb{J}_{\boldsymbol{\eta}}(\boldsymbol{\theta}) := \sup_{\mathbb{Q}\in\mathbb{B}(\hat{\mathbb{P}}(\boldsymbol{\eta}))} \mathbb{E}_{\mathbb{Q}}\,\ell(\tilde{\boldsymbol{z}},\boldsymbol{\theta})$$

as the *robust decoupled performative risk*, separating the decision $\boldsymbol{\eta}$ associated with the ambiguity set and the decision $\boldsymbol{\theta}$ associated with the risk; then, $\boldsymbol{\theta}_{\mathrm{RPS}} \in \arg\min_{\boldsymbol{\theta}} \mathbb{J}_{\boldsymbol{\theta}_{\mathrm{RPS}}}(\boldsymbol{\theta})$.

# 3 Repeated Robust Risk Minimization Algorithm

In this section, we introduce the *repeated robust risk minimization* algorithm for solving the distributionally robust performative risk minimization problem and investigate its fundamental properties. The algorithm starts with an initial solution $\boldsymbol{\theta}_0$, and for every $t \geq 0$, the subsequent $\boldsymbol{\theta}_{t+1}$ can be obtained by solving the following robust risk minimization problem:

$$\min_{\boldsymbol{\theta}\in\boldsymbol{\Theta}} \mathbb{J}_{\boldsymbol{\theta}_t}(\boldsymbol{\theta}). \tag{RRMP}$$

The algorithm addresses the computational challenges posed by decision-dependent distributions by constructing the ambiguity set using the reference distribution based on the optimal decision from the previous iteration. Thus, it effectively decouples the current decision from the ambiguity set, simplifying the optimization process.

## 3.1 Models and Their Tractable Reformulations

We present three models that cover a broad spectrum of problems in machine learning and decision-making under uncertainty. We first consider the case where the loss function is given by the composition of a Lipschitz continuous function and a quadratic function. This class of problems is particularly relevant in machine learning and robust statistics, as it captures many commonly used models such as linear regression [36], logistic regression [22], and certain types of support vector machines [48].

**Model 1.** *Assume that the loss function is defined as*

$$\ell(\boldsymbol{Z},\boldsymbol{\theta}) = \mathcal{L}(\boldsymbol{\theta}^\top \boldsymbol{Y}\boldsymbol{\theta} + 2\boldsymbol{z}^\top\boldsymbol{\theta} + z^0), \tag{5}$$

*where $\tilde{\boldsymbol{Z}} = (\tilde{\boldsymbol{Y}}, \tilde{\boldsymbol{z}}, \tilde{z}^0)$ includes random variables $\tilde{\boldsymbol{Y}} \in \mathbb{S}^N, \tilde{\boldsymbol{z}} \in \mathbb{R}^d$ and $\tilde{z}^0 \in \mathbb{R}$. The function $\mathcal{L}(\cdot)$ is assumed to be $L$-Lipschitz continuous.*

*We consider the 1-Wasserstein ball, where the ground cost function $c$ is given by the Schatten-$\infty$ norm. Under this setting,* (RRMP) *is equivalent to the Tikhonov regularized problem*

$$\inf_{\boldsymbol{\theta}\in\boldsymbol{\Theta}} \mathbb{E}_{\hat{\mathbb{P}}(\boldsymbol{\theta}_t)}\left[\mathcal{L}\left(\boldsymbol{\theta}^\top\tilde{\boldsymbol{Y}}\boldsymbol{\theta} + 2\tilde{\boldsymbol{z}}^\top\boldsymbol{\theta} + \tilde{z}^0\right)\right] + \rho L\|(\boldsymbol{\theta},1)\|_2^2.$$

This result demonstrates that, under mild assumptions, (RRMP) can be reformulated as a regularized risk minimization problem. This substantially generalizes the findings of [30], who established an equivalence to Tikhonov regularization for strongly convex quadratic loss functions and martingale-constrained Wasserstein ambiguity sets. Our result reveals that Tikhonov regularization can be obtained for a broader class of loss functions without requiring complicating martingale constraints, thereby sharpening the theoretical understanding of the connection between distributional robustness and regularization.

The following model provides a general formulation that is typically considered in the distributionally robust optimization literature [43, 44]. This model is pertinent to many applications in decision-making under uncertainty, including in inventory management [29] and and energy systems [27]. To the best of our knowledge, such a formulation has not previously been proposed in the performative optimization literature, which mainly focuses on prediction problems. Note that if the $L$-lipschitz continuous function $\mathcal{L}$ in (5) is piecewise linear, then this model constitutes a generalization.

**Model 2.** *Assume that the loss function is defined as*

$$\ell(\boldsymbol{Z}, \boldsymbol{\theta}) = \max_{j \in [J]} Q_j(\boldsymbol{Z}, \boldsymbol{\theta}),$$

*where $\tilde{\boldsymbol{Z}} = (\tilde{\boldsymbol{Y}}, \tilde{\boldsymbol{z}}, \tilde{z}^0)$ includes random variables $\tilde{\boldsymbol{Y}} \in \mathbb{S}^N, \tilde{\boldsymbol{z}} \in \mathbb{R}^d$ and $\tilde{z}^0 \in \mathbb{R}$. Each component $Q_j(\boldsymbol{Z}, \boldsymbol{\theta})$ is a quadratic function of the form*

$$Q_j(\boldsymbol{Z}, \boldsymbol{\theta}) = \boldsymbol{a}_j(\boldsymbol{\theta})^\top \boldsymbol{Y} \boldsymbol{a}_j(\boldsymbol{\theta}) + 2\boldsymbol{b}_j(\boldsymbol{\theta})^\top \boldsymbol{z} + c_j(\boldsymbol{\theta}) z^0,$$

*with parameter-dependent coefficients given by affine functions $\boldsymbol{a}_j(\boldsymbol{\theta}) = \overline{\boldsymbol{a}}_j + \overline{\boldsymbol{A}}_j \boldsymbol{\theta}$, $\boldsymbol{b}_j(\boldsymbol{\theta}) = \overline{\boldsymbol{b}}_j + \overline{\boldsymbol{B}}_j \boldsymbol{\theta}$, and $c_j(\boldsymbol{\theta}) = \overline{c}_{j0} + \overline{\boldsymbol{c}}_j^\top \boldsymbol{\theta}$ for all $j \in [J]$, where $\overline{\boldsymbol{a}}_j, \overline{\boldsymbol{b}}_j \in \mathbb{R}^N, \overline{c}_{j0} \in \mathbb{R}, \overline{\boldsymbol{A}}_j, \overline{\boldsymbol{B}}_j \in \mathbb{R}^{N \times d}$, and $\overline{\boldsymbol{c}}_j \in \mathbb{R}^d$.*

*Consider the 1-Wasserstein ball with the Schatten-$\infty$ norm ground cost. Then, the optimal value of the following exponential conic program provides an arbitrarily tight conservative approximation for* (RRMP).

$$
\begin{aligned}
\inf \quad & \sum_{s \in [S]} \hat{p}_s(\boldsymbol{\theta}_t) t_s + t_{S+1} \\
\text{s.t.} \quad & \boldsymbol{\theta} \in \boldsymbol{\Theta}, \ t_s \in \mathbb{R} & \forall s \in [S+1] \\
& \zeta_{s,j}, r_{s,j} \in \mathbb{R}, \ (r_{s,j}, \mu, \zeta_{s,j} - t_s) \in K_{\exp} & \forall s \in [S+1] \ j \in [J] \\
& \boldsymbol{\theta}^\top \overline{\boldsymbol{A}}_j^\top \hat{\boldsymbol{Y}}_s \overline{\boldsymbol{A}}_j \boldsymbol{\theta} + (2\overline{\boldsymbol{a}}_j^\top \hat{\boldsymbol{Y}}_s \overline{\boldsymbol{A}}_j + 2\hat{\boldsymbol{z}}_s^\top \overline{\boldsymbol{B}}_j + \hat{z}_s^0 \overline{\boldsymbol{c}}_j^\top) \boldsymbol{\theta} \\
& \qquad + \overline{\boldsymbol{a}}_j^\top \hat{\boldsymbol{Y}}_s \overline{\boldsymbol{a}}_j + 2\overline{\boldsymbol{b}}_j^\top \hat{\boldsymbol{z}}_s + \hat{z}_s^0 \overline{c}_{j0} \leq \zeta_{s,j} & \forall s \in [S+1] \ j \in [J] \\
& \sum_{j \in J} r_{s,j} \leq \mu & \forall s \in [S+1].
\end{aligned}
\tag{6}
$$

*where $\hat{\boldsymbol{Y}}_{S+1} = \rho\mathbb{I}, \hat{\boldsymbol{z}}_{S+1} = \boldsymbol{0}, \hat{z}_{S+1}^0 = \rho$, and $\mu \in \mathbb{R}_+$ is the smoothing parameter.*

The formulation (6) relies on the exponential smoothing techniques described in [7, Section 2.2]. The primary motivation for this approach lies in the need to handle the nonsmoothness incurred by the inner maximization over quadratic functions, which will hinder the convergence of our proposed algorithm. To address this challenge, we apply a log-sum-exp smoothing approximation with smoothing parameter $\mu > 0$, which yields a smoothed robust decoupled performative risk, denoted as $\mathbb{J}_{\boldsymbol{\theta}_t}^\mu$. And problem (6) is equivalent to $\inf_{\boldsymbol{\theta} \in \boldsymbol{\Theta}} \mathbb{J}_{\boldsymbol{\theta}_t}^\mu(\boldsymbol{\theta})$ (see details in Appendix D).

As is standard in exponential smoothing, the smoothed objective serves as a uniform upper bound to the original nonsmooth function: $\mathbb{J}_{\boldsymbol{\theta}_t}(\boldsymbol{\theta}) \leq \mathbb{J}_{\boldsymbol{\theta}_t}^\mu(\boldsymbol{\theta}) \ \forall \boldsymbol{\theta} \in \boldsymbol{\Theta}$, which ensures that optimization over the smooth surrogate does not underestimate the original objective. Importantly, $\mathbb{J}_{\boldsymbol{\theta}_t}^\mu$ epi-converges to $\mathbb{J}_{\boldsymbol{\theta}_t}$ as the smoothing parameter $\mu \downarrow 0$ [46, Theorem 7.17], which ensures convergence of optimal solutions whenever $\boldsymbol{\Theta}$ is compact [46, Theorem 7.33].

Finally, inspired by minimax [52] and adversarially robust optimization literature [49], we turn to the setting where the loss function is convex-concave. In this setting, we exploit the 2-Wasserstein ambiguity set to ensure the convergence of the repeated risk minimization algorithm. This model further allows one to impose additional structural support information that may reduce the overconservatism of the distributionally robust solutions. The following theorem provides a convex reformulation for the model, leveraging convex conjugate representations and support functions.

**Model 3.** *Let $\mathcal{Z} \subseteq \mathbb{R}^m$ be a nonempty, convex and closed set, and consider the 2-Wasserstein ball, where the ground cost $c$ is given by the Euclidean norm on $\mathbb{R}^d$. Suppose that for every $\boldsymbol{\theta}$, the function $\ell(\boldsymbol{z}, \boldsymbol{\theta})$ is proper, concave, and upper-semicontinuous in $\boldsymbol{z}$. Then, the optimal value of the following finite convex program provides an arbitrarily tight conservative approximation for* (RRMP)*:*

$$
\begin{aligned}
\inf \quad & \sum_{s \in [S]} \hat{p}_s(\boldsymbol{\theta}_t) \left( [-\ell]^*(\boldsymbol{r}_s - \boldsymbol{\zeta}_s, \boldsymbol{\theta}) + \sigma_{\mathcal{Z}}(\boldsymbol{\zeta}_s) - \boldsymbol{r}_s^\top \hat{\boldsymbol{z}}_s + \frac{1}{4\lambda} \|\boldsymbol{r}_s\|_2^2 \right) + \rho^2 \lambda + \tau \lambda^2 \\
\text{s.t.} \quad & \boldsymbol{\theta} \in \boldsymbol{\Theta}, \ \ \lambda \in \mathbb{R}_+, \ \ \boldsymbol{r}_s \in \mathbb{R}^m \ \forall s \in [S], \ \ \boldsymbol{\zeta}_s \in \mathbb{R}^m \ \forall s \in [S]
\end{aligned}
\tag{7}
$$

*where $\tau \in \mathbb{R}_{++}$ is a constant, $[-\ell]^*(\boldsymbol{\xi}, \boldsymbol{\theta}) = \sup_{\boldsymbol{z} \in \mathbb{R}^m} \boldsymbol{\xi}^\top \boldsymbol{z} - [-\ell(\boldsymbol{z}, \boldsymbol{\theta})]$ denotes the conjugate of $-\ell$ with respect to $\boldsymbol{z}$, and $\sigma_{\mathcal{Z}}(\boldsymbol{\xi}) = \sup_{\boldsymbol{z} \in \mathcal{Z}} \boldsymbol{\xi}^\top \boldsymbol{z}$ is the support function of $\mathcal{Z} \in \mathbb{R}^m$.*

Note that in problem (7), the loss function $\ell(\boldsymbol{z}, \boldsymbol{\theta})$ and the support set $\mathcal{Z}$ enter the formulation through the convex conjugate of the negative loss $[-\ell]^*(\cdot, \boldsymbol{\theta})$, and the support function $\sigma_{\mathcal{Z}}(\cdot)$, respectively. Both transformations yield convex functions under the assumptions stated in Model 3. Furthermore, the term $(1/4\lambda)\|\boldsymbol{r}_s\|_2^2$ is jointly convex in $(\lambda, \boldsymbol{r}_s)$ [9, section 3.2.6]. Consequently, all objective and constraint functions in problem (7) are convex, and the overall optimization problem is manifestly convex.

The reformulation in Model 3 introduces the dual variable $\lambda$, which appears alongside the decision variable $\boldsymbol{\theta}$. This motivates the use of an augmented vector $\overline{\boldsymbol{\theta}} = (\boldsymbol{\theta}, \lambda) \in \boldsymbol{\Theta} \times \mathbb{R}_+ = \overline{\boldsymbol{\Theta}}$. To ensure strong convexity in the joint variable $\overline{\boldsymbol{\theta}}$ which is important for the convergence guarantees of the R³M algorithm, we introduce a regularization term $\tau \lambda^2$, where $\tau > 0$. This leads to the regularized robust decoupled performative risk, denoted by $\mathbb{J}^\tau_{\boldsymbol{\theta}_t}(\overline{\boldsymbol{\theta}})$. And problem (7) is equivalent to $\inf_{\overline{\boldsymbol{\theta}} \in \overline{\boldsymbol{\Theta}}} \mathbb{J}^\tau_{\boldsymbol{\theta}_t}(\overline{\boldsymbol{\theta}})$. Notice that for any $\boldsymbol{\theta} \in \boldsymbol{\Theta}$, $\inf_{\lambda \geq 0} \mathbb{J}^\tau_{\boldsymbol{\theta}_t}(\boldsymbol{\theta}, \lambda)$ converges to $\mathbb{J}_{\boldsymbol{\theta}_t}(\boldsymbol{\theta})$ as $\tau \downarrow 0$. If $\ell(\boldsymbol{z}, \boldsymbol{\theta})$ is lower semicontinuous in $\boldsymbol{\theta}$, then so is $\mathbb{J}_{\boldsymbol{\theta}_t}$. Hence, $\inf_{\lambda \geq 0} \mathbb{J}^\tau_{\boldsymbol{\theta}_t}(\cdot, \lambda)$, epi-converges to $\mathbb{J}_{\boldsymbol{\theta}_t}(\cdot)$ [46, Theorem 7.17], and the minimizer $\hat{\boldsymbol{\theta}}$ of (7) converges to a minimizer of (RRMP) whenever $\boldsymbol{\Theta}$ is compact [46, Theorem 7.33].

### 3.2 Convergence Analysis

We now establish convergence guarantees for the RRRM algorithm when applied to the reformulations introduced in Models 1, 2, and 3. As noted earlier, we employ the smoothed objective $\mathbb{J}^\mu_{\boldsymbol{\theta}_t}(\boldsymbol{\theta})$ for (RRMP) in Model 2 and the regularized objective $\mathbb{J}^\tau_{\boldsymbol{\theta}_t}(\overline{\boldsymbol{\theta}})$ in Model 3. Each model requires specific assumptions to ensure contraction of the risk map and hence convergence:

**(A1)** Model 1: The loss function satisfies the $\gamma$-strong convexity (B1) and $\beta$-smoothness (B2).

**(A2)** Model 2: For all $j \in [J]$, $\overline{\boldsymbol{A}}_j^\top \overline{\boldsymbol{A}}_j \succ \boldsymbol{0}$ and let $\alpha = 2 \min_{j \in [J]} \lambda_{\min}(\overline{\boldsymbol{A}}_j^\top \overline{\boldsymbol{A}}_j)$. In addition, the feasible set $\boldsymbol{\Theta}$ and the support of $\hat{\mathbb{P}}(\boldsymbol{\theta}_t)$ are bounded.

**(A3)** Model 3: The loss function satisfies the $\gamma$-strong convexity (B1) and $\beta$-jointly smoothness (B2). Additionally the support $\mathcal{Z}$ has a finite diameter $D < \infty$.

Under these respective conditions, the algorithm converges linearly to a stable point.

**Theorem 1.** *Suppose the loss functions in Models 1, 2, and 3 satisfy Assumptions (A1), (A2), and (A3) respectively, and that the distribution map $\hat{\mathbb{P}}(\cdot)$ satisfies the $\epsilon$-sensitivity condition (B3). Then:*

*(a) $\|\boldsymbol{\theta}_{t+1} - \boldsymbol{\theta}'_{t+1}\|_2 \leq \epsilon \kappa \|\boldsymbol{\theta}_t - \boldsymbol{\theta}'_t\|_2$ for all $\boldsymbol{\theta}_t, \boldsymbol{\theta}'_t \in \boldsymbol{\Theta}$.*

*(b) if $\epsilon \kappa < 1$, the iterate $\boldsymbol{\theta}_t$ of (RRMP) converges linearly to a unique performatively stable point $\boldsymbol{\theta}_{\mathrm{RPS}}$:*

$$
\|\boldsymbol{\theta}_t - \boldsymbol{\theta}_{\mathrm{RPS}}\|_2 \leq \Delta \ \text{for } t \geq (1 - \epsilon \kappa)^{-1} \log\left(\|\boldsymbol{\theta}_0 - \boldsymbol{\theta}_{\mathrm{RPS}}\|_2 / \Delta\right).
$$

*where $\Delta > 0$ is a predefined tolerance level. The fixed point $\boldsymbol{\theta}_{\mathrm{RPS}}$ depends on the model and is denoted $\boldsymbol{\theta}^\mu_{\mathrm{RPS}}$ under Model 2, and $\boldsymbol{\theta}^\tau_{\mathrm{RPS}}$ under Model 3.*

*Here, $\kappa = \beta/(\gamma + 2\rho L)$ for Model 1, $\kappa = Jdk_3\left(\frac{k_1 k_2}{\mu} + 1\right)/\rho\alpha$ for Model 2, and $\kappa = (\beta + 4D)/\min\{2\tau, \gamma\}$ for Model 3, where $k_1, k_2, k_3 < \infty$ are model-dependent constants.*

This convergence result highlights several compelling advantages of our DRPO framework, particularly when contrasted with traditional approaches to performative optimization that necessitate strong convexity and smoothness for convergence. First, our DRPO framework establishes convergence guarantees for loss functions that are convex but not necessarily strongly convex. Additionally, our exponential smoothing tricks applied in Model 2 enable the RRRM algorithm to converge for non-smooth loss functions. Finally, our DRPO framework accelerates the convergence rate compared with its non-robust counterpart. These advancements significantly broadens the scope of problems amenable to performative optimization and enhance the computational efficiency in practical problems.

### 3.3 Suboptimality Guarantees

Our next result demonstrates that the robust performatively stable solution $\boldsymbol{\theta}_{\mathrm{RPS}}$ is close to the robust performatively optimal solution $\boldsymbol{\theta}_{\mathrm{RPO}}$ whenever the $\epsilon$-sensitivity of the distribution map is small, the loss function has a large strong convexity parameter $\gamma$, or the distributionally robust model induces a regularization with large strong convexity parameter $\rho$.

**Theorem 2.** *Suppose all conditions in Theorem 1 hold. Additionally, assume that the loss function is $L_z$-Lipschitz in $\boldsymbol{z}$ for both models, and $L_\theta$-Lipschitz in $\boldsymbol{\theta}$ for Model 3. Then, the suboptimality gap between the robust performatively stable solution and the optimal solution under the true distribution is bounded as follows:*

*(a) **Model 1:** $\mathbb{J}_{\boldsymbol{\theta}_{\mathrm{RPS}}}(\boldsymbol{\theta}_{\mathrm{RPS}}) - \mathbb{J}_{\boldsymbol{\theta}_{\mathrm{RPO}}}(\boldsymbol{\theta}_{\mathrm{RPO}}) \leq \frac{2\epsilon^2 L_z^2}{\gamma + 2\rho L}.$*

*(b) **Model 2:** $\mathbb{J}_{\boldsymbol{\theta}_{\mathrm{RPS}}^\mu}^\mu(\boldsymbol{\theta}_{\mathrm{RPS}}^\mu) - \mathbb{J}_{\boldsymbol{\theta}_{\mathrm{RPO}}}(\boldsymbol{\theta}_{\mathrm{RPO}}) \leq \frac{2(\epsilon L_z + 2\mu' \log J)^2}{\rho\alpha}.$*

*(c) **Model 3:** $\mathbb{J}_{\boldsymbol{\theta}_{\mathrm{RPS}}^\tau}^\tau(\overline{\boldsymbol{\theta}}_{\mathrm{RPS}}^\tau) - \mathbb{J}_{\boldsymbol{\theta}_{\mathrm{RPO}}}(\overline{\boldsymbol{\theta}}_{\mathrm{RPO}}) \leq \tau\overline{\lambda}^2 + \frac{(2\tau\overline{\lambda} + \rho^2 + D^2 + L_\theta + \epsilon L_z)2\epsilon L_z}{\min(\gamma, 2\tau)}.$*

*Here, $\mu' \in [0, 1]$ is a constant that satisfies $\mu \leq \mu'\|\boldsymbol{\theta}_{\mathrm{RPS}}^\mu - \boldsymbol{\theta}_{\mathrm{RPO}}\|_2$, and $\overline{\lambda}$ is the upper bound on the optimal $\lambda$ as given in Lemma F.1.*

The suboptimality bound for Model 2 also highlights the advantages of our DRPO framework alongside the smoothing tricks. For non-smooth, convex but not strongly-convex functions, the suboptimality bound under the traditional performative prediction framework can be arbitrarily large (as $\rho \to 0, \mu \to 0$) as suggested by our result.

## 4 Experiments

In this section, we present numerical experiments to evaluate the performance of our proposed models across three applications: strategic classification, revenue management, and portfolio optimization. All experiments were conducted on a laptop equipped with a 6-core, 2.3 GHz Intel Core i7 CPU and 16 GB of RAM. The optimization problems were implemented in Python 3.11.

### 4.1 Strategic Classification

We consider a simulated strategic classification problem from [40] using a class-balanced subset of a Kaggle credit scoring dataset [25]. The dataset contains features $\tilde{\boldsymbol{x}} \in \mathbb{R}^P$ about borrowers, such as their ages and the number of open loans. The outcomes $\tilde{y} \in \{-1, 1\}$ are equal to 1 if the individual defaulted on a loan and $-1$ otherwise. The institution's objective is to predict whether an individual will default on their debt.

Under the strategic classification setting, individuals respond to the institution's classifier by altering their features to increase their likelihood of receiving a favorable classification. The institution employs logistic regression for classification, with $\tilde{\boldsymbol{z}} = \tilde{\boldsymbol{x}}\tilde{y}$, and the loss function is given by $\log(1 + \exp(-\boldsymbol{x}^\top \boldsymbol{\theta} y))$. This setting aligns with Theorem 1, where $\mathcal{L}$ represents the logloss function with Lipschitz constant $L = 1$, and the quadratic function $\boldsymbol{\theta}^\top \boldsymbol{Y} \boldsymbol{\theta} + 2\boldsymbol{z}^\top \boldsymbol{\theta} + z^0$ simplifies to the affine function $\boldsymbol{z}^\top \boldsymbol{\theta}$. See Appendix G for additional details.

We compare the performance of our robust models (with type-1 and type-2 Wasserstein ambiguity sets) against the alternating minimization algorithm under KL divergence ambiguity set (AMKL)

from [58]. Additionally, we include a non-robust model as a baseline for comparison. The training set is fixed at 200 samples, while approximately 3,600 data points are used for out-of-sample testing. This setup reflects realistic scenarios where data collection is costly or limited. In credit scoring, for example, obtaining labeled data often requires expensive evaluations, expert assessments, or lengthy observation periods. All models are trained for 40 iterations, with the robust parameter set to 0.1 for all robust variants.

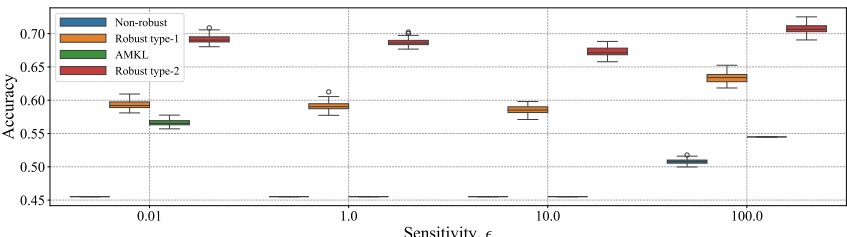

Figure 1: Out of sample performance of different approaches

Figure 1 displays the box plots of 50 independent trials. As shown, our robust models outperform the AMKL algorithm, which performs quite poorly. This may be due to its susceptibility to local optima, as well as the limitations of the KL divergence ambiguity set, which may not sufficiently guard against distributional shifts that are not well represented in the reference distribution. All robust models significantly outperform the non-robust baseline, highlighting the effectiveness of distributional robustness. Finally, we find that the robust model with a type-2 Wasserstein ambiguity set outperforms the one with a type-1 Wasserstein ambiguity set. This arises primarily due to the geometry of the ambiguity set. As discussed in [10], selecting the optimal radius $\rho$ is challenging, and the 2-Wasserstein ball often provides better performance because it offers a wider range of radius values for which the robust solution outperforms its non-robust counterpart.

## 4.2 Revenue Management

In this experiment, we address the revenue management problem where the decision-maker determines the unit price $\theta \geq 0$ for a fixed quantity of perishable products $q \in \mathbb{Z}_{++}$, such as hotel rooms or airplane seats, under uncertain demand $\tilde{z} \sim \mathbb{P}(\theta)$, with higher prices inducing lower demand [1, 13, 42].

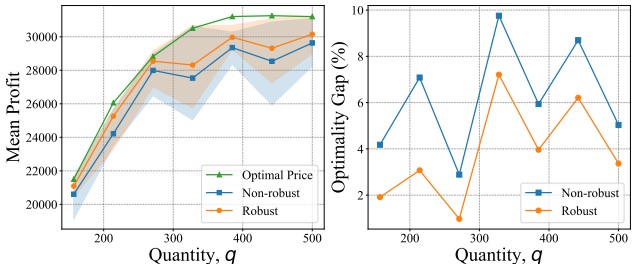

Following [41], we model the price-dependent demands using an additive function $\tilde{z}(\theta) = -a\theta + b + \tilde{\epsilon}$. Here,

Figure 2: Out-of-sample performance of pricing schemes

unknown parameters $a > 0$ and $b > 0$ capture a deterministic linear demand curve, and $\tilde{\epsilon}$ is a random variable with a bounded support that is characterized by an unknown density function. Instead of the true model, we only have access to a surrogate model $-\bar{a}\theta + \bar{b}$, which deviates from the true model as the parameters $\bar{a}$ and $\bar{b}$ do not accurately represent their counterparts. Under this additive demand setting, the associated loss function becomes a piecewise quadratic function in $\theta$, allowing us to apply Model 2 to formulate a robust version of the revenue management problem, which can be efficiently solved within seconds using an off-the-shelf commercial solver as it is a convex problem.

Figure 2 compares the out-of-sample performance of our robust scheme (orange) with benchmarks. Due to the non-smoothness of the loss function, the AMKL approach is not applicable in this setting. As shown in [41], when the true distribution map is available, the optimal price (green) can be derived in closed form, achieving the highest expected profit for each fixed quantity $q$ in the left subplot. Similarly, the non-robust price (blue) can be obtained by treating the surrogate model as the true model. The confidence region in the left subplot, representing the 10th-90th percentile range of 100 independent tests, and the smaller optimality gap relative to the mean profit induced by the optimal price in the right subplot show that our robust price consistently outperforms the benchmark.

## 4.3   Demand Response Portfolio Optimization

We evaluate our robust scheme in a power system application, focusing on demand response (DR) portfolio optimization [51, 60]. In electricity markets, consumers capable of lowering electricity consumption during certain periods are called DR resources. In this experiment, we consider a DR aggregator (decision-maker) managing $n$ DR resources over a planning horizon of $T$ periods. The goal is to maximize the expected profit by determining commitment level $\boldsymbol{\theta}_t \in \mathbb{R}^n_+$ to meet a required deterministic demand reduction $D_t$ at each time $t \in [T]$.

The challenge lies in the uncertainty of DR resource's performance, where the scheduled commitment level $\boldsymbol{\theta}_t$ may significantly differ from the actual reduction level $\tilde{\boldsymbol{\theta}}_t$ due to random noise $\tilde{\boldsymbol{z}}_t$. This noise is decision-dependent, as larger commitments lead to higher variability. In this experiment, we model the actual reduction of each resource $i$ as $\tilde{\theta}_{t,i} = \theta_{t,i}\tilde{z}_{t,i}$ for all $i \in [n]$. Here, the multiplicative noise follows a beta distribution whose parameters $\alpha$ and $\beta$ depend linearly on the commitment level: $\tilde{z}_{t,i} \sim 2 \cdot \text{Beta}(\alpha = \beta = a_i\theta_{t,i} + b_i)$ with $a_i < 0$. This beta distribution has a support of $[0, 2]$ and a mean of 1, regardless of the value of $a_i\theta_{t,i} + b_i$. Therefore, the decision $\theta_{t,i}$ only influences the distribution shape, with higher commitment levels leading to heavier tails, hence, higher variability of the actual reduction level $\tilde{\theta}_{t,i}$.

We consider three DR resources with distinct characteristics: Resource 1 has high revenue but large variability, Resource 2 has low revenue but high predictability, and Resource 3 offers a balanced trade-off between the two. We follow the experiment setup in [12], including the loss function and the values of unit revenue, over-commitment cost, and under-commitment penalty. As their loss function is piecewise-linear, the resulting optimization problem can be formulated using Model 2 and efficiently solved. For further details, we refer the reader to the original paper. We conduct two out-of-sample tests: low and high demand loads over the planning horizon, corresponding to small

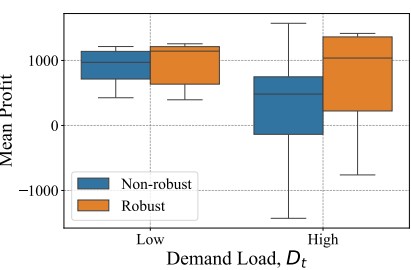

Figure 3: Out-of-sample performance of DR scheduling

$D_t$ and large $D_t$, respectively. Figure 3 compares the mean profits of our robust scheme (orange) and the non-robust scheme (blue), showing that the robust approach outperforms the non-robust one. Notably, under high demand, the non-robust scheme often performs significantly poorly, resulting in losses in some cases.

## 5   Concluding Remarks

We have presented the first Wasserstein distributionally robust optimization framework for performative optimization. In contrast to existing approaches, our framework accommodates a broader class of problems in decision-making under uncertainty, thereby extending the original scope of performative prediction. We proposed an efficient algorithm and established its convergence and suboptimality guarantees. To our knowledge, these theoretical results have not been previously established in the literature on robust performative prediction. Our experimental results demonstrate the superiority of our approach over existing methods on the standard strategic classification benchmark, as well as in two decision-making applications: revenue management and demand response portfolio optimization.

Notably, as the ambiguity set radius $\rho$ approaches zero, the robust objective coincides with the non-robust counterpart, which more directly targets the performative risk. This observation suggests a possible direction: designing algorithms that gradually shrink the ambiguity set over time, potentially trading robustness for improved approximation of the true performative risk as more information becomes available. Another promising direction is contextual performative optimization, where incorporating side information could further improve decision quality by enabling more accurate modeling of uncertainty.

**Broader impacts.** Our framework extends the scope of performative prediction beyond its original focus, enabling its application to a wider range of decision-making problems. In high-stakes settings, adopting a distributionally robust optimization perspective allows our approach to prioritize safe and reliable deployment in the presence of uncertainty and potential adversarial conditions.

## Acknowledgments and Disclosure of Funding

Grani A. Hanasusanto is supported in part by the National Science Foundation (NSF) under Grants CCF-2343869 and ECCS-2404413. Roy Dong is supported in part by NSF under Grant CCF-2236484. Yijie Wang is supported in part by the Fundamental Research Funds for the Central Universities. We thank Hyuk Park for assistance with the numerical experiments and the anonymous reviewers for their constructive feedback that helped improve this work.

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

# Technical Appendices and Supplementary Material

## A  Preliminary Definitions

**Definition 5** (Generalized strong convexity). We say that a loss function $\ell(\boldsymbol{z}, \boldsymbol{\theta})$ is $\gamma$-strongly convex in $\boldsymbol{\theta}$ if

$$\ell(\boldsymbol{z}, \boldsymbol{\theta}) \geq \ell(\boldsymbol{z}, \boldsymbol{\theta}') + \nabla_{\boldsymbol{\theta}}\ell(\boldsymbol{z}, \boldsymbol{\theta}')^{\top}(\boldsymbol{\theta} - \boldsymbol{\theta}') + \frac{\gamma}{2}\left\|\boldsymbol{\theta} - \boldsymbol{\theta}'\right\|_2^2, \tag{B1}$$

for all $\boldsymbol{\theta}, \boldsymbol{\theta}' \in \boldsymbol{\Theta}$ and $\boldsymbol{z} \in \mathcal{Z}$. If $\gamma = 0$, this condition reduces to the standard definition of convexity. We will also use the following equivalent definition of strong convexity. A loss function $\ell(\boldsymbol{z}, \boldsymbol{\theta})$ is $\gamma$-strongly convex in $\boldsymbol{\theta}$ if the function

$$f(\boldsymbol{z}, \boldsymbol{\theta}) = \ell(\boldsymbol{z}, \boldsymbol{\theta}) - \frac{\gamma}{2}\|\boldsymbol{\theta}\|_2^2 \tag{B1'}$$

is convex for all $\boldsymbol{z} \in \mathcal{Z}$.

**Definition 6** (Smoothness). We say that a loss function $\ell(\boldsymbol{z}, \boldsymbol{\theta})$ is $\beta$-smooth if the gradient $\nabla_{\boldsymbol{\theta}}\ell(\boldsymbol{z}, \boldsymbol{\theta})$ is $\beta$-Lipschitz in $\boldsymbol{z}$, that is

$$\left\|\nabla_{\boldsymbol{\theta}}\ell(\boldsymbol{z}, \boldsymbol{\theta}) - \nabla_{\boldsymbol{\theta}}\ell(\boldsymbol{z}', \boldsymbol{\theta})\right\|_2 \leq \beta\left\|\boldsymbol{z} - \boldsymbol{z}'\right\|_2, \tag{B2}$$

for all $\boldsymbol{z}, \boldsymbol{z}' \in \mathcal{Z}$.

**Definition 7** ($\epsilon$-sensitivity). We say that a distribution map $\mathbb{P}(\cdot)$ is $\epsilon$-*sensitive* if for all $\theta, \theta' \in \Theta$

$$\mathbb{W}_1\big(\mathbb{P}(\boldsymbol{\theta}), \mathbb{P}(\boldsymbol{\theta}')\big) \leq \epsilon\|\boldsymbol{\theta} - \boldsymbol{\theta}'\|_2, \tag{B3}$$

where $\mathbb{W}_1$ denotes the 1-Wasserstein metric.

## B  Background on Exponential Smoothing

Let $\mathcal{Z} = \{z_1, \ldots, z_n\}$ denote a finite support set. Given a loss function $\ell : \Theta \times \mathcal{Z} \to \mathbb{R}$ and a distribution $\hat{\boldsymbol{p}} \in \Delta_n$, where $\Delta_n$ denotes the probability simplex. We define the exponentially smoothed objective as:

$$f_\mu(\theta) := \mu \cdot \log\left(\sum_{i=1}^n \hat{p}_i \cdot \exp\left(\frac{\ell(\theta, z_i)}{\mu}\right)\right),$$

where $\mu > 0$ is a smoothing parameter.

This function, also known as the log-sum-exp function, is a widely used smooth approximation to the pointwise maximum function and has well-established properties in convex analysis [Bertsekas, 2015; Section 2.2]. This function provides a smooth approximation to $\max_i \ell(\theta, z_i)$ whenever $\hat{p}_i > 0$ for all $i$.

**Properties.** The function $f_\mu(\theta)$ satisfies the following:

- Approximation Bounds: The approximation error can be precisely quantified:

$$\max_i \ell(\theta, z_i) \leq f_\mu(\theta) \leq \max_i \ell(\theta, z_i) + \mu \cdot \log\left(\frac{1}{\min_i \hat{p}_i}\right),$$

  provided $\hat{p}_i > 0$ for all $i$. Thus, as $\mu \to 0$, the smoothed objective $f_\mu(\theta)$ approaches the exact maximum, with the error vanishing linearly in $\mu$ up to a logarithmic multiplicative factor.

- Convexity and Differentiability: If each function $\ell(\theta, z_i)$ is convex in $\theta$, then $f_\mu(\theta)$ is also convex, as it is a composition of convex functions closed under nonnegative weighted log-sum-exp operations. Moreover, $f_\mu(\theta)$ is continuously differentiable for all $\mu > 0$, with gradient:

$$\nabla_\theta f_\mu(\theta) = \sum_{i=1}^n \pi_\mu(z_i; \theta) \cdot \nabla_\theta \ell(\theta, z_i),$$

  where the weight vector $\pi_\mu(\cdot; \theta) \in \Delta_n$ defines a softmax distribution:

$$\pi_\mu(z_i; \theta) := \frac{\hat{p}_i \cdot \exp\left(\ell(\theta, z_i)/\mu\right)}{\sum_{j=1}^n \hat{p}_j \cdot \exp\left(\ell(\theta, z_j)/\mu\right)}.$$

This smoothing mechanism not only ensures differentiability but also facilitates efficient computation of gradients for robust optimization objectives involving maxima.

## C   Auxiliary Lemmas

**Lemma C.1** (First-order optimality condition; Section 4.2.3 in [9])**.** *Let $f$ be a convex function and let $\mathbf{\Omega}$ be a closed convex set on which $f$ is differentiable, then*

$$\mathbf{x}^{\star} \in \arg\min_{\mathbf{x} \in \mathbf{\Omega}} f(\mathbf{x})$$

*if and only if*

$$\nabla f(\mathbf{x}^{\star})^{T}(\mathbf{y} - \mathbf{x}^{\star}) \geq 0 \quad \forall \mathbf{y} \in \mathbf{\Omega}.$$

**Lemma C.2** ([26])**.** *A distribution map $\mathbb{P}(\cdot)$ is $\epsilon$-sensitive if and only if for all $\boldsymbol{\theta}, \boldsymbol{\theta}' \in \mathbf{\Theta}$, we have*

$$\sup_{g \in \mathcal{L}} \left| \mathbb{E}_{\mathbb{P}(\boldsymbol{\theta})}[g(\tilde{\mathbf{z}})] - \mathbb{E}_{\mathbb{P}(\boldsymbol{\theta}')}[g(\tilde{\mathbf{z}})] \right| \leq \epsilon L \|\boldsymbol{\theta} - \boldsymbol{\theta}'\|_{2},$$

*where*

$$\mathcal{L} = \{g : \mathbb{R}^{p} \to \mathbb{R} \mid |g(\mathbf{z}) - g(\mathbf{z}')| \leq L \|\mathbf{z} - \mathbf{z}'\|_{2} \quad \forall \mathbf{z}, \mathbf{z}' \in \mathcal{Z}\}$$

*is the space of all $L$-Lipschitz continuous functions.*

**Lemma C.3.** *If $\ell(\mathbf{z}, \boldsymbol{\theta})$ is $\gamma$-strongly convex, then the worst case expectation*

$$\mathbb{J}_{\boldsymbol{\eta}}(\boldsymbol{\theta}) = \sup_{\mathbb{Q} \in \mathbb{B}(\hat{\mathbb{P}}(\boldsymbol{\eta}))} \mathbb{E}_{\mathbb{Q}}[\ell(\tilde{\mathbf{z}}, \boldsymbol{\theta})]$$

*is $\gamma$-strongly convex in $\boldsymbol{\theta}$.*

*Proof.* By the equivalent definition of $\gamma$-strong convexity in (B1'), we have that $\ell(\mathbf{z}, \boldsymbol{\theta}) - \frac{\gamma}{2}\|\boldsymbol{\theta}\|^{2}$ is convex in $\boldsymbol{\theta}$. Hence, the worst-case expectation

$$\sup_{\mathbb{Q} \in \mathbb{B}(\hat{\mathbb{P}}(\boldsymbol{\eta}))} \mathbb{E}_{\mathbb{Q}}\left[\ell(\tilde{\mathbf{z}}, \boldsymbol{\theta}) - \frac{\gamma}{2}\|\boldsymbol{\theta}\|_{2}^{2}\right]$$

is convex in $\boldsymbol{\theta}$ since the expectation and the pointwise supremum operations preserve convexity. Thus, we have

$$\mathbb{J}_{\boldsymbol{\eta}}(\boldsymbol{\theta}) = \sup_{\mathbb{Q} \in \mathbb{B}(\hat{\mathbb{P}}(\boldsymbol{\eta}))} \mathbb{E}_{\mathbb{Q}}\left[\ell(\tilde{\mathbf{z}}, \boldsymbol{\theta}) - \frac{\gamma}{2}\|\boldsymbol{\theta}\|_{2}^{2}\right] + \frac{\gamma}{2}\|\boldsymbol{\theta}\|_{2}^{2}$$

is $\gamma$-strongly convex in $\boldsymbol{\theta}$ by the definition (B1'). $\qquad\square$

**Lemma C.4.** *Let $\mathbf{x} \in \mathbb{R}^{J}$ and define the smooth maximum function*

$$f_{\mu}(\mathbf{x}) = \mu \log\left(\sum_{j \in [J]} e^{x_{j}/\mu}\right)$$

*for any $\mu > 0$. Then the following bounds hold:*

$$\max_{j \in [J]} x_{j} \leq f_{\mu}(\mathbf{x}) \leq \max_{j \in [J]} x_{j} + \mu \log J.$$

*Proof.* Let $M := \max_{j \in [J]} x_{j}$. Then for each $j \in [J]$, we have $x_{j} \leq M$, and hence:

$$\sum_{j \in [J]} e^{x_{j}/\mu} \leq \sum_{j \in [J]} e^{M/\mu} = J e^{M/\mu}.$$

Taking logarithms and multiplying by $\mu$, we obtain the upper bound:

$$f_{\mu}(\mathbf{x}) = \mu \log\left(\sum_{j \in [J]} e^{x_{j}/\mu}\right) \leq \mu \log\left(J e^{M/\mu}\right) = \mu \log J + M. \tag{8}$$

For the lower bound, observe that

$$\sum_{j\in[J]} e^{x_j/\mu} \geq e^{M/\mu}$$

since at least one term in the sum equals $e^{M/\mu}$. Thus:

$$f_\mu(\boldsymbol{x}) = \mu \log \left( \sum_{j\in[J]} e^{x_j/\mu} \right) \geq \mu \log \left( e^{M/\mu} \right) = M \tag{9}$$

Combining both bounds in (8) and (9), we have the desired result:

$$\max_{j\in[J]} x_j \leq f_\mu(\boldsymbol{x}) \leq \max_{j\in[J]} x_j + \mu \log J.$$

$\square$

## D   Deferred Proofs Related to Reformulations

### D.1   Proof of Reformulation for Model 1

*Proof.* We begin by rewriting the random parameters $\tilde{\boldsymbol{Z}}$ as

$$\tilde{\boldsymbol{Z}} = \begin{bmatrix} \tilde{\boldsymbol{Y}} & \tilde{\boldsymbol{z}} \\ \tilde{\boldsymbol{z}}^\top & \tilde{z}^0 \end{bmatrix} \in \mathbb{S}^{N+1}.$$

Next, we introduce the matrix variable

$$\boldsymbol{\Gamma} = \begin{bmatrix} \boldsymbol{\theta}\boldsymbol{\theta}^\top & \boldsymbol{\theta} \\ \boldsymbol{\theta}^\top & 1 \end{bmatrix}, \tag{10}$$

which allows us to rewrite the loss function as $\ell(\boldsymbol{Z}, \boldsymbol{\theta}) = \mathcal{L}(\langle \boldsymbol{\Gamma}, \boldsymbol{Z} \rangle)$. According to [8, Remark 1], the worst-case expected loss over a 1-Wasserstein ambiguity set $\mathbb{B}(\hat{\mathbb{P}}(\boldsymbol{\theta}_t))$, with cost induced by the Schatten-$\infty$ norm, is given by:

$$\sup_{\mathbb{Q}\in\mathbb{B}(\hat{\mathbb{P}}(\boldsymbol{\theta}_t))} \mathbb{E}_\mathbb{Q}[\ell(\tilde{\boldsymbol{Z}}, \boldsymbol{\theta})] = \inf_{\lambda\in\mathbb{R}_+} \rho\lambda + \mathbb{E}_{\hat{\mathbb{P}}(\boldsymbol{\theta}_t)} \left[ \sup_{\boldsymbol{Z}\in\mathbb{S}^{N+1}} \mathcal{L}(\langle \boldsymbol{\Gamma}, \boldsymbol{Z} \rangle) - \lambda\|\boldsymbol{Z} - \tilde{\boldsymbol{Z}}\|_\infty \right].$$

By applying [48, Lemma 47], the inner maximization problem can be simplified as:

$$\sup_{\boldsymbol{Z}\in\mathbb{S}^{N+1}} \mathcal{L}(\langle \boldsymbol{\Gamma}, \boldsymbol{Z} \rangle) - \lambda\|\boldsymbol{Z} - \hat{\boldsymbol{Z}}\|_\infty = \begin{cases} \mathcal{L}(\langle \boldsymbol{\Gamma}, \hat{\boldsymbol{Z}} \rangle) & \text{if } L\|\boldsymbol{\Gamma}\|_1 \leq \lambda \\ +\infty & \text{otherwise.} \end{cases}$$

Hence, the worst-case expectation reduces to

$$\sup_{\mathbb{Q}\in\mathbb{B}(\hat{\mathbb{P}}(\boldsymbol{\theta}_t))} \mathbb{E}_\mathbb{Q}[\ell(\tilde{\boldsymbol{Z}}, \boldsymbol{\theta})] = \rho L\|\boldsymbol{\Gamma}\|_1 + \mathbb{E}_{\hat{\mathbb{P}}(\boldsymbol{\theta}_t)}[\mathcal{L}(\langle \boldsymbol{\Gamma}, \tilde{\boldsymbol{Z}} \rangle)].$$

To conclude, we observe that

$$\|\boldsymbol{\Gamma}\|_1 = \left\| \begin{bmatrix} \boldsymbol{\theta}\boldsymbol{\theta}^\top & \boldsymbol{\theta} \\ \boldsymbol{\theta}^\top & 1 \end{bmatrix} \right\|_1 = \text{tr} \left( \begin{bmatrix} \boldsymbol{\theta}\boldsymbol{\theta}^\top & \boldsymbol{\theta} \\ \boldsymbol{\theta}^\top & 1 \end{bmatrix} \right) = \|(\boldsymbol{\theta}, 1)\|_2^2.$$

This completes the proof. $\square$

### D.2   Proof of Reformulation for Model 2

*Proof.* Rewriting the random parameters $\tilde{\boldsymbol{Z}}$ as

$$\tilde{\boldsymbol{Z}} = \begin{bmatrix} \tilde{\boldsymbol{Y}} & \tilde{\boldsymbol{z}} \\ \tilde{\boldsymbol{z}}^\top & \tilde{z}^0 \end{bmatrix} \in \mathbb{S}^{N+1},$$

and introducing the matrix variable

$$\boldsymbol{\Gamma}_j = \begin{bmatrix} \boldsymbol{a}_j(\boldsymbol{\theta})\boldsymbol{a}_j(\boldsymbol{\theta})^\top & \boldsymbol{b}_j(\boldsymbol{\theta}) \\ \boldsymbol{b}_j(\boldsymbol{\theta})^\top & c_j(\boldsymbol{\theta}) \end{bmatrix}, \tag{11}$$

allow us to rewrite $Q_j(\boldsymbol{Z}, \boldsymbol{\theta})$ as $\langle \boldsymbol{\Gamma}_j, \boldsymbol{Z} \rangle$. By [35, Remark 6.6], the robust decoupled performative risk can be expressed as

$$\mathbb{J}_{\boldsymbol{\theta}_t}(\boldsymbol{\theta}) = \sup_{\mathbb{Q} \in \mathbb{B}(\hat{\mathbb{P}}(\boldsymbol{\theta}_t))} \mathbb{E}_{\mathbb{Q}}[\ell(\tilde{\boldsymbol{Z}}, \boldsymbol{\theta})] = \sum_{s \in [S]} \hat{p}_s(\boldsymbol{\theta}_t) \left[ \max_{j \in [J]} Q_j(\hat{\boldsymbol{Z}}_s, \boldsymbol{\theta}) \right] + \max_{j \in [J]} \rho \|\boldsymbol{\Gamma}_j\|_1,$$

where

$$\|\boldsymbol{\Gamma}_j\|_1 = \left\| \begin{bmatrix} \boldsymbol{a}_j(\boldsymbol{\theta})\boldsymbol{a}_j(\boldsymbol{\theta})^\top & \boldsymbol{b}_j(\boldsymbol{\theta}) \\ \boldsymbol{b}_j(\boldsymbol{\theta})^\top & c_j(\boldsymbol{\theta}) \end{bmatrix} \right\|_1 = \mathrm{tr}\left( \begin{bmatrix} \boldsymbol{a}_j(\boldsymbol{\theta})\boldsymbol{a}_j(\boldsymbol{\theta})^\top & \boldsymbol{b}_j(\boldsymbol{\theta}) \\ \boldsymbol{b}_j(\boldsymbol{\theta})^\top & c_j(\boldsymbol{\theta}) \end{bmatrix} \right) = \boldsymbol{a}_j(\boldsymbol{\theta})^\top \boldsymbol{a}_j(\boldsymbol{\theta}) + c_j(\boldsymbol{\theta}).$$

Applying the exponential smoothing techniques described in [7, Section 2.2], we obtain the following smooth approximation of $\mathbb{J}_{\boldsymbol{\eta}}(\boldsymbol{\theta})$;

$$\mathbb{J}_{\boldsymbol{\theta}_t}^\mu(\boldsymbol{\theta}) = \mu \sum_{s \in [S]} \hat{p}_s(\boldsymbol{\theta}_t) \log \left( \sum_{j \in [J]} e^{Q_j(\hat{\boldsymbol{Z}}_s, \boldsymbol{\theta})/\mu} \right) + \mu \log \left( \sum_{j \in [J]} e^{\rho \|\boldsymbol{\Gamma}_j\|_1/\mu} \right), \qquad (12)$$

where $\mu \in \mathbb{R}_{++}$ is smoothing parameter. Introducing epigraphical variables, we reformulate the objective function as the optimal value of the convex program

$$
\begin{aligned}
\inf \quad & \sum_{s \in [S]} \hat{p}_s(\boldsymbol{\theta}_t) t_s + t_{S+1} \\
\mathrm{s.\,t.} \quad & t_s \in \mathbb{R} && \forall s \in [S+1] \\
& \zeta_{s,j} \in \mathbb{R} && \forall s \in [S+1]\, j \in [J] \\
& \rho \|\boldsymbol{\Gamma}_j\|_1 \leq \zeta_{S+1,j} && \forall j \in [J] \\
& Q_j(\hat{\boldsymbol{Z}}_s, \boldsymbol{\theta}) \leq \zeta_{s,j} && \forall s \in [S+1]\, j \in [J] \\
& \mu \log \left( \sum_{j \in [J]} e^{\zeta_{s,j}/\mu} \right) \leq t_s && \forall s \in [S+1].
\end{aligned}
\qquad (13)
$$

The last constraint of (13) is equivalent to $\sum_{j \in [J]} \mu e^{\zeta_{s,j}/\mu - t_s/\mu} \leq \mu$ and can be reformulated using the exponential cone:

$$\sum_{j \in [J]} r_{s,j} \leq \mu, \quad (r_{s,j}, \mu, \zeta_{s,j} - t_s) \in K_{\exp} \forall j \in [J]$$

where the exponential cone $K_{\exp}$ is defined as

$$K_{\exp} = \{(x_1, x_2, x_3) : x_1 \geq x_2 e^{x_3/x_2}, x_2 > 0\} \cup \{(x_1, 0, x_3) : x_1 \geq 0, x_3 \leq 0\}.$$

To complete the proof, we substitute the expressions for $\|\boldsymbol{\Gamma}_j\|_1 \ \forall j \in [J]$ with

$$\boldsymbol{\theta}^\top \overline{\boldsymbol{A}}_j^\top \overline{\boldsymbol{A}}_j \boldsymbol{\theta} + (2\overline{\boldsymbol{a}}_j^\top \overline{\boldsymbol{A}}_j + \overline{\boldsymbol{c}}_j^\top)\boldsymbol{\theta} + \overline{\boldsymbol{a}}_j^\top \overline{\boldsymbol{a}}_j + \overline{c}_{j0},$$

and for $Q_j(\hat{\boldsymbol{Z}}_s, \boldsymbol{\theta}) \ \forall s \in [S]\, j \in [J]$ with

$$\boldsymbol{\theta}^\top \overline{\boldsymbol{A}}_j^\top \hat{\boldsymbol{Y}}_s \overline{\boldsymbol{A}}_j \boldsymbol{\theta} + (2\overline{\boldsymbol{a}}_j^\top \hat{\boldsymbol{Y}}_s \overline{\boldsymbol{A}}_j + 2\hat{\boldsymbol{z}}_s^{0\top} \overline{\boldsymbol{B}}_j + \hat{z}_s^0 \overline{\boldsymbol{c}}_j^\top)\boldsymbol{\theta} + \overline{\boldsymbol{a}}_j^\top \hat{\boldsymbol{Y}}_s \overline{\boldsymbol{a}}_j + 2\overline{\boldsymbol{b}}_j^\top \hat{\boldsymbol{z}}_s + \hat{z}_s^0 \overline{c}_{j0}.$$

This completes the proof. $\qquad\qquad\qquad\qquad\qquad\qquad\qquad\qquad\qquad\qquad\qquad\qquad \square$

### D.3 Proof of Reformulation for Model 3

*Proof.* By using Definition 1, the robust decoupled performative risk $\mathbb{J}_{\boldsymbol{\theta}_t}(\boldsymbol{\theta})$ can be rewritten as

$$
\begin{aligned}
\mathbb{J}_{\boldsymbol{\theta}_t}(\boldsymbol{\theta}) &= \sup_{\mathbb{Q} \in \mathbb{B}(\hat{\mathbb{P}}(\boldsymbol{\theta}_t))} \mathbb{E}_{\mathbb{Q}}[\ell(\tilde{\boldsymbol{z}}, \boldsymbol{\theta})] \\
&= \begin{cases} \displaystyle \sup_{\mathbb{Q}_s \in \mathcal{M}(\mathcal{Z})} \sum_{s \in [S]} \hat{p}_s(\boldsymbol{\theta}_t) \int_{\mathcal{Z}} \ell(\boldsymbol{z}, \boldsymbol{\theta}) \mathbb{Q}_s(\mathrm{d}\boldsymbol{z}) \\ \mathrm{s.\,t.} \quad \displaystyle \sum_{s \in [S]} \hat{p}_s(\boldsymbol{\theta}_t) \int_{\mathcal{Z}} \|\boldsymbol{z} - \hat{\boldsymbol{z}}_s\|_2^2 \mathbb{Q}_s(\mathrm{d}\boldsymbol{z}) \leq \rho^2. \end{cases}
\end{aligned}
$$

where $\mathcal{M}(\mathcal{Z})$ denotes the space of all probability distributions $\mathbb{Q}$ supported on $\mathcal{Z}$ satisfying $\mathbb{E}_{\mathbb{Q}}[\|z\|_2^2] = \int_{\mathcal{Z}} \|z\|_2^2 \mathbb{Q}(dz) < \infty$. This reformulation follows from the law of total probability, where $\mathbb{Q}_s$ represents the conditional distribution of $\tilde{z}$ given that the scenario $\hat{z}_s(\theta_t)$ is realized. Using the Lagrangian, we have

$$\mathbb{J}_{\theta_t}(\theta) = \sup_{\mathbb{Q}_s \in \mathcal{M}(\mathcal{Z})} \inf_{\lambda \in \mathbb{R}_+} \sum_{s \in [S]} \hat{p}_s(\theta_t) \int_{\mathcal{Z}} \ell(z, \theta) \mathbb{Q}_s(dz)$$
$$+ \lambda \left( \rho^2 - \sum_{s \in [S]} \hat{p}_s(\theta_t) \int_{\mathcal{Z}} \|z - \hat{z}_s\|_2^2 \mathbb{Q}_s(dz) \right).$$

By the minimax theorem [53], which is valid under the assumption that $\ell$ is upper-semicontinuous and concave in $z$, and the support set $\mathcal{Z}$ is convex, we can exchange the supremum and infimum to obtain:

$$\mathbb{J}_{\theta_t}(\theta) = \inf_{\lambda \in \mathbb{R}_+} \sup_{\mathbb{Q}_s \in \mathcal{M}(\mathcal{Z})} \rho^2 \lambda + \sum_{s \in [S]} \hat{p}_s(\theta_t) \int_{\mathcal{Z}} \left( \ell(z, \theta) - \lambda \|z - \hat{z}_s\|_2^2 \right) \mathbb{Q}_s(dz).$$

From the fact that the space $\mathcal{M}(\mathcal{Z})$ contains all the Dirac distributions supported on $\mathcal{Z}$, we have

$$\mathbb{J}_{\theta_t}(\theta) = \inf_{\lambda \in \mathbb{R}_+} \rho^2 \lambda + \sum_{s \in [S]} \hat{p}_s(\theta_t) \sup_{z \in \mathcal{Z}} (\ell(z, \theta) - \lambda \|z - \hat{z}_s\|_2^2).$$

Adding a regularized term $\tau \lambda^2$ where $\tau \in \mathbb{R}_{++}$ is a positive constant, we have

$$\mathbb{J}_{\theta_t}(\theta) \leq \inf_{\lambda \in \mathbb{R}_+} \sum_{s \in [S]} \hat{p}_s(\theta_t) \sup_{z \in \mathcal{Z}} (\ell(z, \theta) - \lambda \|z - \hat{z}_s\|_2^2 + \rho^2 \lambda + \tau \lambda^2).$$

Therefore, minimizing the right-hand side provides an upper bound on $\mathbb{J}_{\theta_t}(\theta)$. Next, we introduce auxiliary variables $t_s \ \forall s \in [S]$, which yields the equivalent formulation for the right hand side of the above inequality

$$\begin{aligned}
\inf \quad & \sum_{s \in [S]} \hat{p}_s(\theta_t) t_s + \rho^2 \lambda + \tau \lambda^2 \\
\text{s.t.} \quad & \lambda \in \mathbb{R}_+, \ t_s \in \mathbb{R} \ \forall s \in [S] \\
& \sup_{z \in \mathcal{Z}} (\ell(z, \theta) - \lambda \|z - \hat{z}_s\|_2^2) \leq t_s \ \forall s \in [S].
\end{aligned} \tag{14}$$

By the definition of conjugate functions, we have

$$\sup_{z \in \mathcal{Z}} (\ell(z, \theta) - \lambda \|z - \hat{z}_s\|_2^2) = [-\ell + \chi_{\mathcal{Z}} + \lambda \|z - \hat{z}_s\|_2^2]^*(0),$$

where $\chi_{\mathcal{Z}}$ denotes the characteristic function of $\mathcal{Z}$. Based on results from [35, Theorem 4.2], [46, Theorem 11.23], and [64, Lemma B.8], the conjugate functions of infimal convolutions and 2-norm balls is given by

$$[-\ell + \chi_{\mathcal{Z}} + \lambda \|z - \hat{z}_s\|_2^2]^*(0) = \inf_{r_s, \zeta_s} ([-\ell]^*(r_s - \zeta_s, \theta) + \sigma_{\mathcal{Z}}(\zeta_s) + [\lambda \|z - \hat{z}_s\|_2^2]^*(-r_s)).$$

with

$$[\lambda \|z - \hat{z}_s\|_2^2]^*(-r_s) = \sup_{v_s} (-r_s^\top v_s - \lambda \|z - \hat{z}_s\|_2^2) = -r_s^\top \hat{z}_s + \frac{1}{4\lambda} \|r_s\|_2^2,$$

Substituting this back into the formulation (14), we thus obtain that $\mathbb{J}_{\theta_t}(\theta)$ is upper bounded by the optimal value of the following convex program:

$$\begin{aligned}
\inf \quad & \sum_{s \in [S]} \hat{p}_s(\theta_t) \left( [-\ell]^*(r_s - \zeta_s, \theta) + \sigma_{\mathcal{Z}}(\zeta_s) - r_s^\top \hat{z}_s + \frac{1}{4\lambda} \|r_s\|_2^2 \right) + \rho^2 \lambda + \tau \lambda^2 \\
\text{s.t.} \quad & \lambda \in \mathbb{R}_+, \ t_s \in \mathbb{R} \ \forall s \in [S], \ r_s \in \mathbb{R}^m \ \forall s \in [S], \ \zeta_s \in \mathbb{R}^m \ \forall s \in [S].
\end{aligned}$$

Combining with outer minimization over $\theta \in \Theta$ completes the proof. □

# E   Deferred Proofs Related to Convergence

**Lemma E.1.** *Consider the loss function defined in Model 2 and assume $\boldsymbol{\Theta}$ is bounded. We define the smoothed loss function*

$$\ell_\mu(\boldsymbol{Z}, \boldsymbol{\theta}) = \mu \log \left( \sum_{j \in [J]} e^{Q_j(\boldsymbol{Z}, \boldsymbol{\theta})/\mu} \right), \tag{15}$$

*where $\boldsymbol{Z} \sim \hat{\mathbb{P}}(\boldsymbol{\theta}_t)$ satisfies and $\|\boldsymbol{Z}\|_2 \leq k_1$ for some constant $k_1 < \infty$. Then the gradient $\nabla_{\boldsymbol{\theta}} \ell_\mu(\boldsymbol{Z}, \boldsymbol{\theta})$ is $\beta$-Lipschitz in $\boldsymbol{Z}$ for $\beta = Jdk_3 \left( \frac{k_1 k_2}{\mu} + 1 \right)$ for some constants $k_2, k_3 < \infty$ defined below.*

*Proof.* Since the coefficients $\boldsymbol{a}_j(\boldsymbol{\theta}), \boldsymbol{b}_j(\boldsymbol{\theta})$, and $c_j(\boldsymbol{\theta})$ are affine for all $j \in [J]$ and $\boldsymbol{\Theta}$ is bounded, there exist some constants $k_2, k_3 < \infty$ such that

- $\|\boldsymbol{\Gamma}_j\|_2 \leq k_2$ for all $j \in [J]$,

- $\left\| \nabla_{\theta_i} \boldsymbol{\Gamma}_j \right\|_2 \leq k_3$ for all $i \in [d], j \in [J]$.

By definition, we have the gradient of the smoothed loss (15):

$$\nabla_{\boldsymbol{\theta}} \ell_\mu(\boldsymbol{Z}, \boldsymbol{\theta}) = \sum_{j \in [J]} w_j(\boldsymbol{Z}, \boldsymbol{\theta}) \nabla_{\boldsymbol{\theta}} Q_j(\boldsymbol{Z}, \boldsymbol{\theta})$$

where the softmax weights are

$$w_j(\boldsymbol{Z}, \boldsymbol{\theta}) = \frac{e^{Q_j(\boldsymbol{Z}, \boldsymbol{\theta})/\mu}}{\sum_{i \in [J]} e^{Q_i(\boldsymbol{Z}, \boldsymbol{\theta})/\mu}},$$

and the gradient of $Q_j$ with respect to $\boldsymbol{\theta}$ is given by

$$\nabla_{\boldsymbol{\theta}} Q_j(\boldsymbol{Z}, \boldsymbol{\theta}) = 2\overline{\boldsymbol{A}}_j^\top \boldsymbol{Y} \overline{\boldsymbol{A}}_j \boldsymbol{\theta} + 2\overline{\boldsymbol{A}}_j^\top \boldsymbol{Y}^\top \overline{\boldsymbol{a}}_j + 2\overline{\boldsymbol{B}}_j^\top \boldsymbol{z} + z^0 \overline{\boldsymbol{c}}_j.$$

To prove that $\nabla_{\boldsymbol{\theta}} \ell_\mu$ is Lipschitz in $\boldsymbol{Z}$, we examine $\|\nabla_{\boldsymbol{\theta}} \ell_\mu(\boldsymbol{Z}_1, \boldsymbol{\theta}) - \nabla_{\boldsymbol{\theta}} \ell_\mu(\boldsymbol{Z}_2, \boldsymbol{\theta})\|_2$. We first decompose the difference and apply the triangle inequality

$$
\begin{aligned}
&\left\| \sum_{j \in [J]} w_j(\boldsymbol{Z}_1, \boldsymbol{\theta}) \nabla_{\boldsymbol{\theta}} Q_j(\boldsymbol{Z}_1, \boldsymbol{\theta}) - \sum_{j \in [J]} w_j(\boldsymbol{Z}_2, \boldsymbol{\theta}) \nabla_{\boldsymbol{\theta}} Q_j(\boldsymbol{Z}_2, \boldsymbol{\theta}) \right\|_2 \\
\leq\ & \left\| \sum_{j \in [J]} w_j(\boldsymbol{Z}_1, \boldsymbol{\theta}) \nabla_{\boldsymbol{\theta}} Q_j(\boldsymbol{Z}_1, \boldsymbol{\theta}) - \sum_{j \in [J]} w_j(\boldsymbol{Z}_2, \boldsymbol{\theta}) \nabla_{\boldsymbol{\theta}} Q_j(\boldsymbol{Z}_1, \boldsymbol{\theta}) \right\|_2 \\
& + \left\| \sum_{j \in [J]} w_j(\boldsymbol{Z}_2, \boldsymbol{\theta}) \nabla_{\boldsymbol{\theta}} Q_j(\boldsymbol{Z}_1, \boldsymbol{\theta}) - \sum_{j \in [J]} w_j(\boldsymbol{Z}_2, \boldsymbol{\theta}) \nabla_{\boldsymbol{\theta}} Q_j(\boldsymbol{Z}_2, \boldsymbol{\theta}) \right\|_2 \\
=\ & \left\| \sum_{j \in [J]} (w_j(\boldsymbol{Z}_1, \boldsymbol{\theta}) - w_j(\boldsymbol{Z}_2, \boldsymbol{\theta})) \nabla_{\boldsymbol{\theta}} Q_j(\boldsymbol{Z}_1, \boldsymbol{\theta}) \right\|_2 + \left\| \sum_{j \in [J]} (\nabla_{\boldsymbol{\theta}} Q_j(\boldsymbol{Z}_1, \boldsymbol{\theta}) - \nabla_{\boldsymbol{\theta}} Q_j(\boldsymbol{Z}_2, \boldsymbol{\theta})) \right\|_2.
\end{aligned}
\tag{16}
$$

Next, we bound the terms involved:

- Weight difference:

$$
\begin{aligned}
\|w_j(\boldsymbol{Z}_1, \boldsymbol{\theta}) - w_j(\boldsymbol{Z}_2, \boldsymbol{\theta})\|_2 \; &= \left\| \frac{e^{Q_j(\boldsymbol{Z}_1, \boldsymbol{\theta})/\mu}}{\sum_{i \in [J]} e^{Q_i(\boldsymbol{Z}_1, \boldsymbol{\theta})/\mu}} - \frac{e^{Q_j(\boldsymbol{Z}_2, \boldsymbol{\theta})/\mu}}{\sum_{i \in [J]} e^{Q_i(\boldsymbol{Z}_2, \boldsymbol{\theta})/\mu}} \right\|_2 \\
&\overset{(a)}{\leq} \frac{1}{\mu} \|Q_j(\boldsymbol{Z}_1, \boldsymbol{\theta}) - Q_j(\boldsymbol{Z}_2, \boldsymbol{\theta})\|_2 \\
&= \frac{1}{\mu} \|\langle \boldsymbol{\Gamma}_j, \boldsymbol{Z}_1 \rangle - \langle \boldsymbol{\Gamma}_j, \boldsymbol{Z}_2 \rangle\|_2 \\
&\overset{(b)}{\leq} \frac{1}{\mu} \|\boldsymbol{\Gamma}_j\|_2 \|\boldsymbol{Z}_1 - \boldsymbol{Z}_2\|_2 \\
&\leq \frac{k_2}{\mu} \|\boldsymbol{Z}_1 - \boldsymbol{Z}_2\|_2,
\end{aligned}
$$

where (a) comes from the Lipschitz continuity of the softmax function [17], and (b) uses the Cauchy–Schwarz inequality.

- Gradient:

$$
\|\nabla_{\boldsymbol{\theta}} Q_j(\boldsymbol{Z}, \boldsymbol{\theta})\|_2 = \sum_{i \in [d]} \|\langle \nabla_{\theta_i} \boldsymbol{\Gamma}_j, \boldsymbol{Z} \rangle\|_2 \overset{(a)}{\leq} \sum_{i \in [d]} \|\nabla_{\theta_i} \boldsymbol{\Gamma}_j\|_2 \|\boldsymbol{Z}\|_2 \overset{(b)}{\leq} k_1 k_3 d,
$$

where $\nabla_{\theta_i} \boldsymbol{\Gamma}_j$ is a matrix whose $(i_1, i_2)$-th matrix slice is the gradient of the $(i_1, i_2)$-th component of the matrix with respect to $\theta_i$.

- Gradient difference:

$$
\|\nabla_{\boldsymbol{\theta}} Q_j(\boldsymbol{Z}_1, \boldsymbol{\theta}) - \nabla_{\boldsymbol{\theta}} Q_j(\boldsymbol{Z}_2, \boldsymbol{\theta})\|_2 \leq d \|\nabla_{\theta_i} \boldsymbol{\Gamma}_j\|_2 \|\boldsymbol{Z}_1 - \boldsymbol{Z}_2\|_2 \leq dk_3 \|\boldsymbol{Z}_1 - \boldsymbol{Z}_2\|_2.
$$

Substituting the above bounds into (16), we obtain

$$
\|\nabla_{\boldsymbol{\theta}} \ell_\mu(\boldsymbol{Z}_1, \boldsymbol{\theta}) - \nabla_{\boldsymbol{\theta}} \ell_\mu(\boldsymbol{Z}_2, \boldsymbol{\theta})\|_2 \leq Jdk_3 \left( \frac{k_1 k_2}{\mu} + 1 \right) \|\boldsymbol{Z}_1 - \boldsymbol{Z}_2\|_2.
$$

Thus, the claim follows. $\qquad \square$

**Lemma E.2.** *Consider the loss function defined in Model 2, and assume that $\overline{\boldsymbol{A}}_j^\top \overline{\boldsymbol{A}}_j \succ \boldsymbol{0}$ for all $j \in [J]$. Define the smoothed loss function*

$$
\ell_\mu^{\mathrm{reg}}(\boldsymbol{\theta}) = \mu \log \left( \sum_{j \in [J]} e^{\rho \|\boldsymbol{\Gamma}_j\|_1 / \mu} \right), \tag{17}
$$

*where $\boldsymbol{\Gamma}_j$ is defined in (11). Then $\ell_\mu^{\mathrm{reg}}(\boldsymbol{\theta})$ is $\rho\alpha$-strongly convex in $\boldsymbol{\theta}$ where*

$$
\alpha = 2 \min_{j \in [J]} \lambda_{\min}(\overline{\boldsymbol{A}}_j^\top \overline{\boldsymbol{A}}_j).
$$

*Proof.* By definition, the gradient of the smoothed loss (17) is given by

$$
\nabla_{\boldsymbol{\theta}} \ell_\mu^{\mathrm{reg}}(\boldsymbol{\theta}) = \rho \sum_{j \in [J]} w_j(\boldsymbol{\theta}) \nabla_{\boldsymbol{\theta}} \boldsymbol{\Gamma}_j(\boldsymbol{\theta}),
$$

where the softmax weights are defined as

$$
w_j(\boldsymbol{\theta}) = \frac{e^{\rho \boldsymbol{\Gamma}_j(\boldsymbol{\theta})/\mu}}{\sum_{i \in [J]} e^{\rho \boldsymbol{\Gamma}_i(\boldsymbol{\theta})/\mu}} \quad \forall j \in [J],
$$

and the gradient of $\boldsymbol{\Gamma}_j$ with respect to $\boldsymbol{\theta}$ is given by

$$
\nabla_{\boldsymbol{\theta}} \boldsymbol{\Gamma}_j(\boldsymbol{\theta}) = 2\overline{\boldsymbol{A}}_j^\top \overline{\boldsymbol{A}}_j \boldsymbol{\theta} + 2\overline{\boldsymbol{A}}_j^\top \overline{\boldsymbol{a}}_j + \overline{\boldsymbol{c}}_j.
$$

Using the product rule and softmax identity:

$$\nabla^2_{\boldsymbol{\theta}}\ell^{\text{reg}}_\mu(\boldsymbol{\theta}) = \rho \sum_{j\in[J]} w_j(\boldsymbol{\theta})\nabla^2_{\boldsymbol{\theta}}\boldsymbol{\Gamma}_j(\boldsymbol{\theta}) + \frac{\rho 2}{\mu}\sum_{j\in[J]} w_j(\boldsymbol{\theta})\left[\nabla_{\boldsymbol{\theta}}\boldsymbol{\Gamma}_j(\boldsymbol{\theta}) - \bar{g}\right]\left[\nabla_{\boldsymbol{\theta}}\boldsymbol{\Gamma}_j(\boldsymbol{\theta}) - \bar{g}\right]^\top$$

$$= 2\rho \sum_{j\in[J]} w_j(\boldsymbol{\theta})\overline{\boldsymbol{A}}_j^\top\overline{\boldsymbol{A}}_j + \frac{\rho^2}{\mu}\sum_{j\in[J]} w_j(\boldsymbol{\theta})\left[\nabla_{\boldsymbol{\theta}}\boldsymbol{\Gamma}_j(\boldsymbol{\theta}) - \bar{g}\right]\left[\nabla_{\boldsymbol{\theta}}\boldsymbol{\Gamma}_j(\boldsymbol{\theta}) - \bar{g}\right]^\top,$$

where

$$\bar{g} = \rho \sum_{j\in[J]} w_j(\boldsymbol{\theta})\nabla_{\boldsymbol{\theta}}\boldsymbol{\Gamma}_j(\boldsymbol{\theta}).$$

Notice that the first term is a convex combination of positive definite matrices $\overline{\boldsymbol{A}}_j^\top\overline{\boldsymbol{A}}_j$, so it is positive definite. The second term is a Gram matrix, hence positive semidefinite. Therefore, the minimum eigenvalue of the Hessian is lower bounded by the minimum eigenvalue of the first term and we have

$$\nabla^2_{\boldsymbol{\theta}}\ell^{\text{reg}}_\mu(\boldsymbol{\theta}) \succeq 2\rho \min_{j\in[J]} \lambda_{\min}(\overline{\boldsymbol{A}}_j^\top\overline{\boldsymbol{A}}_j)\mathbb{I}.$$

Thus, the claim follows. $\qquad\square$

**Corollary E.1.** *Consider the setting of Model 2, and define the following smoothed loss function*

$$g(\boldsymbol{Z}, \boldsymbol{\theta}) = \ell_\mu(\boldsymbol{Z}, \boldsymbol{\theta}) + \ell^{reg}_\mu(\boldsymbol{\theta})$$

*where $\ell_\mu(\boldsymbol{Z}, \boldsymbol{\theta})$ is the smoothed loss function defined in (15), and $\ell^{reg}_\mu(\boldsymbol{\theta})$ is defined in (17). Suppose that $\overline{\boldsymbol{A}}_j^\top\overline{\boldsymbol{A}}_j \succ 0$ for all $j \in [J]$. Then $g(\boldsymbol{Z}, \boldsymbol{\theta})$ is $\rho\alpha$-strongly convex in $\boldsymbol{\theta}$ with $\alpha = 2\min_{j\in[J]} \lambda_{\min}(\overline{\boldsymbol{A}}_j^\top\overline{\boldsymbol{A}}_j)$, and the gradient $\nabla_{\boldsymbol{\theta}}g(\boldsymbol{Z}, \boldsymbol{\theta})$ is $\beta$-Lipschitz in $\boldsymbol{Z}$ for $\beta = Jdk_3\left(\frac{k_1k_2}{\mu} + 1\right)$ for some constants $k_1, k_2, k_3 < \infty$.*

*Proof.* By a standard result in convex analysis, the function $\ell_\mu(\boldsymbol{Z}, \boldsymbol{\theta})$ is convex in $\boldsymbol{\theta}$, as the log-sum-exp operator preserves convexity when applied to a collection of convex functions. Furthermore, Lemma E.2 establishes that $\ell^{reg}_\mu(\boldsymbol{\theta})$ is $\rho\alpha$-strongly convex, where

$$\alpha = \min_j \lambda_{\min}(\overline{\boldsymbol{A}}_j^\top\overline{\boldsymbol{A}}_j).$$

As the sum of a convex function and a strongly convex function is strongly convex, $g(\boldsymbol{Z}, \overline{\boldsymbol{\theta}})$ is $\rho\alpha$-strongly convex.

The dependence of $g$ on $\boldsymbol{Z}$ comes only through $\ell_\mu(\boldsymbol{Z}, \boldsymbol{\theta})$. Therefore, the Lipschitz continuity of the gradient $\nabla_{\boldsymbol{\theta}}g(\boldsymbol{Z}, \boldsymbol{\theta})$ with respect to $\boldsymbol{Z}$ follows from Lemma 2, which provides the bound on the Lipschitz constant $\beta$. This concludes the proof. $\qquad\square$

**Lemma E.3.** *Assume that the loss function $\ell(\boldsymbol{z}, \boldsymbol{\theta})$ is concave in $\boldsymbol{z}$, and that the set $\mathcal{Z}$ has a finite diameter $D = \sup_{\boldsymbol{z},\boldsymbol{z}'\in\mathcal{Z}} \|\boldsymbol{z} - \boldsymbol{z}'\|_2 < \infty$. Define the function*

$$g(\boldsymbol{v}, (\boldsymbol{\theta}, \lambda)) = \sup_{\boldsymbol{z}\in\mathcal{Z}} \ell(\boldsymbol{z}, \boldsymbol{\theta}) - \lambda\|\boldsymbol{z} - \boldsymbol{v}\|_2^2 + \rho^2\lambda + \tau\lambda^2.$$

*Then the function $g$ is $\alpha$-strongly convex in $(\boldsymbol{\theta}, \lambda)$ where*

$$\alpha = \min(\gamma, 2\tau),$$

*and the gradient $\nabla_{(\boldsymbol{\theta},\lambda)}g(\boldsymbol{v}, (\boldsymbol{\theta}, \lambda))$ is $(\beta + 4D)$-Lipschitz in $\boldsymbol{v}$.*

*Proof.* Define

$$\phi((\boldsymbol{\theta}, \lambda), \boldsymbol{v}, \boldsymbol{z}) = \ell(\boldsymbol{z}, \boldsymbol{\theta}) - \lambda\|\boldsymbol{z} - \boldsymbol{v}\|_2^2 + \rho^2\lambda + \tau\lambda^{\cdot}2$$

Then we have

$$g(\boldsymbol{v}, (\boldsymbol{\theta}, \lambda)) = \sup_{\boldsymbol{z}\in\mathcal{Z}} \phi((\boldsymbol{\theta}, \lambda), \boldsymbol{v}, \boldsymbol{z}).$$

Since $\phi((\boldsymbol{\theta}, \lambda), \boldsymbol{v}, \cdot)$ is $2\lambda$-strongly concave in $\boldsymbol{v}$, the maximizer
$$\boldsymbol{z}^*((\boldsymbol{\theta}, \lambda), \boldsymbol{v}) = \arg\max_{\boldsymbol{z} \in \mathcal{Z}} \phi((\boldsymbol{\theta}, \lambda), \boldsymbol{v}, \boldsymbol{z})$$
is unique. By Danskin's theorem [6, Section 6.11], the function $g$ is differentiable, and its gradient with respect to $(\boldsymbol{\theta}, \lambda)$ is
$$\nabla_{(\boldsymbol{\theta}, \lambda)} g(\boldsymbol{v}, (\boldsymbol{\theta}, \lambda)) = \nabla_{(\boldsymbol{\theta}, \lambda)} \phi((\boldsymbol{\theta}, \lambda), \boldsymbol{v}, \boldsymbol{z}^*)$$
$$= \begin{bmatrix} \nabla_{\boldsymbol{\theta}} \ell(\boldsymbol{z}^*, \boldsymbol{\theta}) \\ -\|\boldsymbol{z}^* - \boldsymbol{v}\|_2^2 + \rho^2 + 2\tau\lambda \end{bmatrix},$$
where $\boldsymbol{z}^* = \boldsymbol{z}^*((\boldsymbol{\theta}, \lambda), \boldsymbol{v})$. The Hessian is
$$\begin{aligned}
\nabla_{(\boldsymbol{\theta}, \lambda)}^2 g(\boldsymbol{v}, (\boldsymbol{\theta}, \lambda)) &= \nabla_{(\boldsymbol{\theta}, \lambda)}^2 \phi((\boldsymbol{\theta}, \lambda), \boldsymbol{v}, \boldsymbol{z}^*) \\
&= \begin{bmatrix} \nabla_{\boldsymbol{\theta}}^2 \ell(\boldsymbol{z}^*, \boldsymbol{\theta}) & 0 \\ 0 & 2\tau \end{bmatrix} \succeq \alpha \mathbb{I}.
\end{aligned} \tag{18}$$
where $\alpha = \min(\gamma, 2\tau)$. This proves strong convexity in $(\boldsymbol{\theta}, \lambda)$.

Before we move to the Lipschitz smoothness, we first prove several inequalities. Fix $\boldsymbol{z}_1, \boldsymbol{z}_2 \in \mathcal{Z}$, we have
$$\begin{aligned}
&\left\| \|\boldsymbol{z}^*((\boldsymbol{\theta}, \lambda), \boldsymbol{v}_2) - \boldsymbol{v}_2\|_2^2 - \|\boldsymbol{z}^*((\boldsymbol{\theta}, \lambda), \boldsymbol{v}_1) - \boldsymbol{v}_1\|_2^2 \right\|_2 \\
&\overset{(a)}{=} | (\|\boldsymbol{z}^*((\boldsymbol{\theta}, \lambda), \boldsymbol{v}_2) - \boldsymbol{v}_2\|_2 + \|\boldsymbol{z}^*((\boldsymbol{\theta}, \lambda), \boldsymbol{v}_1) - \boldsymbol{v}_1\|_2) \\
&\quad \times (\|\boldsymbol{z}^*((\boldsymbol{\theta}, \lambda), \boldsymbol{v}_2) - \boldsymbol{v}_2\|_2 - \|\boldsymbol{z}^*((\boldsymbol{\theta}, \lambda), \boldsymbol{v}_1) - \boldsymbol{v}_1\|_2) | \\
&\overset{(b)}{\leq} 2D\|\boldsymbol{z}^*((\boldsymbol{\theta}, \lambda), \boldsymbol{v}_2) - \boldsymbol{z}^*((\boldsymbol{\theta}, \lambda), \boldsymbol{v}_1)\|_2 + 2D\|\boldsymbol{v}_2 - \boldsymbol{v}_1\|_2
\end{aligned} \tag{19}$$
where (a) uses the difference of squares, and (b) uses the triangular inequality and the boundness of $\mathcal{Z}$. Next we prove $\boldsymbol{z}^*((\boldsymbol{\theta}, \lambda), \boldsymbol{v})$ is 1-Lipschitz continuous in $\boldsymbol{v}$ as follows.
$$\begin{aligned}
&2\lambda \|\boldsymbol{z}^*((\boldsymbol{\theta}, \lambda), \boldsymbol{v}_1) - \boldsymbol{z}^*((\boldsymbol{\theta}, \lambda), \boldsymbol{v}_2)\|_2^2 \\
&\overset{(a)}{\leq} \langle -\nabla_{\boldsymbol{z}} \phi((\boldsymbol{\theta}, \lambda), \boldsymbol{v}_2, \boldsymbol{z}^*((\boldsymbol{\theta}, \lambda), \boldsymbol{v}_2)) + \nabla_{\boldsymbol{z}} \phi((\boldsymbol{\theta}, \lambda), \boldsymbol{v}_2, \boldsymbol{z}^*((\boldsymbol{\theta}, \lambda), \boldsymbol{v}_1)), \\
&\qquad \boldsymbol{z}^*((\boldsymbol{\theta}, \lambda), \boldsymbol{v}_2) - \boldsymbol{z}^*((\boldsymbol{\theta}, \lambda), \boldsymbol{v}_1) \rangle \\
&\overset{(b)}{\leq} \langle \nabla_{\boldsymbol{z}} \phi((\boldsymbol{\theta}, \lambda), \boldsymbol{v}_2, \boldsymbol{z}^*((\boldsymbol{\theta}, \lambda), \boldsymbol{v}_1)), \boldsymbol{z}^*((\boldsymbol{\theta}, \lambda), \boldsymbol{v}_2) - \boldsymbol{z}^*((\boldsymbol{\theta}, \lambda), \boldsymbol{v}_1) \rangle \\
&\overset{(c)}{\leq} \langle \nabla_{\boldsymbol{z}} \phi((\boldsymbol{\theta}, \lambda), \boldsymbol{v}_2, \boldsymbol{z}^*((\boldsymbol{\theta}, \lambda), \boldsymbol{v}_1)) - \nabla_{\boldsymbol{z}} \phi((\boldsymbol{\theta}, \lambda), \boldsymbol{v}_1, \boldsymbol{z}^*((\boldsymbol{\theta}, \lambda), \boldsymbol{v}_1)), \\
&\qquad \boldsymbol{z}^*((\boldsymbol{\theta}, \lambda), \boldsymbol{v}_2) - \boldsymbol{z}^*((\boldsymbol{\theta}, \lambda), \boldsymbol{v}_1) \rangle \\
&\overset{(d)}{\leq} 2\lambda \|\boldsymbol{v}_1 - \boldsymbol{v}_2\| \|\boldsymbol{z}^*((\boldsymbol{\theta}, \lambda), \boldsymbol{v}_2) - \boldsymbol{z}^*((\boldsymbol{\theta}, \lambda), \boldsymbol{v}_1)\|
\end{aligned}$$
where (a) uses strong concavity of $\phi((\boldsymbol{\theta}, \lambda), \boldsymbol{v}, \cdot)$, (b) and (c) come from the first order optimality conditions for $\boldsymbol{z}^*((\boldsymbol{\theta}, \lambda), \boldsymbol{v}_1))$ and $\boldsymbol{z}^*((\boldsymbol{\theta}, \lambda), \boldsymbol{v}_2))$:
$$\langle \nabla_{\boldsymbol{z}} \phi((\boldsymbol{\theta}, \lambda), \boldsymbol{v}, \boldsymbol{z}^*((\boldsymbol{\theta}, \lambda), \boldsymbol{v})), \boldsymbol{z} - \boldsymbol{z}^*((\boldsymbol{\theta}, \lambda), \boldsymbol{v}) \rangle \leq 0,$$
and (d) uses Cauchy-Schwarz inequality. Hence,
$$\|\boldsymbol{z}^*((\boldsymbol{\theta}, \lambda), \boldsymbol{v}_1) - \boldsymbol{z}^*((\boldsymbol{\theta}, \lambda), \boldsymbol{v}_2)\|_2 \leq \|\boldsymbol{v}_1 - \boldsymbol{v}_2\| \tag{20}$$
Finally we show $\nabla_{(\boldsymbol{\theta}, \lambda)} g(\boldsymbol{v}, (\boldsymbol{\theta}, \lambda))$ is Lipschitz in $\boldsymbol{v}$, consider the gradient difference
$$\begin{aligned}
&\|\nabla_{(\boldsymbol{\theta}, \lambda)} g(\boldsymbol{v}_1, (\boldsymbol{\theta}, \lambda)) - \nabla_{(\boldsymbol{\theta}, \lambda)} g(\boldsymbol{v}_2, (\boldsymbol{\theta}, \lambda))\|_2 \\
&= \left\| \begin{bmatrix} \nabla_{\boldsymbol{\theta}} \ell(\boldsymbol{z}^*((\boldsymbol{\theta}, \lambda), \boldsymbol{v}_1), \boldsymbol{\theta}) - \nabla_{\boldsymbol{\theta}} \ell(\boldsymbol{z}^*((\boldsymbol{\theta}, \lambda), \boldsymbol{v}_2), \boldsymbol{\theta}) \\ \|\boldsymbol{z}^*((\boldsymbol{\theta}, \lambda), \boldsymbol{v}_2) - \boldsymbol{v}_2\|_2^2 - \|\boldsymbol{z}^*((\boldsymbol{\theta}, \lambda), \boldsymbol{v}_1) - \boldsymbol{v}_1\|_2^2 \end{bmatrix} \right\|_2 \\
&\overset{(a)}{\leq} \|\nabla_{\boldsymbol{\theta}} \ell(\boldsymbol{z}^*((\boldsymbol{\theta}, \lambda), \boldsymbol{v}_1), \boldsymbol{\theta}) - \nabla_{\boldsymbol{\theta}} \ell(\boldsymbol{z}^*((\boldsymbol{\theta}, \lambda), \boldsymbol{v}_2), \boldsymbol{\theta})\|_2 \\
&\quad + \left\| \|\boldsymbol{z}^*((\boldsymbol{\theta}, \lambda), \boldsymbol{v}_2) - \boldsymbol{v}_2\|_2^2 - \|\boldsymbol{z}^*((\boldsymbol{\theta}, \lambda), \boldsymbol{v}_1) - \boldsymbol{v}_1\|_2^2 \right\|_2 \\
&\overset{(b)}{\leq} \beta \|\boldsymbol{z}^*((\boldsymbol{\theta}, \lambda), \boldsymbol{v}_1) - \boldsymbol{z}^*((\boldsymbol{\theta}, \lambda), \boldsymbol{v}_2)\|_2 + \\
&\quad 2D\|\boldsymbol{z}^*((\boldsymbol{\theta}, \lambda), \boldsymbol{v}_2) - \boldsymbol{z}^*((\boldsymbol{\theta}, \lambda), \boldsymbol{v}_1)\|_2 + 2D\|\boldsymbol{v}_2 - \boldsymbol{v}_1\|_2 \\
&\overset{(c)}{\leq} (\beta + 4D)\|\boldsymbol{v}_2 - \boldsymbol{v}_1\|_2
\end{aligned}$$
where (a) comes from the triangular inequality, (b) uses the $\beta$-jointly smoothness of $\ell(\boldsymbol{z}, \boldsymbol{\theta})$ and 19, and (c) uses 20. Hence the gradient $\nabla_{\boldsymbol{\theta}, \lambda} g(\boldsymbol{v}, (\boldsymbol{\theta}, \lambda))$ is $(\beta + 4D)$-Lipschitz in $\boldsymbol{v}$. $\qquad \square$

## E.1 Proof of Theorem 1

*Proof.* Let $G(\boldsymbol{\theta}_t)$ denote an optimal solution of (RRMP) at iteration $t$, i.e.,

$$\boldsymbol{\theta}_{t+1} = G(\boldsymbol{\theta}_t) \in \arg\min_{\boldsymbol{\theta} \in \boldsymbol{\Theta}} \mathbb{J}_{\boldsymbol{\theta}_t}(\boldsymbol{\theta}).$$

where $\boldsymbol{\theta}_t$ is the current solution and $\boldsymbol{\theta}_{t+1} \in \boldsymbol{\Theta}$ denotes the optimal solution for next iteration.

We first prove the convergence result for Model 1. Observe that

$$\mathbb{J}_{\boldsymbol{\theta}_t}(\boldsymbol{\theta}) = \mathbb{E}_{\hat{\mathbb{P}}(\boldsymbol{\theta}_t)}[g(\boldsymbol{Z}, \boldsymbol{\theta})]$$

where $g(\boldsymbol{Z}, \overline{\boldsymbol{\theta}}) = \ell(\boldsymbol{Z}, \boldsymbol{\theta}) + \rho L \|(\boldsymbol{\theta}, 1)\|_2^2$.

Fix $\boldsymbol{\eta}, \boldsymbol{\eta}' \in \boldsymbol{\Theta}$. Since $\mathbb{J}_{\boldsymbol{\eta}}(\cdot)$ is $(\gamma + 2\rho L)$-strongly convex, where $2\rho L$ comes from the strong convexity of the regularization term $\rho L \|(\boldsymbol{\theta}, 1)\|_2^2$. we have

$$\mathbb{J}_{\boldsymbol{\eta}}(G(\boldsymbol{\eta})) - \mathbb{J}_{\boldsymbol{\eta}}(G(\boldsymbol{\eta}')) \geq (G(\boldsymbol{\eta}) - G(\boldsymbol{\eta}'))^\top \nabla \mathbb{J}_{\boldsymbol{\eta}}(G(\boldsymbol{\eta}')) + \frac{\gamma + 2\rho L}{2} \|G(\boldsymbol{\eta}) - G(\boldsymbol{\eta}')\|_2^2,$$

$$\mathbb{J}_{\boldsymbol{\eta}}(G(\boldsymbol{\eta}')) - \mathbb{J}_{\boldsymbol{\eta}}(G(\boldsymbol{\eta})) \geq \frac{\gamma + 2\rho L}{2} \|G(\boldsymbol{\eta}) - G(\boldsymbol{\eta}')\|_2^2,$$

where the second inequality follows from the fact that

$$(G(\boldsymbol{\eta}') - G(\boldsymbol{\eta}))^\top \nabla \mathbb{J}_{\boldsymbol{\eta}}(G(\boldsymbol{\eta})) \geq 0$$

in view of the first-order optimality condition in Lemma C.1 since $G(\boldsymbol{\eta}) \in \arg\min_{\overline{\boldsymbol{\theta}} \in \overline{\boldsymbol{\Theta}}} \mathbb{J}_{\boldsymbol{\eta}}(\overline{\boldsymbol{\theta}})$. Combining the two inequalities, we obtain

$$(\gamma + 2\rho L)\|G(\boldsymbol{\eta}) - G(\boldsymbol{\eta}')\|_2^2 \quad \begin{aligned} &\leq -(G(\boldsymbol{\eta}) - G(\boldsymbol{\eta}'))^\top \nabla \mathbb{J}_{\boldsymbol{\eta}}(G(\boldsymbol{\eta}')) \\ &\leq (G(\boldsymbol{\eta}) - G(\boldsymbol{\eta}'))^\top [\nabla \mathbb{J}_{\boldsymbol{\eta}'}(G(\boldsymbol{\eta}')) - \nabla \mathbb{J}_{\boldsymbol{\eta}}(G(\boldsymbol{\eta}'))], \end{aligned} \quad (21)$$

where the second inequality follows from the fact that $(G(\boldsymbol{\eta}) - G(\boldsymbol{\eta}'))^\top \nabla \mathbb{J}_{\boldsymbol{\eta}'}(G(\boldsymbol{\eta}')) \geq 0$ in view of the first-order optimality condition of $G(\boldsymbol{\eta}')$. Next, we will upper bound (21) using Cauchy-Schwarz inequality, as follows:

$$(G(\boldsymbol{\eta}) - G(\boldsymbol{\eta}'))^\top [\nabla \mathbb{J}_{\boldsymbol{\eta}'}(G(\boldsymbol{\eta}')) - \nabla \mathbb{J}_{\boldsymbol{\eta}}(G(\boldsymbol{\eta}'))]$$

$$\leq \|(G(\boldsymbol{\eta}) - G(\boldsymbol{\eta}'))\|_2 \|\nabla \mathbb{J}_{\boldsymbol{\eta}'}(G(\boldsymbol{\eta}')) - \nabla \mathbb{J}_{\boldsymbol{\eta}}(G(\boldsymbol{\eta}'))\|_2$$

$$\overset{(a)}{=} \|(G(\boldsymbol{\eta}) - G(\boldsymbol{\eta}'))\|_2 \left\| \mathbb{E}_{\hat{\mathbb{P}}(\boldsymbol{\eta}')}[\nabla g(\tilde{\boldsymbol{Z}}; G(\boldsymbol{\eta}'))] - \mathbb{E}_{\hat{\mathbb{P}}(\boldsymbol{\eta})}[\nabla g(\tilde{\boldsymbol{Z}}; G(\boldsymbol{\eta}'))] \right\|_2$$

$$\overset{(b)}{\leq} \|G(\boldsymbol{\eta}) - G(\boldsymbol{\eta}')\|_2 \cdot \epsilon\beta \|\boldsymbol{\eta} - \boldsymbol{\eta}'\|_2.$$

Here, (a) follows from the representation of the loss function, while (b) uses the Kantorovich-Rubinstein Lemma C.2 since the loss function is $\beta$-jointly smooth from Lemma E.3 and the map $\hat{\mathbb{P}}(\boldsymbol{\theta})$ is $\epsilon$-sensitive. Combining this bound with (21), we get

$$\|G(\boldsymbol{\eta}) - G(\boldsymbol{\eta}')\|_2 \leq \frac{\epsilon\beta}{\gamma + 2\rho L} \|\boldsymbol{\eta} - \boldsymbol{\eta}'\|_2.$$

Our claim (a) is then established by simply performing the change of variables $\boldsymbol{\eta} \leftarrow \boldsymbol{\theta}$ and $\boldsymbol{\eta}' \leftarrow \boldsymbol{\theta}'$.

To prove claim (b), we observe that $\boldsymbol{\theta}_t = G(\boldsymbol{\theta}_{t-1})$ by the definition of (RRMP), and $\boldsymbol{\theta}_{\text{RPS}} = G(\boldsymbol{\theta}_{\text{RPS}})$ by the definition of stability. Applying the result of the claim (a) yields

$$\|\boldsymbol{\theta}_t - \boldsymbol{\theta}_{\text{RPS}}\|_2 \leq \frac{\epsilon\beta}{\gamma + 2\rho L} \|\boldsymbol{\theta}_{t-1} - \boldsymbol{\theta}_{\text{RPS}}\|_2 \leq \left( \frac{\epsilon\beta}{\gamma + 2\rho L} \right)^t \|\boldsymbol{\theta}_0 - \boldsymbol{\theta}_{\text{RPS}}\|_2.$$

Setting the right-hand side expression to be at most $\delta$ and solving for $t$ completes the proof for Model 1.

For Model 2. Observe that

$$\mathbb{J}_{\boldsymbol{\theta}_t}^\mu(\boldsymbol{\theta}) = \mathbb{E}_{\hat{\mathbb{P}}(\boldsymbol{\theta}_t)}[g(\boldsymbol{Z}, \boldsymbol{\theta})],$$

where $g(\boldsymbol{Z}, \boldsymbol{\theta}) = \ell_\mu(\boldsymbol{Z}, \boldsymbol{\theta}) + \ell_\mu^{reg}(\boldsymbol{\theta})$. Here $\ell_\mu(\boldsymbol{Z}, \boldsymbol{\theta})$ is the smoothed loss function defined in (15), and $\ell_\mu^{reg}(\boldsymbol{\theta})$ is defined in (17). From Corollary E.1, we know that $g(\boldsymbol{Z}, \boldsymbol{\theta})$ is $\rho\alpha$-strongly convex

in $\boldsymbol{\theta}$ with $\alpha = 2\min_{j\in[J]}\lambda_{\min}(\overline{\boldsymbol{A}}_j^\top\overline{\boldsymbol{A}}_j)$, and the gradient $\nabla_{\boldsymbol{\theta}}g(\boldsymbol{Z},\boldsymbol{\theta})$ is $\beta$-Lipschitz in $\boldsymbol{Z}$ for $\beta = Jdk_3\left(\frac{k_1k_2}{\mu}+1\right)$ for some constants $k_1,k_2,k_3 < \infty$. Using the same techniques as in the proof of Model 1, we can therefore establish the desired result for Model 2.

Finally, for Model 3, one can observe that

$$\mathbb{J}_{\boldsymbol{\theta}_t}^\tau(\overline{\boldsymbol{\theta}}) = \mathbb{E}_{\hat{\mathbb{P}}(\boldsymbol{\theta}_t)}[g(\boldsymbol{Z},\overline{\boldsymbol{\theta}})],$$

where

$$g(\boldsymbol{Z},\overline{\boldsymbol{\theta}}) = \sup_{\boldsymbol{z}\in\mathcal{Z}} \ell(\boldsymbol{z},\boldsymbol{\theta}) - \lambda\|\boldsymbol{z}-\boldsymbol{Z}\|_2^2 + \rho^2\lambda + \tau\lambda^2.$$

From Lemma E.3, we know that $g(\boldsymbol{Z},\overline{\boldsymbol{\theta}})$ is $\alpha$-strongly convex in $\overline{\boldsymbol{\theta}}$ with $\alpha = \min(\gamma, 2\tau)$, and that its gradient $\nabla_{\overline{\boldsymbol{\theta}}}g(\boldsymbol{Z},\overline{\boldsymbol{\theta}})$ is $(\beta+4D)$-Lipschitz continuous in $\boldsymbol{Z}$. Hence, by applying the same arguments as in the preceding analysis, we obtain the desired result.

$\square$

# F  Deferred Proofs Related to Sub-optimality Guarantees

## F.1  Proof of Sub-optimality Guarantee for Model 1

**Theorem 3.** *Suppose the loss function in Model 1 is $\gamma$-strongly convex in $\boldsymbol{\theta}$ (B1) and is $\beta$-smooth (B2). Furthermore, assume that the loss function $\ell(\boldsymbol{z},\boldsymbol{\theta})$ is $L_z$-Lipschitz in $\boldsymbol{z}$. Then, the following suboptimality bound holds:*

$$\mathbb{J}_{\boldsymbol{\theta}_{\mathrm{RPS}}}(\boldsymbol{\theta}_{\mathrm{RPS}}) - \mathbb{J}_{\boldsymbol{\theta}_{\mathrm{RPO}}}(\boldsymbol{\theta}_{\mathrm{RPO}}) \leq \frac{2\epsilon^2 L_z^2}{\gamma + 2\rho L}.$$

*Proof.* Since $\mathbb{J}_{\boldsymbol{\theta}_{\mathrm{RPS}}}(\theta)$ is $(\gamma + 2\rho L)$-strongly convex in $\theta$, we have

$$\frac{\gamma + 2\rho L}{2}\|\boldsymbol{\theta}_{\mathrm{RPO}} - \boldsymbol{\theta}_{\mathrm{RPS}}\|_2^2 \leq \mathbb{J}_{\boldsymbol{\theta}_{\mathrm{RPS}}}(\boldsymbol{\theta}_{\mathrm{RPO}}) - \mathbb{J}_{\boldsymbol{\theta}_{\mathrm{RPS}}}(\boldsymbol{\theta}_{\mathrm{RPS}})$$

since $(\boldsymbol{\theta}_{\mathrm{RPO}} - \boldsymbol{\theta}_{\mathrm{RPS}})^\top\nabla\mathbb{J}_{\boldsymbol{\theta}_{\mathrm{RPS}}}(\boldsymbol{\theta}_{\mathrm{RPS}}) \geq 0$ by the optimality of $\boldsymbol{\theta}_{\mathrm{RPS}}$. Using the fact that $\mathbb{J}_{\boldsymbol{\theta}_{\mathrm{RPS}}}(\boldsymbol{\theta}_{\mathrm{RPS}}) \geq \mathbb{J}_{\boldsymbol{\theta}_{\mathrm{RPO}}}(\boldsymbol{\theta}_{\mathrm{RPO}})$, we can further upper bound the right-hand side

$$\begin{aligned}\mathbb{J}_{\boldsymbol{\theta}_{\mathrm{RPS}}}(\boldsymbol{\theta}_{\mathrm{RPO}}) - \mathbb{J}_{\boldsymbol{\theta}_{\mathrm{RPS}}}(\boldsymbol{\theta}_{\mathrm{RPS}}) \quad &\leq \mathbb{J}_{\boldsymbol{\theta}_{\mathrm{RPS}}}(\boldsymbol{\theta}_{\mathrm{RPO}}) - \mathbb{J}_{\boldsymbol{\theta}_{\mathrm{RPO}}}(\boldsymbol{\theta}_{\mathrm{RPO}}) \\ &\leq \epsilon L_z\|\boldsymbol{\theta}_{\mathrm{RPS}} - \boldsymbol{\theta}_{\mathrm{RPO}}\|_2,\end{aligned} \tag{22}$$

where the second inequality holds due the $\epsilon$-sensitivity of the distribution map $\hat{\mathbb{P}}(\cdot)$ and the $L_z$-Lipschitz continuity of the loss function in $\boldsymbol{z}$. In summary, we obtain

$$\|\boldsymbol{\theta}_{\mathrm{RPS}} - \boldsymbol{\theta}_{\mathrm{RPO}}\|_2 \leq \frac{2\epsilon L_z}{\gamma + 2\rho L}. \tag{23}$$

Next, we derive a bound on the suboptimality of the robust performatively stable solution $\boldsymbol{\theta}_{\mathrm{RPS}}$. We have

$$\begin{aligned}\mathbb{J}_{\boldsymbol{\theta}_{\mathrm{RPS}}}(\boldsymbol{\theta}_{\mathrm{RPS}}) - \mathbb{J}_{\boldsymbol{\theta}_{\mathrm{RPO}}}(\boldsymbol{\theta}_{\mathrm{RPO}}) &\leq \mathbb{J}_{\boldsymbol{\theta}_{\mathrm{RPS}}}(\boldsymbol{\theta}_{\mathrm{RPO}}) - \mathbb{J}_{\boldsymbol{\theta}_{\mathrm{RPO}}}(\boldsymbol{\theta}_{\mathrm{RPO}}) \\ &\leq \epsilon L_z\|\boldsymbol{\theta}_{\mathrm{RPS}} - \boldsymbol{\theta}_{\mathrm{RPO}}\|_2 \\ &\leq \frac{2\epsilon^2 L_z^2}{\gamma + 2\rho L},\end{aligned}$$

where the first inequality follows from the suboptimality of $\boldsymbol{\theta}_{\mathrm{RPO}}$ in $\mathbb{J}_{\boldsymbol{\theta}_{\mathrm{RPS}}}(\theta)$, the second inequality is from (22), and the last inequality is from (23). This completes the proof. $\square$

## F.2  Proof of Sub-optimality Guarantee for Model 2

**Theorem 4.** *Consider the loss function $\ell(\boldsymbol{Z},\boldsymbol{\theta})$ from Model 2. Suppose that $\overline{\boldsymbol{A}}_j^\top\overline{\boldsymbol{A}}_j \succ \boldsymbol{0}$ for all $j \in [J]$, and that $\boldsymbol{Y}_s \succeq \boldsymbol{0}$ for all $s \in [S]$. Additionally, assume that $\ell(\boldsymbol{Z},\boldsymbol{\theta})$ is $L_z$-Lipschitz in $\boldsymbol{Z}$.*

*Then, the suboptimality of the stable point $\boldsymbol{\theta}_{\mathrm{RPS}}^{\mu}$ of the smoothed robust objective satisfies:*

$$\mathbb{J}_{\boldsymbol{\theta}_{\mathrm{RPS}}^{\mu}}^{\mu}(\boldsymbol{\theta}_{\mathrm{RPS}}^{\mu}) - \mathbb{J}_{\boldsymbol{\theta}_{\mathrm{RPO}}}(\boldsymbol{\theta}_{\mathrm{RPO}}) \leq \frac{2(\epsilon L_z + 2\mu' \log J)^2}{\rho\alpha}. \tag{24}$$

*where $\alpha = 2\min_{j\in[J]} \lambda_{min}(\overline{\boldsymbol{A}}_j^{\top}\overline{\boldsymbol{A}}_j)$, $\mu' \in [0,1]$ is a constant that satisfies $\mu \leq \mu'\|\boldsymbol{\theta}_{\mathrm{RPS}}^{\mu} - \boldsymbol{\theta}_{\mathrm{RPO}}\|_2$, $\mathbb{J}_{\boldsymbol{\theta}_t}^{\mu}(\boldsymbol{\theta})$ is the smoothed robust objective defined in (12), and $\boldsymbol{\theta}_{\mathrm{RPO}}$ denotes the minimizer of the original robust objective $\mathbb{J}_{\boldsymbol{\theta}_t}(\boldsymbol{\theta})$.*

*Proof.* By Lemma C.4, we have for any $\boldsymbol{\theta}$,

$$\mathbb{J}_{\boldsymbol{\theta}_t}(\boldsymbol{\theta}) \leq \mathbb{J}_{\boldsymbol{\theta}_t}^{\mu}(\boldsymbol{\theta}) \leq \mathbb{J}_{\boldsymbol{\theta}_t}(\boldsymbol{\theta}) + 2\mu \log J. \tag{25}$$

Additionally, from Corollary E.1, the function $\mathbb{J}_{\boldsymbol{\theta}_t}^{\mu}(\boldsymbol{\theta})$ is $\rho\alpha$-strongly convex, with $\alpha = 2\min_{j\in[J]} \lambda_{min}(\overline{\boldsymbol{A}}_j^{\top}\overline{\boldsymbol{A}}_j)$. By the definition of strong convexity, we have:

$$\frac{\rho\alpha}{2}\|\boldsymbol{\theta}_{\mathrm{RPO}} - \boldsymbol{\theta}_{\mathrm{RPS}}^{\mu}\|_2^2 \leq \mathbb{J}_{\boldsymbol{\theta}_{\mathrm{RPS}}^{\mu}}^{\mu}(\boldsymbol{\theta}_{\mathrm{RPO}}) - \mathbb{J}_{\boldsymbol{\theta}_{\mathrm{RPS}}^{\mu}}^{\mu}(\boldsymbol{\theta}_{\mathrm{RPS}}^{\mu}), \tag{26}$$

where the inequality follows from the first-order optimality condition of $\boldsymbol{\theta}_{\mathrm{RPS}}^{\mu}$, i.e.,

$$(\boldsymbol{\theta}_{\mathrm{RPO}} - \boldsymbol{\theta}_{\mathrm{RPS}}^{\mu})^{\top}\nabla\mathbb{J}_{\boldsymbol{\theta}_{\mathrm{RPS}}^{\mu}}^{\mu}(\boldsymbol{\theta}_{\mathrm{RPS}}^{\mu}) \geq 0.$$

We next bound the right-hand side of (26). Using the fact that $\mathbb{J}_{\boldsymbol{\theta}_{\mathrm{RPO}}}(\boldsymbol{\theta}_{\mathrm{RPO}}) \leq \mathbb{J}_{\boldsymbol{\theta}_{\mathrm{RPS}}^{\mu}}(\boldsymbol{\theta}_{\mathrm{RPS}}^{\mu}) \leq \mathbb{J}_{\boldsymbol{\theta}_{\mathrm{RPS}}^{\mu}}^{\mu}(\boldsymbol{\theta}_{\mathrm{RPS}}^{\mu})$, we have

$$\begin{aligned}
\mathbb{J}_{\boldsymbol{\theta}_{\mathrm{RPS}}^{\mu}}^{\mu}(\boldsymbol{\theta}_{\mathrm{RPO}}) - \mathbb{J}_{\boldsymbol{\theta}_{\mathrm{RPS}}^{\mu}}^{\mu}(\boldsymbol{\theta}_{\mathrm{RPS}}^{\mu}) &\leq \mathbb{J}_{\boldsymbol{\theta}_{\mathrm{RPS}}^{\mu}}^{\mu}(\boldsymbol{\theta}_{\mathrm{RPO}}) - \mathbb{J}_{\boldsymbol{\theta}_{\mathrm{RPO}}}(\boldsymbol{\theta}_{\mathrm{RPO}}) \\
&\leq \mathbb{J}_{\boldsymbol{\theta}_{\mathrm{RPS}}^{\mu}}(\boldsymbol{\theta}_{\mathrm{RPO}}) - \mathbb{J}_{\boldsymbol{\theta}_{\mathrm{RPO}}}(\boldsymbol{\theta}_{\mathrm{RPO}}) + 2\mu\log J \\
&\leq \epsilon L_z\|\boldsymbol{\theta}_{\mathrm{RPS}}^{\mu} - \boldsymbol{\theta}_{\mathrm{RPO}}\|_2 + 2\mu\log J.
\end{aligned} \tag{27}$$

where the second inequality comes from (25), and the last inequality holds due to the Lipschitz continuity of $\ell(\boldsymbol{Z},\boldsymbol{\theta})$ in $\boldsymbol{Z}$. Substituting (27) into the strong convexity inequality (26) gives:

$$\frac{\rho\alpha}{2}\|\boldsymbol{\theta}_{\mathrm{RPO}} - \boldsymbol{\theta}_{\mathrm{RPS}}^{\mu}\|_2^2 \leq \epsilon L_z\|\boldsymbol{\theta}_{\mathrm{RPS}}^{\mu} - \boldsymbol{\theta}_{\mathrm{RPO}}\|_2 + 2\mu\log J. \tag{28}$$

Assuming $\mu \leq \mu'\|\boldsymbol{\theta}_{\mathrm{RPS}}^{\mu} - \boldsymbol{\theta}_{\mathrm{RPO}}\|_2$ where $\mu' \in [0,1]$, we can divide both sides by $\|\boldsymbol{\theta}_{\mathrm{RPS}}^{\mu} - \boldsymbol{\theta}_{\mathrm{RPO}}\|_2$ to obtain:

$$\|\boldsymbol{\theta}_{\mathrm{RPS}}^{\mu} - \boldsymbol{\theta}_{\mathrm{RPO}}\|_2 \leq \frac{2(\epsilon L_z + 2\mu'\log J)}{\rho\alpha}. \tag{29}$$

Finally, we derive a bound on the suboptimality of the robust performatively stable solution $\boldsymbol{\theta}_{\mathrm{RPS}}^{\mu}$:

$$\begin{aligned}
\mathbb{J}_{\boldsymbol{\theta}_{\mathrm{RPS}}^{\mu}}^{\mu}(\boldsymbol{\theta}_{\mathrm{RPS}}^{\mu}) - \mathbb{J}_{\boldsymbol{\theta}_{\mathrm{RPO}}}(\boldsymbol{\theta}_{\mathrm{RPO}}) &\leq \mathbb{J}_{\boldsymbol{\theta}_{\mathrm{RPS}}^{\mu}}^{\mu}(\boldsymbol{\theta}_{\mathrm{RPO}}) - \mathbb{J}_{\boldsymbol{\theta}_{\mathrm{RPO}}}(\boldsymbol{\theta}_{\mathrm{RPO}}) \\
&\leq (\epsilon L_z + 2\mu'\log J)\|\boldsymbol{\theta}_{\mathrm{RPS}}^{\mu} - \boldsymbol{\theta}_{\mathrm{RPO}}\|_2 \\
&\leq \frac{2(\epsilon L_z + 2\mu'\log J)^2}{\rho\alpha},
\end{aligned}$$

where the first inequality uses suboptimality of $\boldsymbol{\theta}_{\mathrm{RPO}}$ in $\mathbb{J}_{\boldsymbol{\theta}_{\mathrm{RPS}}^{\mu}}^{\mu}(\boldsymbol{\theta})$, the second follows from (27), and the last inequality uses the bound in (29). This concludes the proof. $\qquad\square$

### F.3 Proof of Sub-optimality Guarantee for Model 3

**Theorem 5.** *Suppose that $\mathcal{Z}$ has a finite diameter $D = \sup_{\boldsymbol{z},\boldsymbol{z}'\in\mathcal{Z}}\|\boldsymbol{z} - \boldsymbol{z}'\|_2 < \infty$ and $\Theta$ is bounded. Assume that the loss function $\ell(\boldsymbol{z},\boldsymbol{\theta})$ in Theorem (3) is $L_\theta$ Lipschitz continuous in $\boldsymbol{\theta}$ and $L_z$-Lipschitz in $\boldsymbol{z}$. Let $\mathbb{J}_{\boldsymbol{\theta}_t}^{\tau}(\overline{\boldsymbol{\theta}})$ denote the objective defined in problem (7), and let $\overline{\boldsymbol{\theta}}_{\mathrm{RPS}}^{\tau}$ denote a robust performative stable point under this objective. Then the following suboptimality bound holds:*

$$\mathbb{J}_{\boldsymbol{\theta}_{\mathrm{RPS}}^{\tau}}^{\tau}(\overline{\boldsymbol{\theta}}_{\mathrm{RPS}}^{\tau}) - \mathbb{J}_{\boldsymbol{\theta}_{\mathrm{RPO}}}(\overline{\boldsymbol{\theta}}_{\mathrm{RPO}}) \leq \tau\overline{\lambda}^2 + \frac{(2\tau\overline{\lambda} + \rho^2 + D^2 + L_\theta + \epsilon L_z)2\epsilon L_z}{\alpha}. \tag{30}$$

*where $\alpha := \min(\gamma, 2\tau)$ is the strong convexity parameter, and $\overline{\lambda}$ is the upper bound on the optimal $\lambda$ as given in Lemma F.1.*

*Proof.* From Lemma E.3, the objective function $\mathbb{J}^{\tau}_{\boldsymbol{\theta}_t}(\overline{\boldsymbol{\theta}})$ is $\alpha$-strongly convex in $\overline{\boldsymbol{\theta}} = (\boldsymbol{\theta}, \lambda)$ with $\alpha = \min(\gamma, 2\tau)$. Applying the strong convexity inequality, we obtain:

$$\frac{\alpha}{2}\left\|\overline{\boldsymbol{\theta}}^{\tau}_{\text{RPO}} - \overline{\boldsymbol{\theta}}^{\tau}_{\text{RPS}}\right\|_2^2 \leq \mathbb{J}^{\tau}_{\boldsymbol{\theta}^{\tau}_{\text{RPS}}}(\overline{\boldsymbol{\theta}}^{\tau}_{\text{RPO}}) - \mathbb{J}^{\tau}_{\boldsymbol{\theta}^{\tau}_{\text{RPS}}}(\overline{\boldsymbol{\theta}}^{\tau}_{\text{RPS}}),$$

where the inequality follows from $(\overline{\boldsymbol{\theta}}^{\tau}_{\text{RPO}} - \overline{\boldsymbol{\theta}}^{\tau}_{\text{RPS}})^{\top} \nabla \mathbb{J}^{\tau}_{\boldsymbol{\theta}^{\tau}_{\text{RPS}}}(\overline{\boldsymbol{\theta}}^{\tau}_{\text{RPS}}) \geq 0$ by the optimality of $\overline{\boldsymbol{\theta}}^{\tau}_{\text{RPS}}$.
Using the fact that $\mathbb{J}^{\tau}_{\boldsymbol{\theta}^{\tau}_{\text{RPO}}}(\overline{\boldsymbol{\theta}}^{\tau}_{\text{RPO}}) \leq \mathbb{J}^{\tau}_{\boldsymbol{\theta}^{\tau}_{\text{RPS}}}(\overline{\boldsymbol{\theta}}^{\tau}_{\text{RPS}})$, we can further upper bound the right-hand side

$$
\begin{aligned}
\mathbb{J}^{\tau}_{\boldsymbol{\theta}^{\tau}_{\text{RPS}}}(\overline{\boldsymbol{\theta}}^{\tau}_{\text{RPO}}) - \mathbb{J}^{\tau}_{\boldsymbol{\theta}^{\tau}_{\text{RPS}}}(\overline{\boldsymbol{\theta}}^{\tau}_{\text{RPS}}) &\leq \mathbb{J}^{\tau}_{\boldsymbol{\theta}^{\tau}_{\text{RPS}}}(\overline{\boldsymbol{\theta}}^{\tau}_{\text{RPO}}) - \mathbb{J}^{\tau}_{\boldsymbol{\theta}^{\tau}_{\text{RPO}}}(\overline{\boldsymbol{\theta}}^{\tau}_{\text{RPO}}) \\
&\leq \epsilon L_z \|\boldsymbol{\theta}^{\tau}_{\text{RPS}} - \boldsymbol{\theta}^{\tau}_{\text{RPO}}\|_2 \\
&\leq \epsilon L_z \|\overline{\boldsymbol{\theta}}^{\tau}_{\text{RPS}} - \overline{\boldsymbol{\theta}}^{\tau}_{\text{RPO}}\|_2,
\end{aligned}
\tag{31}
$$

where the second inequality holds due the $\epsilon$-sensitivity of the distribution map $\hat{\mathbb{P}}(\cdot)$ and the $L_z$-Lipschitz continuity of the loss function in $\boldsymbol{z}$. In summary, we obtain

$$\|\overline{\boldsymbol{\theta}}^{\tau}_{\text{RPS}} - \overline{\boldsymbol{\theta}}^{\tau}_{\text{RPO}}\|_2 \leq \frac{2\epsilon L_z}{\alpha}. \tag{32}$$

Now consider the suboptimality decomposition:

$$
\begin{aligned}
&\mathbb{J}^{\tau}_{\boldsymbol{\theta}^{\tau}_{\text{RPS}}}(\overline{\boldsymbol{\theta}}^{\tau}_{\text{RPS}}) - \mathbb{J}_{\boldsymbol{\theta}_{\text{RPO}}}(\overline{\boldsymbol{\theta}}_{\text{RPO}}) \\
&= \left[\mathbb{J}^{\tau}_{\boldsymbol{\theta}^{\tau}_{\text{RPS}}}(\overline{\boldsymbol{\theta}}^{\tau}_{\text{RPS}}) - \mathbb{J}^{\tau}_{\boldsymbol{\theta}^{\tau}_{\text{RPO}}}(\overline{\boldsymbol{\theta}}^{\tau}_{\text{RPO}})\right] + \left[\mathbb{J}^{\tau}_{\boldsymbol{\theta}^{\tau}_{\text{RPO}}}(\overline{\boldsymbol{\theta}}^{\tau}_{\text{RPO}}) - \mathbb{J}_{\boldsymbol{\theta}_{\text{RPO}}}(\overline{\boldsymbol{\theta}}_{\text{RPO}})\right].
\end{aligned}
\tag{33}
$$

We now bound each term:

1. **First term**. Before we start, we first provide a bound on $\mathbb{J}^{\tau}_{\boldsymbol{\theta}^{\tau}_{\text{RPS}}}(\overline{\boldsymbol{\theta}}^{\tau}_{\text{RPS}}) - \mathbb{J}^{\tau}_{\boldsymbol{\theta}^{\tau}_{\text{RPS}}}(\overline{\boldsymbol{\theta}}^{\tau}_{\text{RPO}})$. From Lemma (F.2), the function $f(\boldsymbol{Z}, \overline{\boldsymbol{\theta}})$ is Lipschitz in $\overline{\boldsymbol{\theta}}$. Hence:

$$
\begin{aligned}
&\mathbb{J}^{\tau}_{\boldsymbol{\theta}^{\tau}_{\text{RPS}}}(\overline{\boldsymbol{\theta}}^{\tau}_{\text{RPS}}) - \mathbb{J}^{\tau}_{\boldsymbol{\theta}^{\tau}_{\text{RPS}}}(\overline{\boldsymbol{\theta}}^{\tau}_{\text{RPO}}) \\
&= \mathbb{E}_{\hat{\mathbb{P}}(\boldsymbol{\theta}^{\tau}_{\text{RPS}})}[\tau \lambda^2_{\text{RPS}} + f(\boldsymbol{Z}, \overline{\boldsymbol{\theta}}^{\tau}_{\text{RPS}})] - \mathbb{E}_{\hat{\mathbb{P}}(\boldsymbol{\theta}^{\tau}_{\text{RPS}})}[\tau \lambda^2_{\text{RPO}} + f(\boldsymbol{Z}, \overline{\boldsymbol{\theta}}^{\tau}_{\text{RPO}})] \\
&= \tau|(\lambda_{\text{RPS}} + \lambda_{\text{RPO}})(\lambda_{\text{RPS}} - \lambda_{\text{RPO}})| + \mathbb{E}_{\hat{\mathbb{P}}(\boldsymbol{\theta}^{\tau}_{\text{RPS}})}[f(\boldsymbol{Z}, \overline{\boldsymbol{\theta}}^{\tau}_{\text{RPS}}) - f(\boldsymbol{Z}, \overline{\boldsymbol{\theta}}^{\tau}_{\text{RPO}})] \\
&\leq (2\tau\overline{\lambda} + \rho^2 + D^2)|\lambda_{\text{RPS}} - \lambda_{\text{RPO}}| + L_{\theta}\|\boldsymbol{\theta}^{\tau}_{\text{RPS}} - \boldsymbol{\theta}^{\tau}_{\text{RPO}}\|_2.
\end{aligned}
\tag{34}
$$

where $\overline{\lambda}$ is the upper bound on the optimal $\lambda$ defined in (38). Now we provide a bound on the first term, using the decomposition:

$$
\begin{aligned}
&\mathbb{J}^{\tau}_{\boldsymbol{\theta}^{\tau}_{\text{RPS}}}(\overline{\boldsymbol{\theta}}^{\tau}_{\text{RPS}}) - \mathbb{J}^{\tau}_{\boldsymbol{\theta}^{\tau}_{\text{RPO}}}(\overline{\boldsymbol{\theta}}^{\tau}_{\text{RPO}}) \\
&= \mathbb{J}^{\tau}_{\boldsymbol{\theta}^{\tau}_{\text{RPS}}}(\overline{\boldsymbol{\theta}}^{\tau}_{\text{RPS}}) - \mathbb{J}^{\tau}_{\boldsymbol{\theta}^{\tau}_{\text{RPS}}}(\overline{\boldsymbol{\theta}}^{\tau}_{\text{RPO}}) + \mathbb{J}^{\tau}_{\boldsymbol{\theta}^{\tau}_{\text{RPS}}}(\overline{\boldsymbol{\theta}}^{\tau}_{\text{RPO}}) - \mathbb{J}^{\tau}_{\boldsymbol{\theta}^{\tau}_{\text{RPO}}}(\overline{\boldsymbol{\theta}}^{\tau}_{\text{RPO}}) \\
&\leq (2\tau\overline{\lambda} + \rho^2 + D^2)|\lambda_{\text{RPS}} - \lambda_{\text{RPO}}| + L_{\theta}\|\boldsymbol{\theta}^{\tau}_{\text{RPS}} - \boldsymbol{\theta}^{\tau}_{\text{RPO}}\|_2 + \epsilon L_z\|\boldsymbol{\theta}^{\tau}_{\text{RPS}} - \boldsymbol{\theta}^{\tau}_{\text{RPO}}\|_2 \\
&= (2\tau\overline{\lambda} + \rho^2 + D^2)|\lambda_{\text{RPS}} - \lambda_{\text{RPO}}| + (L_{\theta} + \epsilon L_z)\|\boldsymbol{\theta}^{\tau}_{\text{RPS}} - \boldsymbol{\theta}^{\tau}_{\text{RPO}}\|_2 \\
&\leq \frac{(2\tau\overline{\lambda} + \rho^2 + D^2 + L_{\theta} + \epsilon L_z)2\epsilon L_z}{\alpha}
\end{aligned}
\tag{35}
$$

where the first inequality comes from (31) and (34), the last inequality holds because of (32).

2. **Second term.** Using the fact that $\mathbb{J}^{\tau}_{\boldsymbol{\eta}}(\overline{\boldsymbol{\theta}})$ augments $\mathbb{J}_{\boldsymbol{\eta}}(\overline{\boldsymbol{\theta}})$ by $\tau\lambda^2$:

$$\mathbb{J}^{\tau}_{\boldsymbol{\theta}^{\tau}_{\text{RPO}}}(\overline{\boldsymbol{\theta}}^{\tau}_{\text{RPO}}) - \mathbb{J}_{\boldsymbol{\theta}_{\text{RPO}}}(\overline{\boldsymbol{\theta}}_{\text{RPO}}) \leq \mathbb{J}^{\tau}_{\boldsymbol{\theta}_{\text{RPO}}}(\overline{\boldsymbol{\theta}}_{\text{RPO}}) - \mathbb{J}_{\overline{\boldsymbol{\theta}}_{\text{RPO}}}(\overline{\boldsymbol{\theta}}_{\text{RPO}}) \leq \tau\overline{\lambda}^2 \tag{36}$$

Substituting these into (33), we obtain

$$\mathbb{J}^{\tau}_{\boldsymbol{\theta}^{\tau}_{\text{RPS}}}(\overline{\boldsymbol{\theta}}^{\tau}_{\text{RPS}}) - \mathbb{J}_{\boldsymbol{\theta}_{\text{RPO}}}(\overline{\boldsymbol{\theta}}_{\text{RPO}}) \leq \tau\overline{\lambda}^2 + \frac{(2\tau\overline{\lambda} + \rho^2 + D^2 + L_{\theta} + \epsilon L_z)2\epsilon L_z}{\alpha}.$$

This concludes the proof. $\qquad\square$

**Lemma F.1.** *Suppose that $\mathcal{Z}$ has a finite diameter $D = \sup_{z,z' \in \mathcal{Z}} \|z - z'\|_2 < \infty$ and $\Theta$ is bounded. Assume that the loss function $\ell(z, \theta)$ is $L_\theta$ Lipschitz continuous in $\theta$ and $L_z$-Lipschitz in $z$. Then there exist $\underline{\ell}, \bar{\ell} \in \mathbb{R}$ such that for all $(z, \theta) \in \mathcal{Z} \times \Theta$,*

$$\underline{\ell} \leq \ell(z, \theta) \leq \bar{\ell}.$$

*Consider the following univariate minimization problem*

$$\inf_{\lambda \in \mathbb{R}_+} \rho^2 \lambda + \tau \lambda^2 + \mathbb{E}_{\hat{\mathbb{P}}(\theta_t)}[\ell_c(z, \theta, \lambda)] \tag{37}$$

*where*

$$\ell_c(\hat{z}_s, \theta, \lambda) = \sup_{z \in \mathcal{Z}} (\ell(z, \theta) - \lambda \|z - \hat{z}_s\|_2^2).$$

*Then, the problem (37) admits a minimizer $\lambda^* \leq \bar{\lambda}$, where*

$$\bar{\lambda} = \frac{\sqrt{\rho^4 + 4\tau(\bar{\ell} - \underline{\ell})} - \rho^2}{2\tau}. \tag{38}$$

*Proof.* First, we observe that for fixed $(\theta, \hat{z}_s)$, the function $\ell_c(z, \theta, \lambda)$ is convex and lower semi-continuous in $\lambda$, as it is the supremum of functions affine in $\lambda$. Since lower semi-continuity is preserved under expectation over a discrete distribution $\hat{\mathbb{P}}(\theta_t)$, the objective function in (37) is convex and lower semi-continuous in $\lambda$.

Next, we establish a lower bound. Note that for all $\lambda \geq 0$,

$$\ell_c(\hat{z}_s, \theta, \lambda) \geq \ell(\hat{z}_s, \theta) \geq \underline{\ell},$$

since the supremum is attained at $z = \hat{z}_s$. Hence, we can bound the objective from below:

$$\rho^2 \lambda + \tau \lambda^2 + \mathbb{E}_{\hat{\mathbb{P}}(\theta_t)}[\ell_c(\hat{z}_s, \theta, \lambda)] \geq \rho^2 \lambda + \tau \lambda^2 + \underline{\ell}.$$

Thus, the infimum of problem (37) is attained for some $\lambda^* \in [0, +\infty)$. By optimality of $\lambda^*$, we then have:

$$\begin{aligned}
\rho^2 \lambda^* + \tau \lambda^{*2} + \mathbb{E}_{\hat{\mathbb{P}}(\theta_t)}[\ell(\hat{z}_s, \theta)] &\leq \rho^2 \lambda^* + \tau \lambda^{*2} + \mathbb{E}_{\hat{\mathbb{P}}(\theta_t)} \ell_c(\hat{z}_s, \theta, \lambda) \\
&\leq \mathbb{E}_{\hat{\mathbb{P}}(\theta_t)}[\ell_c(\hat{z}_s, \theta, 0)] \\
&\leq \mathbb{E}_{\hat{\mathbb{P}}(\theta_t)}[\sup_{z \in \mathcal{Z}} \ell(z, \theta)] \leq \bar{\ell},
\end{aligned}$$

where the second inequality comes from evaluating the objective at $\lambda = 0$. Rearranging the inequality, we obtain:

$$\rho^2 \lambda^* + \tau \lambda^{*2} \leq \bar{\ell} - \mathbb{E}_{\hat{\mathbb{P}}(\theta_t)}[\ell(\hat{z}_s, \theta)] \leq \bar{\ell} - \underline{\ell}.$$

Solving the quadratic inequality yields the upper bound

$$\lambda^* \leq \bar{\lambda} = \frac{\sqrt{\rho^4 + 4\tau(\bar{\ell} - \underline{\ell})} - \rho^2}{2\tau}.$$

This concludes the proof. $\square$

**Lemma F.2.** *Suppose $\mathcal{Z}$ has a finite diameter $D = \sup_{z,z' \in \mathcal{Z}} \|z - z'\|_2 < \infty$ and the loss function $\ell(z, \theta)$ is $L_\theta$ Lipschitz in $\theta$. For $\hat{z}_s \in \mathcal{Z}$, define the function*

$$f(\hat{z}_s, \bar{\theta}) = \sup_{z \in \mathcal{Z}} h(z, \hat{z}_s, \bar{\theta}), \tag{39}$$

*where $\bar{\theta} := (\theta, \lambda) \in \Theta \times \mathbb{R}_+ := \bar{\Theta}$, and*

$$h(z, \hat{z}_s, \bar{\theta}) = \ell(z, \theta) - \lambda \|z - \hat{z}_s\|_2^2 + \rho^2 \lambda.$$

*Then for any $\bar{\theta}, \bar{\theta}' \in \bar{\Theta}$, the function $f$ satisfies the Lipschitz bound:*

$$|f(\hat{z}_s, \bar{\theta}) - f(\hat{z}_s, \bar{\theta}')| \leq (\rho^2 + D^2)|\lambda - \lambda'| + L_\theta \|\theta - \theta'\|_2.$$

*Proof.* We start with the absolute difference between the two evaluations of $f$:

$$|f(\hat{z}_s, \overline{\theta}) - f(\hat{z}_s, \overline{\theta}')| = \max\{f(\hat{z}_s, \overline{\theta}) - f(\hat{z}_s, \overline{\theta}'), f(\hat{z}_s, \overline{\theta}') - f(\hat{z}_s, \overline{\theta})\}.$$

Next, we bound the first term; the second is symmetric. Let

$$\Delta = f(\hat{z}_s, \overline{\theta}) - f(\hat{z}_s, \overline{\theta}') = \sup_{z \in \mathcal{Z}} h(z, \hat{z}_s, \overline{\theta}) - \sup_{z' \in \mathcal{Z}} h(z', \hat{z}_s, \overline{\theta}').$$

Using the inequality

$$\sup_z a(z) - \sup_z b(z) \leq \sup_z (a(z) - b(z)),$$

we obtain

$$\begin{aligned}
\Delta &\leq \sup_{z \in \mathcal{Z}} \left[ h(z, \hat{z}_s, \overline{\theta}) - h(z, \hat{z}_s, \overline{\theta}') \right] \\
&= \sup_{z \in \mathcal{Z}} \left[ \ell(z, \theta) - \ell(z, \theta') - (\lambda - \lambda')\|z - \hat{z}_s\|^2 + \rho^2(\lambda - \lambda') \right] \\
&\leq \sup_{z \in \mathcal{Z}} |\ell(z, \theta) - \ell(z, \theta')| + \sup_{z \in \mathcal{Z}} |(\rho^2 - \|z - \hat{z}_s\|_2^2)(\lambda - \lambda')|
\end{aligned}$$

For the first term, since $\ell$ is $L_\theta$-Lipschitz in $\theta$:

$$|\ell(z, \theta) - \ell(z, \theta')| \leq L_\theta \|\theta - \theta'\|_2.$$

For the second term, observe that $\|z - \hat{z}_s\|_2 \leq D$ implies

$$|\rho^2 - \|z - \hat{z}_s\|_2^2| \leq \rho^2 + D^2.$$

Thus,

$$|(\rho^2 - \|z - \hat{z}_s\|_2^2)(\lambda - \lambda')| \leq (\rho^2 + D^2)|\lambda - \lambda'|.$$

By symmetry, the same bound holds for $f(\hat{z}_s, \overline{\theta}') - f(\hat{z}_s, \overline{\theta})$. This concludes the proof. $\square$

### F.4   Proof of Theorem 2

The proof of Theorem 2 follows directly from the results of Theorem 3, Theorem 4 and Theorem 5.

## G   Experiment Details

### G.1   Strategic Classification

Following [40, 32], we assume that individuals have linear utilities $u(\theta, \tilde{x}) = -\theta^\top \tilde{x}$ and quadratic costs $c(\tilde{x}', \tilde{x}) = -\frac{1}{2\epsilon}\|\tilde{x}' - \tilde{x}\|_2^2$, where $\epsilon$ is a positive constant regulating the cost of altering features and thus the sensitivity of the distribution map. In other words, individuals aim to minimize their assigned probability of default but are unable to change their true outcome $\tilde{y}$. We select $S \subseteq [P-1]$ strategic features, such as the number of open credit lines. Each time an individual manipulates their strategic features as depicted in [40, Section 5], the best response for an individual results in the update

$$\tilde{x}'_S = \tilde{x}_S - \epsilon \theta_S$$

where $\tilde{x}'_S, \tilde{x}_S, \theta_S \in \mathbb{R}^{|S|}$.

**Robust type-1.**   Consider the 1-Wasserstein ball, it follows from Theorem (1), at each time $t$, we can solve the following Tikhonov regularization problem

$$\mathbb{J}_{\theta_t}(\theta) = \inf_{\theta \in \Theta} \mathbb{E}_{\hat{\mathbb{P}}(\theta_t)} \left[ \log \left( 1 + \exp(-x^\top \theta \hat{y}) \right) \right] + \rho L \|(\theta, 1)\|_2^2.$$

**Robust type-2.** Consider the 2-Wasserstein ball, it follows from Proposition (1), at each time $t$, we can solve the following problem

$$\mathbb{J}_{\boldsymbol{\theta}_t}(\boldsymbol{\theta}) = \inf_{\lambda \in \mathbb{R}_+} \frac{1}{S} \sum_{s \in [S]} \sup_{\alpha \in (-1,0)} \left\{ \alpha \hat{\boldsymbol{x}}_s^\top \boldsymbol{\theta} \hat{y}_s + \frac{\alpha^2}{4\lambda} \boldsymbol{\theta}^\top \boldsymbol{\theta} - h(\alpha) \right\} + \rho^2 \lambda,$$

where $h(\alpha) = (\alpha + 1) \log(1 + \alpha) - \alpha \log(-\alpha)$.

Next, we introduce auxiliary variables $t_s \ \forall s \in [S]$ and combine with outer minimization over $\boldsymbol{\theta} \in \Theta$, which yield the equivalent formulation for the right hand side of the above inequality

$$\begin{aligned} \inf \quad & \frac{1}{S} \sum_{s \in [S]} t_s + \rho^2 \lambda \\ \text{s.t.} \quad & \boldsymbol{\theta} \in \Theta, \ \lambda \in \mathbb{R}_+, \ t_s \in \mathbb{R} \ \forall s \in [S] \\ & \sup_{\alpha \in (-1,0)} \left\{ \alpha \hat{\boldsymbol{x}}_s^\top \boldsymbol{\theta} \hat{y}_s + \frac{\alpha^2}{4\lambda} \boldsymbol{\theta}^\top \boldsymbol{\theta} - h(\alpha) \right\} \leq t_s \ \forall s \in [S]. \end{aligned} \tag{40}$$

To handle the last constraints, we discretize over some finite set $\mathcal{S}_\alpha$, i.e.,

$$\alpha \hat{\boldsymbol{x}}_s^\top \boldsymbol{\theta} \hat{y}_s + \frac{\alpha^2}{4\lambda} \boldsymbol{\theta}^\top \boldsymbol{\theta} \hat{y}_s^2 - h(\alpha) \leq t_s \ \forall \alpha \in \mathcal{S}_\alpha \ \forall s \in [S].$$

For the experiment we choose $\mathcal{S}_\alpha = \{-0.9, -0.8, \ldots, -0.1\}$.

**Proposition 1.** *Consider the logistic loss*

$$\ell(\boldsymbol{z}, \boldsymbol{\theta}) = \log \left( 1 + \exp(-\boldsymbol{x}^\top \boldsymbol{\theta} \hat{y}) \right),$$

*with $\boldsymbol{z} = (\boldsymbol{x}, \hat{y})$ and $\boldsymbol{\theta} \in \mathbb{R}^d$. Consider the 2-Wasserstein ball, where the ground cost $c$ is given by the Euclidean norm on $\mathbb{R}^d$. Assume the support set $\mathcal{X}$ is convex and closed, then we have*

$$\mathbb{J}_{\boldsymbol{\theta}_t}(\boldsymbol{\theta}) = \inf_{\lambda \in \mathbb{R}_+} \frac{1}{S} \sum_{s \in [S]} \sup_{\alpha \in (-1,0)} \left\{ \alpha \hat{\boldsymbol{x}}_s^\top \boldsymbol{\theta} \hat{y}_s + \frac{\alpha^2}{4\lambda} \boldsymbol{\theta}^\top \boldsymbol{\theta} - h(\alpha) \right\} + \rho^2 \lambda,$$

*where $h(\alpha) = (\alpha + 1) \log(1 + \alpha) - \alpha \log(-\alpha)$.*

*Proof.* We follow the argument in the proof of Theorem (3). For fixed $\boldsymbol{\theta}_t$, we have

$$\mathbb{J}_{\boldsymbol{\theta}_t}(\boldsymbol{\theta}) = \inf_{\lambda \in \mathbb{R}_+} \frac{1}{S} \sum_{s \in [S]} \sup_{\boldsymbol{z} \in \mathcal{Z}} \left[ \ell(\boldsymbol{z}, \boldsymbol{\theta}) - \lambda \| \boldsymbol{z} - \hat{\boldsymbol{z}}_s \|_2^2 + \rho^2 \lambda \right].$$

where $\hat{\boldsymbol{z}} = \hat{\boldsymbol{x}} \hat{y}_s$. Note that label $\hat{y}_s$ is fixed, we can simplify the inner supremum

$$\sup_{\boldsymbol{x} \in \mathcal{X}} \left[ \ell(\boldsymbol{x} \hat{y}_s, \boldsymbol{\theta}) - \lambda \| \boldsymbol{x} - \hat{\boldsymbol{x}}_s \|_2^2 + \rho^2 \lambda \right]$$

Next, substitute the logistic loss

$$\ell(\boldsymbol{x} \hat{y}_s, \boldsymbol{\theta}) = \log \left( 1 + \exp(-\boldsymbol{x}^\top \boldsymbol{\theta} \hat{y}_s) \right).$$

Using the Fenchel conjugate dual formulation [45] of the logistic loss:

$$\log(1 + e^{-u}) = \sup_{\alpha \in (-1,0)} \{\alpha u - h(\alpha)\}, \text{ where } h(\alpha) = (\alpha + 1) \log(1 + \alpha) - \alpha \log(-\alpha),$$

we rewrite the inner supremum as:

$$\begin{aligned} & \sup_{\boldsymbol{x} \in \mathcal{X}} \log(1 + \exp(-\boldsymbol{x}^\top \boldsymbol{\theta} \hat{y}_s)) - \lambda \| \boldsymbol{x} - \hat{\boldsymbol{x}}_s \|_2^2 \\ = & \sup_{\boldsymbol{x} \in \mathcal{X}} \sup_{\alpha \in (-1,0)} \left\{ \alpha \boldsymbol{x}^\top \boldsymbol{\theta} \hat{y}_s - h(\alpha) - \lambda \| \boldsymbol{x} - \hat{\boldsymbol{x}}_s \|_2^2 \right\} \\ = & \sup_{\alpha \in (-1,0)} \left\{ \sup_{\boldsymbol{x} \in \mathcal{X}} \left( \alpha \boldsymbol{x}^\top \boldsymbol{\theta} \hat{y}_s - \lambda \| \boldsymbol{x} - \hat{\boldsymbol{x}}_s \|_2^2 \right) - h(\alpha) \right\}. \end{aligned}$$

We now compute the inner supremum over $\boldsymbol{x}$. For fixed $\alpha$, $\hat{y}_s$, and $\boldsymbol{\theta}$, the expression

$$\alpha \boldsymbol{x}^\top \boldsymbol{\theta} \hat{y}_s - \lambda \|\boldsymbol{x} - \hat{\boldsymbol{x}}_s\|_2^2$$

is a concave quadratic in $\boldsymbol{x}$. The optimum is achieved at:

$$\boldsymbol{x}^* = \hat{\boldsymbol{x}}_s + \frac{\alpha}{2\lambda} \boldsymbol{\theta} \hat{y}_s.$$

Substituting $\boldsymbol{x}^*$ back in yields:

$$\alpha \hat{\boldsymbol{x}}_s^\top \boldsymbol{\theta} \hat{y}_s + \frac{\alpha^2}{4\lambda} \boldsymbol{\theta}^\top \boldsymbol{\theta}.$$

Therefore, the robust objective becomes:

$$\mathbb{J}_{\boldsymbol{\theta}_t}(\boldsymbol{\theta}) = \inf_{\lambda \in \mathbb{R}_+} \frac{1}{S} \sum_{s \in [S]} \sup_{\alpha \in (-1,0)} \left\{ \alpha \hat{\boldsymbol{x}}_s^\top \boldsymbol{\theta} \hat{y}_s + \frac{\alpha^2}{4\lambda} \boldsymbol{\theta}^\top \boldsymbol{\theta} - h(\alpha) \right\} + \rho^2 \lambda.$$

$\square$

**Alternative minimization under KL divergence ambiguity set (AMKL)** Consider the Kullback–Leibler (KL) divergence ambiguity set. According to [58, Proposition 3.1], the distributionally robust optimization problem under KL divergence can be reformulated as:

$$\min_{\boldsymbol{\theta}} \inf_{\mu \geq 0} \left\{ \mu \log \mathbb{E}_{\hat{\mathbb{P}}(\boldsymbol{\theta}_t)} \left[ \left(1 + \exp(-\boldsymbol{x}^\top \boldsymbol{\theta} \hat{y})\right)^{1/\mu} \right] + \mu \rho \right\},$$

where $\rho > 0$ is the radius of the ambiguity set. To solve this optimization problem, [58] propose an alternating minimization approach, which we refer to as AMKL. The procedure alternates between the following two steps:

1. $\boldsymbol{\theta}$-step: Fix the robustness parameter $\mu$, and minimize the objective with respect to $\boldsymbol{\theta}$.
2. $\mu$-step: Fix the model parameter $\boldsymbol{\theta}$, and minimize the objective with respect to $\mu$.

This iterative process is repeated until convergence. In the $\boldsymbol{\theta}$-step, we apply the repeated risk minimization algorithm, following the suggestion in [58] that this subproblem can be addressed using any suitable performative risk minimization algorithm.

### G.2 Impact of the Robust Parameter $\rho$

We investigate the impact of the robust parameter $\rho$ on out-of-sample performance. Specifically, we consider the revenue management problem described in section (4.2) under different quantities of perishable products, $q \in \{200, 300, 500\}$.

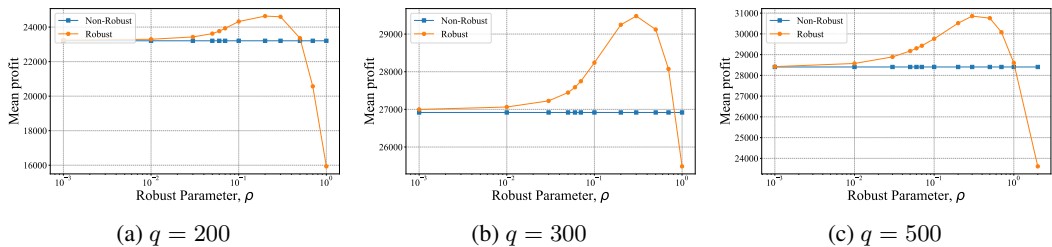

(a) $q = 200$     (b) $q = 300$     (c) $q = 500$

Figure 4: Out-of-sample performance as a function of the robust parameter $\rho$ and estimated on the basis of 100 simulations.

Figure 4 illustrates the optimal mean profit as a function of $\rho$, averaged over 100 independent simulation runs. We observe that the out-of-sample performance improves as $\rho$ increases up to a critical Wasserstein radius $\rho^*$, beyond which it begins to deteriorate. Notably, the robust model outperforms the non-robust counterpart over a wide range of $\rho$ values. This pattern was consistently observed across all simulation settings and provides an empirical justification for adopting a distributionally robust approach.

