# OpenReview forum: "Distributionally Robust Performative Optimization"
_NeurIPS.cc/2025/Conference — NeurIPS 2025 poster_

### Official Review · Reviewer_PzxX · 2025-07-02

**Clarity:** 2
**Significance:** 2
**Originality:** 2
**Rating:** 4
**Confidence:** 3

**Summary:**

This paper extends the distributionally performative robust framework to ambiguity sets captured by Wasserstein balls. The authors show a robust version of the repeated risk minimization algorithm linearly converges to a performatively robust stable fixed point in varying settings. These theoretical results are supplemented by experiments demonstrating the efficacy of their results on robust tasks.

**Questions:**

> * [Line 142] The term random parameters is ambiguous, is this the data point?
> * [Lines 151-152] The empirical reference distribution is a bit vague.
> * What are the primary difficulties in broadening the scope of the assumption to include all strongly convex functions?
> * Are the positive results tight, that is if any of the assumptions are replaced the convergence results will not hold?

**Ethical Concerns:**

["NO or VERY MINOR ethics concerns only"]

**Final Justification:**

This paper extends the distributionally robust framework to study distributions perturbed by Wasserstein balls. While the paper is well written, my initial concerns centered on its technical novelty, as it does not substantially distinguish itself from [1] aside from changes in the setting and the limited generality of the model itself, namely only dealing with quadratic loss functions.

In their response, the authors addressed these concerns by providing detailed insight into how the use of Wasserstein balls increases the proof’s complexity. They also explained how, beyond the usual contraction arguments in this literature, their proof leverages Danzig’s theorem and the exponential smoothing algorithm to handle the specific challenges of this work, and explained in detail during the rebuttal how and why these elements are used to derive the desired results.

Moreover, although the results are constrained to compositions of Lipschitz and quadratic functions, this is a notable generalization from previous work which only considers *bounded* loss functions.

For these reasons, I raise my score to 4 (Borderline Accept).

[1] Xue, S., & Sun, Y. (2024). Distributionally robust performative prediction. Advances in Neural Information Processing Systems, 37, 55030-55052.

**Limitations:**

Yes.

**Quality:**

3

**Strengths And Weaknesses:**

> Strengths
> * The paper is well written with clear exposition, ample motivation and a useful recap of related work.
> * This paper extends the established framework of distributionally robust performative optimization to include the more general case of Wasserstein balls as ambiguity sets rather than the KL divergence balls which fail when the support is not identical
> * This paper provides the first known convergence results for distributionally robust performative optimization.

> Weaknesses
> * The assumptions on the loss function in Model 1 are constraining, as the set of all quadratic and lipschitz combinations is a small subset of all strongly convex functions. A more general assumption would be expanding the quadratic function to be all strongly convex functions.
> * Beyond the reformulations in the models, the proofs of convergence are quite standard in the performative literature.
>  * Some additional discussion concerning the decision dependent nature of the experiments would be appreciated.
>  * The comparison lemma between performatively optimal and stable points is useful, but it would be helpful to construct a model with guarantees in finding performatively robust optimal points
>  * The intuition for the usefulness of exponential smoothing in this setting is not well explained.
> >  * *Minor Weakness* The results in appendix F are useful and might have more merits in the main body.
> >* *Minor Weakness* The notion of model in the paper is vague and it is unclear whether it refers to a Lemma or a set of algorithmic choices.

---

> ### Author Rebuttal · Authors · 2025-07-31
>
> We are grateful to the reviewer for their detailed and insightful comments. We have addressed each point carefully below and will reflect these improvements in the revised manuscript. We kindly invite the reviewer to reassess the paper in light of our responses.
>
> $\textbf{Weekness}$
>
> 1. Thank you for your comment. We would like to clarify that our model encompasses a broad class of problems studied in the existing performative optimization literature, including strategic classification [Perdomo et al. 2020] and strategic regression [Harris et al. 2021].  In fact, it represents one of the most general settings studied in the distributionally robust performative optimization framework. The dependence of the random parameters $Z$ is assumed to be affine in the loss function to ensure a tractable reformulation. Moreover, our Model 3 allows for a convex-concave loss function, which encompasses a wide range of applications. In light of this and related comments, we will revise the paper to highlight Model 3 more prominently in the main body.
> 2. Thank you for your comment. We agree that, like most of the existing performative optimization literature, our convergence analysis builds on the classical fixed-point iteration techniques. However, extending these tools to the distributionally robust setting introduces additional challenges. Specifically, our framework requires additional approximation techniques, such as exponential smoothing in Model 2, to handle non-smooth worst-case objectives. For Model 3, our analysis relies on Danskin’s theorem, which necessitates concavity of the loss function in the uncertainty to obtain a well-defined derivative of the worst-case objective. Adapting these tools to reformulate the problem into a tractable form while ensuring convergence involves careful technical treatment. We will clarify these aspects in the revised version.
> 3. Thank you for the suggestion. We agree that additional discussion on the decision-dependent nature of the experiments would be helpful. In all of our experimental settings, the data distribution changes as a function of the decision variable, reflecting the performative nature of the problem. For instance, in the strategic classification task, individuals respond strategically to the deployed classifier, thereby inducing a distribution shift dependent on the decisions. Similar decision-dependent dynamics are present in the revenue management and portfolio optimization settings, where decisions affect customer behavior and market conditions, respectively. We will revise the experimental section to make this connection more explicit.
> 4. Thank you for your suggestion. In this paper, we focus on convergence to distributionally performatively stable points and provide a comparison lemma that relates these stable points to performatively optimal solutions under distributional robustness. Constructing a model with guarantees for directly finding performatively robust optimal points would require stronger assumptions on the distribution map, as explored in prior work under non-robust settings [Izzo et al. 2022, Kim and Perdomo 2022]. However, such assumptions may not hold in many real-world scenarios. We view this as an important direction for future research and will include a discussion of it in the revised manuscript.
> 5. We apologize for the confusion. The piecewise quadratic structure of the loss function in Model 2 leads to non-smoothness, which hinders the convergence of our proposed algorithm. To address this challenge, we apply a log-sum-exp smoothing approximation with a smoothing parameter, which yields a differentiable surrogate loss function. We will revise the manuscript to better explain this intuition and the role of smoothing in our framework.
>  - Thank you for your suggestion. Due to page limits, we initially placed these results in the appendix. However, in response to your comment, we will move key results from Appendix F into the main body of the paper in the revised version to improve clarity and accessibility for readers.
>  - We apologize for the confusion. In our paper, Model 1, Model 2, and Model 3 each refer to distinct problem formulations that differ in their assumptions on the loss function and structure of the distributional map, which in turn lead to different reformulations of the distributionally robust performative optimization problem. These are not lemmas or algorithms, but rather modeling frameworks that guide both the algorithmic design and the theoretical analysis. We will clarify this terminology in the revised manuscript to avoid ambiguity.
>
> $\textbf{Questions}$
>
> 1. We apologize for the confusion. To clarify,$\tilde z$ is not a specific data point, but rather a random vector drawn from the unknown distribution $\mathbb{P}$.We will revise the text to make this distinction clearer.
> 2. We apologize for the confusion. In the performative optimization setting, the empirical reference distribution refers to the empirical distribution constructed from observed samples collected under a fixed decision $\theta$. Specifically, at each iteration, we observe a finite set of samples drawn from the corresponding distribution $P(\theta)$, and define the empirical reference distribution $\hat P(\theta)$ as the uniform distribution over these samples. Due to the limited sample size, $\hat P(\theta)$ may not represent the true distribution. We will revise the text to make this definition more precise.
> 3. Thank you for your question. Broadening the assumption to include all strongly convex functions poses challenges, primarily because it may prevent a tractable reformulation of the distributionally robust problem. In our setting, we assume that the dependence of the random parameters $Z$ is affine in the loss function to ensure such tractability. Extending the analysis to more general strongly convex functions would likely require new machinery and is an interesting direction for future work.
>
>     That said, we would like to emphasize that our model is already quite general and encompasses a broad class of problems studied in the performative optimization literature, including strategic classification [Perdomo et al. 2020]  and strategic regression [Harris et al. 2021].
> 4. Thank you for your comment. The assumptions in our convergence analysis are chosen to enable a tractable and rigorous analysis using established tools such as Danskin’s theorem and fixed-point methods. While these assumptions are not necessarily tight, they are carefully selected to reflect the key structural properties that underpin the robustness and stability of our algorithm.
>
>     Relaxing any of them would likely require new proof techniques or a different algorithmic framework. At the same time, we believe our assumptions strike a reasonable balance between generality and tractability, as they already cover a broad class of performative optimization problems studied in prior work. We will clarify this point in the revised manuscript and discuss opportunities for generalization in future research.
>
> $\textbf{References}$
>
> - J. Perdomo, T. Zrnic, C. Mendler-Dünner, and M. Hardt. Performative prediction. In International Conference on Machine Learning, pages 7599–7609. PMLR, 2020
> - K. Harris, H. Heidari, and S. Z. Wu. Stateful strategic regression. Advances in Neural Information Processing Systems, 34:28728–28741, 2021
> - Z. Izzo, L. Ying, and J. Zou. How to learn when data reacts to your model: performative gradient descent. In International Conference on Machine Learning, pages 4641–4650. PMLR, 2021.
> - M. P. Kim and J. C. Perdomo. Making decisions under outcome performativity. arXiv preprint arXiv:2210.01745, 2022

---

> > ### Comment · Reviewer_PzxX · 2025-08-05
> >
> > Thank-you for your comments and confirming the theoretical novelty of the work via the arguments which use Danzig's theorem and the exponential smoothing algorithm. It would be nice if there was additional background given on exponential smoothing (perhaps in the appendix). In light of these comments, I will increase my score to borderline accept.

---

> > > ### Author Response · Authors · 2025-08-07
> > >
> > > Dear Reviewer PzxX,
> > >
> > > Thank you very much for your thoughtful comments and for recognizing the $\textbf{theoretical novelty}$ of our work, particularly the use of $\textit{Danskin’s theorem}$ and the $\textit{exponential smoothing algorithm}$.
> > >
> > > We appreciate your suggestion to provide more background on exponential smoothing, and we have accordingly added a detailed discussion in the Appendix of the revised manuscript. Below, we elaborate further on both tools, their motivation, and the nontrivial challenges involved in applying them within our setting.
> > >
> > > $\textbf{Use of Danskin’s Theorem}$
> > >
> > > We leverage $\textit{Danskin’s Theorem}$ to characterize the gradient of our robust objective function, which involves an implicit dependence of the data distribution on the decision variable. This is essential in our distributionally robust performative optimization setting, where the loss function depends on a $\textit{distributionally dependent}$ or $\textit{response sensitive}$ term.
> > >
> > > More concretely, we consider the distributionally robust performative risk:
> > > \begin{equation}
> > > \min_{\theta \in \Theta} \sup_{Q \in B(\hat{P}( \theta_t))} J_{\theta_t}(\theta).
> > > \end{equation}
> > > More concretely, by applying the minimax theorem [Blanchet and Murthy, 2019],  we derive an arbitrarily tight conservative approximation for the original problem (see Model 3 in Appendix F). This yields:
> > > \begin{align*}
> > >     J_{\theta_t}^\tau(\overline \theta ) = E_{ \hat{P}(\theta_t) } [g( Z, \overline \theta )],
> > > \end{align*}
> > > where the function
> > > \begin{align*}
> > > g( Z, {\overline \theta}) = \sup_{ z\in \mathcal Z} \ell(z, \theta) -\lambda \|  z - Z \|_2^2  + \rho^2 \lambda + \tau \lambda^2,
> > > \end{align*}
> > > is the regularized robust decoupled performative risk. Here, $g$ arises from the dual formulation, $\rho$ denotes the radius of the Wasserstein ambiguity set, $\tau \in \mathbb{R}{++}$ is a regularization parameter, and $\overline{ \theta} = ( \theta, \lambda) \in \Theta \times \mathbb{R}+ = {\overline \Theta}$ is the augmented variable including the dual multiplier $\lambda$.
> > >
> > > Danskin’s theorem provides a principled way to compute directional derivatives or subgradients of such nonsmooth, supremum-type objectives. However, applying Danskin’s theorem in our setting is \textbf{nontrivial}. To ensure strong convexity in the joint variable $\overline{\theta}$, which is important for the convergence guarantees of the R$^3$M algorithm, we introduce a regularization term $\tau \lambda^2$, motivated by the reformulation of our Model 3.
> > >
> > > Importantly, for any $\theta\in \Theta$, we have that $\inf_{\lambda\geq 0} J_{ \theta_t}^\tau ( \theta,\lambda)$ converges to $J_{ \theta_t} (\theta)$ as $\tau\downarrow 0$.  If $\ell(z,  \theta)$ is lower semicontinuous in $\theta$, then $J_{\theta_t}$ inherits this property. Consequently, $\inf_{\lambda \geq 0} J_{\theta_t}^\tau(\cdot, \lambda)$ epi-converges to $J_{\theta_t}(\cdot)$ [Rockafellar and Wets, 2009, Thm 7.17].  Moreover, the minimizer $\hat{\theta}$ of $\inf J_{\theta_t}^\tau(\overline \theta)$ converges to a minimizer of $\inf J_{\theta_t}(\theta)$ provided $\Theta$ is compact [Rockafellar and Wets, 2009, Thm 7.33].
> > >
> > > We appreciate your suggestion and have included the above discussion, along with additional references, in the revised appendix. Please let us know if further clarification would be helpful.
> > >
> > > Thank you again for your careful review and constructive feedback.
> > >
> > > Sincerely,
> > >
> > > The Authors
> > >
> > >
> > > $\textbf{References:}$
> > >
> > > - J. Blanchet and K. Murthy. Quantifying distributional model risk via optimal transport. Mathematics of Operations Research, 44(2):565–600, 2019.
> > > - R. T. Rockafellar and R. J.-B. Wets. Variational Analysis. Springer, 2009.

---

> ### Author Response · Authors · 2025-08-07
>
> Dear Reviewer PzxX,
>
> Continuing from our earlier response, we now provide a detailed explanation of the exponential smoothing algorithm, its mathematical role in our framework, and the theoretical challenges it helps address.
>
> $\textbf{Background on Exponential Smoothing Algorithm}$
>
> Let $\mathcal{Z} = \{ z_1, \dots, z_n\}$ denote a finite support set. Given a loss function $\ell: \Theta \times \mathcal{Z} \to \mathbb{R}$ and a distribution ${\hat p}_{\theta} \in \Delta_n$, where $\Delta_n$ denotes the probability simplex. We define the exponentially smoothed objective as:
>
> \begin{align*}
> f_\mu(\theta) := \mu \cdot \log \left( \sum_{i=1}^n \hat{p}_\theta(z_i) \cdot \exp\left( \frac{\ell(\theta, z_i)}{\mu} \right) \right),
> \end{align*}
> where $\mu > 0$ is a smoothing parameter.
>
> This function, also known as the log-sum-exp function, is a widely used smooth approximation to the pointwise maximum function and has well-established properties in convex analysis  [Bertsekas, 2015; Section 2.2]. This function provides a smooth approximation to $\max_{i} \ell(\theta, z_i)$ whenever $\hat{p}_\theta(z_i) > 0$ for all $i$.
>
> $\textbf{Properties.}$ The function $f_\mu(\theta)$ satisfies the following:
>
>  - Approximation Bounds: The approximation error can be precisely quantified:
> \begin{equation}
> \max_{i} \ell(\theta, z_i) \leq f_\mu(\theta) \leq \max_{i} \ell(\theta, z_i) + \mu \cdot \log \left( \frac{1}{\min_i \hat{p}_\theta(z_i)} \right),
> \end{equation}
>
> provided $\hat{p}\_\theta(z_i) > 0$ for all $i$. Thus, as $\mu \to 0$, the smoothed objective $f_\mu(\theta)$ approaches the exact maximum, with the error vanishing linearly in $\mu$ up to a logarithmic multiplicative factor.
>
> - Convexity and Differentiability: If each function $\ell(\theta, z_i)$ is convex in $\theta$, then $f_\mu(\theta)$ is also convex, as it is a composition of convex functions closed under nonnegative weighted log-sum-exp operations. Moreover, $f_\mu(\theta)$ is continuously differentiable for all $\mu > 0$, with gradient:
> \begin{align*}
> \nabla_\theta f_\mu(\theta) = \sum_{i=1}^n \pi_\mu(z_i; \theta) \cdot \nabla_\theta \ell(\theta, z_i),
> \end{align*}
>
> where the weight vector $\pi_\mu(\cdot; \theta) \in \Delta_n$ defines a softmax distribution:
> \begin{align*}
> \pi\_\mu(z_i; \theta) := \frac{\hat{p}\_\theta(z_i) \cdot \exp\left( \ell(\theta, z_i)/\mu \right)}{ \sum_{j=1}^n \hat{p}_\theta(z_j) \cdot \exp\left( \ell(\theta, z_j)/\mu \right)}.
> \end{align*}
>
>
> This smoothing mechanism not only ensures differentiability but also facilitates efficient computation of gradients for robust optimization objectives involving maxima.
>
> -----------------
>
>
> We appreciate your suggestion and have included the above discussion, along with additional references, in the revised appendix. Please let us know if further clarification would be helpful.
>
> Thank you again for your careful review and constructive feedback.
>
> Sincerely,
>
> The Authors
>
>
> - Bertsekas, Dimitri. Convex optimization algorithms. Athena Scientific, 2015.

---

### Official Review · Reviewer_1Jfc · 2025-07-02

**Clarity:** 3
**Significance:** 2
**Originality:** 3
**Rating:** 4
**Confidence:** 3

**Summary:**

This paper explores performative stochastic optimization, where decisions affect the underlying distribution of uncertain parameters, focusing on robust solutions when the true decision-dependent distribution is unknown. The authors propose a distributionally robust performative optimization (DRPO) framework using Wasserstein ambiguity sets, covering both machine learning and more general decision-making under uncertainty. The paper presents theoretical convergence and suboptimality guarantees and reformulations compatible with standard solvers. Experiments in strategic classification, revenue management, and portfolio optimization indicate consistent performance gains over baselines.

**Questions:**

In Figure 1, the performance gap between the robust type-1 and type-2 Wasserstein ball models is shown, but the practical difference is not deeply discussed: is the difference mainly due to the geometry of the ambiguity set, problem-specific characteristics, or hyperparameter tuning?

**Ethical Concerns:**

["NO or VERY MINOR ethics concerns only"]

**Final Justification:**

The authors' rebuttal helps to address some of my concerns and clarify the contribution of this paper, and additional experiments will be added in the future revision. Therefore, I keep my score unchanged. (4: Borderline accept)

**Limitations:**

Yes

**Paper Formatting Concerns:**

No formatting concerns

**Quality:**

3

**Strengths And Weaknesses:**

## Strengths

1. The authors introduce a distributionally robust performative optimization (DRPO) framework that is broader in scope than previous works, notably supporting a wider variety of decision-making beyond traditional prediction tasks.

2. This paper theoretically establishes convergence guarantees for the repeated robust risk minimization algorithm with both smooth and some nonsmooth loss functions, which broadens applicability.

3. This paper conducts experimental evaluations in three distinct application domains. In each, the proposed approach consistently outperforms non-robust and relevant existing rivals.

## Weaknesses

1. While the experimental section spans three domains, the treatment of each is somewhat brief. For example, for the revenue management and portfolio optimization tasks (Section 4.2), quantitative comparisons are mostly limited to a single robust method versus a standard baseline; only the strategic classification task benchmarks multiple methods, as shown in Figure 1.

2. More extensive ablation or sensitivity analyses (e.g., systematically varying ambiguity set sizes, number of iterations, etc.), particularly for the portfolio and revenue management use cases, would strengthen the empirical story.

3. This paper claims the reformulations are “compatible with off-the-shelf solvers,” but fails to provide concrete runtime comparisons or scalability analyses to contextualize practical tractability.

4.  Section 1.1(Related Work) positions the work well within the field but could benefit from a more explicit contrast with the limitations and strengths of direct alternatives, for instance, quantifying where/when the paper’s approach provides superior robustness or convergence.

---

> ### Author Rebuttal · Authors · 2025-07-31
>
> We are grateful to the reviewer for their detailed and insightful comments. We have addressed each point carefully below and will reflect these improvements in the revised manuscript. We kindly invite the reviewer to reassess the paper in light of our responses.
>
> $\textbf{Weekness}$
>
> 1. Thank you for your constructive comment. We would like to clarify that, to the best of our knowledge, unlike in machine learning problems, there are no existing baselines applicable to the decision-dependent distributionally robust setting we consider for the revenue management and portfolio optimization tasks. While there are related works in the stochastic programming literature (e.g., Fonseca and Junca 2023, Noyan et al. 2022) that study decision-dependent uncertainty, such as models where the decision affects the radius of the ambiguity set, these approaches are not directly applicable to our framework. We will revise the manuscript to make this distinction clearer and emphasize that our aim in these experiments is to demonstrate the flexibility and applicability of our approach to realistic decision-making problems beyond strategic classification.
> 2. Thank you for your suggestion. In our current experiments, we fixed the size of the ambiguity set across all robust models to be 0.1 in the strategic classification task. However, due to space constraints, we were unable to include more sensitivity analyses for the portfolio and revenue management tasks in the main paper. In the revised version, we will include additional results that explore the sensitivity analysis.
> 3. We apologize for the confusion. Our claim that the reformulations are "compatible with off-the-shelf solvers" refers to the fact that after applying the reformulations in Models 1–3, the resulting optimization problems can be cast as convex programs (e.g., exponential cone programs or smooth convex problems) that are solvable using standard solvers such as Gurobi, Mosek, or CVXPY-based backends.
>
>     We would like to emphasize that the limited data regime we consider is not just a practical constraint, but a characteristic feature of the performative optimization setting. Unlike traditional machine learning, where data is assumed to be drawn from a fixed or evolving distribution, performative optimization requires sampling from distributions that are themselves induced by the decisions. That is, in order to evaluate or improve a policy, one must actively deploy a decision (e.g., hiring strategies, pricing policy) and then observe the resulting data distribution, which can be expensive, time-consuming, or infeasible at scale [Milli et al. 2019, Mendler-Dünner et al. 2020]. Our method is designed to be data-efficient in this regime, and we aim to achieve robust and stable performance even with limited samples per iteration. We will clarify this distinction and expand on the computational aspects in the revised manuscript.
> 4. Thank you for the suggestion. To the best of our knowledge, there are only a few existing works that address distributionally robust performative prediction. For example, [Peet et al. 2022] propose a distributionally robust performative model to promote fairness in machine learning. However, their framework relies on a robust smoothness assumption on the objective and a robust sensitivity assumption on the worst-case distributions, both of which may be unrealistic in practice. Additionally, their use of a $\phi$-divergence ambiguity set excludes continuous distributions, thereby limiting its ability to provide coverage guarantees over the true distribution. Similarly, [Xue and Sun 2024] study distributionally robust performative prediction based on a KL-divergence ambiguity set. Like [Peet et al. 2022], such divergence-based ambiguity sets may fail to protect against distributional shifts involving novel or unseen scenarios outside the reference distribution. Moreover, their proposed scheme lacks convergence guarantees.
>
>     By contrast, our work adopts a Wasserstein ambiguity set, which provides coverage guarantees for continuous distributions, and we establish provable convergence guarantees under weaker assumptions. We will revise Section 1.1 to make this comparison more explicit.
>
> $\textbf{Questions}$
>
> 1. We thank you for the clarifying question. The performance gap observed between the robust type-1 and type-2 Wasserstein ball models in Figure 1 arises primarily due to the geometry of the ambiguity set. As discussed in [Byeon 2025], selecting the optimal radius $\rho$ is challenging, and the 2-Wasserstein ball often provides better performance because it offers a wider range of radius values for which the DRO solution outperforms its non-robust counterpart. In our experiments, we set the robustness parameter to 0.1 across all robust models to ensure comparability. We will clarify this in the revised manuscript and expand the discussion of how the ambiguity set geometry influences performance.
>
> $\textbf{References}$
>
> - D. Fonseca and M. Junca. Decision-dependent distributionally robust optimization. arXiv preprint arXiv:2303.03971, 2023
> - N. Noyan, G. Rudolf, and M. Lejeune. Distributionally robust optimization under a decision-dependent ambiguity set with applications to machine scheduling and humanitarian logistics. INFORMS Journal on Computing, 34(2):729–751, 2022
> - S. Milli, J. Miller, A. D. Dragan, and M. Hardt. The social cost of strategic classification. In Proceedings of the conference on fairness, accountability, and transparency, pages 230–239, 2019
> - C. Mendler-Dünner, J. Perdomo, T. Zrnic, and M. Hardt. Stochastic optimization for performative prediction. Advances in Neural Information Processing Systems, 33:4929–4939, 2020
> - L. Peet-Pare, N. Hegde, and A. Fyshe. Long term fairness for minority groups via performative distributionally robust optimization. arXiv preprint arXiv:2207.05777, 2022.
> - S. Xue and Y. Sun. Distributionally robust performative prediction. Advances in Neural Information Processing Systems, 37:55030–55052, 2024
> - G. Byeon. Comparative analysis of two-stage distributionally robust optimization over 1-wasserstein and 2-Wasserstein balls. arXiv preprint arXiv:2501.05619, 2025

---

> > ### Comment · Reviewer_1Jfc · 2025-08-03
> >
> > Thanks for the detailed reply, it helps to address some of my concerns. As the additional experiment will be included in the future revision, I will maintain my positive score unchanged.

---

> ### Author Response · Authors · 2025-08-06
>
> Dear Reviewer 1Jfc,
>
> We deeply appreciate the time and effort you devoted to carefully reviewing our work and providing thoughtful, detailed feedback. Your insightful questions and constructive suggestions have been invaluable in enhancing both the technical depth and clarity of our paper.
>
> Thank you once again for your thoughtful review and support.
>
> Sincerely,
>
> The Authors

---

### Official Review · Reviewer_zEFG · 2025-07-03

**Clarity:** 2
**Significance:** 2
**Originality:** 3
**Rating:** 4
**Confidence:** 2

**Summary:**

The paper introduces a distributionally robust framework for performative optimization. It proposes a tractable iterative algorithm with theoretical convergence and suboptimality guarantees. Empirical results on strategic classification, pricing, and energy applications demonstrate improved robustness over baselines.

**Questions:**

Q1: As mentioned in W1, the paper focuses on distributionally robust performative risk (DRPR), but many applications care about performance with respect to the true performative risk (PR). Is it possible to provide any theoretical guarantees or bounds on the PR of the proposed method?

Q2. Could the authors elaborate more on why "While methods grounded in the reference distribution map might perform satisfactorily on the observed samples, they often fail to achieve acceptable performance in out-of-sample circumstances" in the literature? Specifically, if the reference distribution is close to the true distribution, wouldn't those methods still perform reasonably well? If the main challenge is in estimating a good reference distribution, how does the proposed framework help mitigate that?

Q3: How to choose \epsilon in practice?

Q4: What will be the problem-specific parameters, such as alpha, rho, beta..., in the numerical problems? Will the proposed method still work if those parameters are bad in terms of the bound in Theorem 2?

**Ethical Concerns:**

["NO or VERY MINOR ethics concerns only"]

**Final Justification:**

Thank the authors for the detailed explanations, addressing my concerns.

**Quality:**

2

**Strengths And Weaknesses:**

Strengths:
S1: The paper proposes a new tractable method for performative optimization that incorporates distributional robustness to address model uncertainty.

S2: The method is supported by both theoretical analysis and empirical evaluation across multiple applications.

Weaknesses:
W1: While the theoretical results focus on minimizing distributionally robust performative risk (DRPR), it would strengthen the work to connect these guarantees more directly to the performative risk (PR), which is often the ultimate objective.

W2: The paper claims to address the out-of-sample limitations of methods relying on reference distributions. However, it is unclear how the proposed method helps when the reference distribution itself is poorly specified (see Q2 for more details).

W2: Models 1 and 2 seem restrictive. It would be helpful to clarify whether these models cover a broad class of problems studied in prior performative optimization literature.

---

> ### Author Rebuttal · Authors · 2025-07-31
>
> We sincerely appreciate the reviewer’s thoughtful and constructive feedback. We will incorporate these insights into the revised manuscript and hope that the responses below help clarify our contributions and address the reviewer's concerns during the reevaluation process.
>
> $\textbf{Weekness}$
>
> 1. We really appreciate the constructive feedback from the reviewer. We agree that connecting our theoretical guarantees for minimizing the distributionally robust performative risk (DRPR) to the performative risk (PR) is important, especially since PR is often the ultimate objective.
>
>     However, optimizing PR directly relies heavily on modeling the distribution map that captures how deployed decisions influence future data distributions. In practice, this map is often unknown or misspecified, which can result in poor estimates of the true PR. Our framework is specifically designed to address this challenge by optimizing a robust surrogate (DRPR) that mitigates the impact of such model misspecification.
>
>     Unlike existing work [Perdomo et al. 2020, Mendler-Dünner et al. 2020] in performative optimization literature that typically assumes access to the true distribution map, our approach relies on much weaker assumptions. Moreover, we would like to emphasize that the limited data regime we consider is not simply a practical constraint but an inherent characteristic of the performative setting. In contrast to traditional machine learning, where data can be passively observed from a fixed or evolving distribution, performative optimization requires active data collection under deployed decisions (e.g., hiring, pricing), making large-scale sampling expensive or infeasible [Milli et al. 2019, Mendler-Dünner et al. 2020].
>
>     Our method is designed to be data-efficient and robust in this setting, offering reliable performance even with a small number of samples per iteration. We will clarify this distinction and expand on the connection between DRPR and PR in the revised manuscript.
> 2. We apologize for the confusion. To ensure out-of-sample performance guarantees, a standard assumption in distributionally robust optimization is that the true data-generating distribution is contained in the ambiguity set. Leveraging the concentration inequality for the Wasserstein distance between the empirical and true distributions, we can ensure this with high confidence by appropriately choosing the radius. As shown in~\cite[Theorem 3.4]{mohajerin2018data}, given $N$ sample data points, we have the data distribution $P$ lies within a Wasserstein ball around the empirical reference distribution $\hat P$ of radius $\rho_N(\beta)$ with confidence level $1-\beta$ for some prescribed $\beta \in (0,1)$, more specifically,
>     \begin{align*}
>         P^n[W(P, \hat P) \leq \rho_N(\beta)] \geq 1 - \beta.
>     \end{align*}
>     Now, if there is a distribution shift between the empirical reference distribution and the test distribution, and we train the algorithm based on samples from the reference distribution. Then, a typical assumption is that the magnitude of the distribution shift is known in terms of Wasserstein distance $\tau$. Then, by the triangle inequality for Wasserstein distance, the radius of the ambiguity set is given by $\tau$ plus the concentration inequality above. We will clarify this in the revision and more clearly explain how it enables out-of-sample performance guarantees even in the presence of moderate distribution shift.
> 3. We apologize for the confusion and appreciate the reviewer's comment. We would like to clarify that both models are designed to encompass a broad class of problems studied in the existing performative optimization literature. For example, Model 1 captures widely used predictive models such as strategic classification [Perdomo et al. 2020], strategic regression [Harris et al. 2021]. Model 2 accommodates problems in decision-making problems including inventory management [Lee et al. 2021] and energy systems [Kim and Powell, 2011].
>
>     Additionally, we would like to highlight that Model 3, presented in Appendix F, further generalizes the framework to accommodate problems studied in the minimax and adversarially robust optimization literature. In light of Reviewer 4’s comment, we will move the description of Model 3 and its associated results into the main text. We will also clarify how the three models relate to and extend existing approaches in the literature.
>
> $\textbf{Questions}$
>
> 1. We thank the reviewer for the great question. As we noted in response to Weakness 1, our method is designed to be both data-efficient and robust in settings where the distribution map is unknown or potentially misspecified.
>
>     A key challenge in this setting is that, unlike traditional distribution robust optimization, which we typically assume a single known true distribution [Esfahani and Kuhn, 2018], performative optimization involves a distribution that changes with the decision variable. This makes it significantly more difficult to relate the robust objective directly to the true performative risk.
>
>     That said, we note that as the ambiguity set radius $\rho$ tends to zero, the robust objective coincides with the non-robust counterpart, which more directly targets the PR. This observation suggests that one possible direction is to design algorithms that gradually shrink the ambiguity set over time, potentially trading robustness for improved approximation of the true PR as more information becomes available. Developing theoretical guarantees for such adaptive schemes would require new technical tools and is an interesting direction for future work. We will add a discussion of this in the revised version.
> 2. We apologize for the confusion. Please refer to our response to Weakness 2.
> 3. We thank the reviewer for the question. We would like to clarify that $\epsilon$ is the property of the underlying distribution mapping and is not a parameter. Unlike prior work on decision-dependent distribution shift [Perdomo et al. 2020, Mendler-Dünner et al. 2020], where the mapping P itself is known, our assumption is much weaker. In practice, given two decisions $\theta$ and $\theta'$, along with their corresponding empirical reference distributions $\hat P(\theta)$ and $\hat P(\theta')$ based on observed samples, one can compute the 1-Wasserstein distance $W_1(P(\theta), P(\theta'))$ via linear programming. This allows for an estimate of $\epsilon$ using $\frac{ W_1(P(\theta), P(\theta'))}{\|\theta - \theta' \|_2}$.
> 4. We apologize for the confusion. Here, the parameters such as $\alpha,\beta,\gamma, L_z, L, \rho, \epsilon$ are problem-specific constants that characterize properties of the loss function, the distributional map, and the ambiguity set. These are not tunable hyperparameters, and their values are not required for implementing our algorithm. Instead, they appear in the convergence analysis to establish theoretical guarantees. In practice, for example, in the strategic classification task, $L = 1$ for the logistic loss function [Gao and Pavel 2017] and the logloss objective is known to be $\frac{1}{4n}\sum_{i=1}^n \|x_i\|_2^2+ 2\rho L$ smooth [Shwartz and David 2014]. We varied $\epsilon$ as shown in Figure 1. The radius $\rho$ of the Wasserstein ball was fixed to 0.1 across all robust models. We emphasize that our method is robust to the lack of explicit knowledge of these constants and remains effective in practice. We will clarify this in the revised manuscript.
>
> $\textbf{References}$
>
> - J. H. Kim and W. B. Powell. Optimal energy commitments with storage and intermittent supply. Operations Research, 59(6):1347–1360, 2011.
> - S. Shalev-Shwartz and S. Ben-David. Understanding machine learning: From theory to algorithms. Cambridge University Press, 2014
> - B. Gao and L. Pavel. On the properties of the softmax function with application in game theory and reinforcement learning. arXiv preprint arXiv:1704.00805, 2017
> - P. Mohajerin Esfahani and D. Kuhn. Data-driven distributionally robust optimization using the wasserstein metric: Performance guarantees and tractable reformulations. Mathematical Programming, 171(1):115–166, 2018
> - S. Milli, J. Miller, A. D. Dragan, and M. Hardt. The social cost of strategic classification. In Proceedings of the conference on fairness, accountability, and transparency, pages 230–239, 2019
> - J. Perdomo, T. Zrnic, C. Mendler-Dünner, and M. Hardt. Performative prediction. In International Conference on Machine Learning, pages 7599–7609. PMLR, 2020
> - C. Mendler-Dünner, J. Perdomo, T. Zrnic, and M. Hardt. Stochastic optimization for performative prediction. Advances in Neural Information Processing Systems, 33:4929–4939, 2020
> - M. Jagadeesan, T. Zrnic, and C. Mendler-Dünner. Regret minimization with performative feedback. In International Conference on Machine Learning, pages 9760–9785. PMLR, 2022
> - K. Harris, H. Heidari, and S. Z. Wu. Stateful strategic regression. Advances in Neural Information Processing Systems, 34:28728–28741, 2021
> - S. Lee, H. Kim, and I. Moon. A data-driven distributionally robust newsvendor model with a wasserstein ambiguity set. Journal of the Operational Research Society, 72(8):1879–1897, 2021

---

### Official Review · Reviewer_MXSY · 2025-07-05

**Clarity:** 3
**Significance:** 3
**Originality:** 3
**Rating:** 4
**Confidence:** 2

**Summary:**

This paper tackles performative optimization where decisions influence the distribution of uncertain parameters (e.g., algorithmic predictions affecting user behavior). The key challenge is that practitioners only have access to imperfect reference distributions rather than true decision-dependent distribution maps, leading to poor performance under model misspecification. The authors propose a distributionally robust performative optimization (DRPO) framework using Wasserstein ambiguity sets to optimize against worst-case distributions within a neighborhood of the reference distribution. The core innovation is a decoupling strategy that separates the decision variable in the ambiguity set from the objective function, transforming an intractable bilevel optimization into manageable conic programs.

**Questions:**

While the authors mention that 2-Wasserstein seems "limited to machine learning settings," I wonder if there might be alternative mathematical approaches that could potentially overcome this limitation? It would be helpful to better understand how significant this restriction might be in practice.

**Ethical Concerns:**

["NO or VERY MINOR ethics concerns only"]

**Limitations:**

Yes. While the authors acknowledge some limitations (e.g., current focus on 1-Wasserstein, 2-Wasserstein restrictions), authors should include a brief discussion of how experimental scale might affect conclusions and the need for larger-scale validation.

**Paper Formatting Concerns:**

I did not identify any major formatting issues.

**Quality:**

3

**Strengths And Weaknesses:**

Strengths
1.This paper appears to make a contribution by systematically introducing distributionally robust optimization into the performative optimization framework, which seems to address an important gap in this topic.
2.The decoupling strategy appears to be  mathematically elegant with reasonable theoretical justification.
3.The three proposed models seem to cover a reasonably broad spectrum of applications, demonstrating the potential versatility of the framework.
Weaknesses
1.The convergence guarantees appear to depend critically on the ε-sensitivity assumption $W_1(P(\theta)- P(\theta)^`)\leq ε||\theta - \theta^` ||_2$. However, it seems that the paper might benefit from providing more guidance on how practitioners could estimate or verify ε in real applications.
2.The strategic classification experiment appears to use 200 training samples, which might seem modest by current ML standards.
3.The paper might benefit from a clearer explanation of why existing DRO methods cannot be directly adapted to the performative setting. Perhaps a more detailed comparison could help readers better appreciate the unique challenges that necessitate the proposed approach
4.While mentioned in limitation, the significance of restricting 2-Wasserstein to machine learning settings deserves deeper discussion.

---

> ### Author Rebuttal · Authors · 2025-07-31
>
> We sincerely thank the reviewer for the careful, detailed, and constructive feedback, which we will duly consider and integrate into our revised manuscript. We hope the review can reevaluate our paper based on our response below:
>
> $\textbf{Weekness}$
> 1. We thank the reviewer for the clarifying question. We would like to clarify that $\epsilon$ is the property of the underlying distribution mapping and is not a parameter. Unlike prior work on decision-dependent distribution shift [Perdomo et al. 2020, Mendler-Dünner et al. 2020], where the mapping P itself is known, our assumption is much weaker. In practice, given two decisions $\theta$ and $\theta'$, along with their corresponding empirical reference distributions $\hat P(\theta)$ and $\hat P(\theta')$ based on observed samples, one can compute the 1-Wasserstein distance $W_1(P(\theta), P(\theta'))$ via linear programming. This allows for an estimate of $\epsilon$ using $\frac{ W_1(P(\theta), P(\theta'))}{\|\theta - \theta' \|_2}$.
> 2. We thank the reviewer for pointing this out. While the number of training samples in the experiment may seem modest, this choice reflects many real-world scenarios where data collection is expensive or limited. For example, in settings like credit scoring, obtaining labeled examples can require costly evaluations, expert judgments, or long delays. Importantly, one of the motivations for our distributionally robust framework is to address such small-sample effects. By explicitly accounting for uncertainty in the data-generating distribution, DRO-based methods can improve generalization and robustness even when the available data is limited. We will clarify this motivation in the revision.
> 3. We thank the reviewer for the constructive sugggestion. Traditional DRO methods typically assume a fixed but unknown data-generating distribution and aim to minimize the worst-case risk over a predefined ambiguity set. In contrast, the performative setting introduces a unique challenge: the data distribution itself depends on the model parameters. This endogenous dependence means that standard DRO formulations—which are designed to handle exogenous uncertainty—cannot directly account for the feedback loop between decisions and the data-generating process.
>
>    Moreover, a key technical challenge in the performative setting lies in identifying structural assumptions that guarantee the convergence of the proposed algorithm, despite this decision-dependent distributional shift. We will revise the paper to include a more explicit comparison between our framework and traditional DRO approaches, and to better highlight the novel challenges posed by the performative setting.
> 4. We thank the reviewer for the careful feedback.  In this context, "machine learning" refers to regression and classification problems previously studied in the performative prediction literature [Perdomo et al. 2020, Mendler-Dünner et al. 2020, Jagadeesan et al. 2022], where the objective typically involves minimizing a loss over data influenced by the model’s predictions. In contrast, decision-making problems often involve economic costs as well as risk-sensitive objectives, such as the maximum of several functions to capture risk aversion or worst-case scenarios. These formulations introduce additional challenges under the performative setup. We note that Model 2 in our paper is designed to accommodate such settings, providing a more flexible framework that can handle these more complex, decision-oriented objectives. We will clarify this distinction in the revised version to better highlight the generality of our approach beyond standard machine learning formulations.
>
> $\textbf{Questions}$
>
> We thank the reviewer for this excellent question. Our results in Model 3 rely on Danskin's theorem, which requires the loss function to be concave in the uncertainty in order to get a well-defined derivative of the worst-case objective. Extending the analysis to settings where the objective involves the maximum of multiple such loss functions poses significant challenges. In particular, taking a maximum over concave functions generally destroys concavity. Addressing this would require new analytical tools. Exploring these directions is an interesting avenue for future work.
>
> $\textbf{Limitations}$
>
> We thank the reviewer for the careful feedback. We would like to emphasize that the limited data regime we consider is not just a practical constraint, but a characteristic feature of the performative optimization setting. Unlike traditional machine learning, where data is assumed to be drawn from a fixed or evolving distribution, performative optimization requires sampling from distributions that are themselves induced by the decisions. That is, in order to evaluate or improve a policy, one must actively deploy a decision (e.g., hiring policy, pricing policy) and then observe the resulting data distribution, which can be expensive, time-consuming, or infeasible at scale [Milli et al. 2019, Mendler-Dünner et al. 2020]. Our method is designed to be data-efficient in this regime, and we aim to achieve robust and stable performance even with limited samples per iteration. We will clarify this in the revised manuscript.
>
> $\textbf{References}$
>
> - S. Milli, J. Miller, A. D. Dragan, and M. Hardt. The social cost of strategic classification. In Proceedings of the conference on fairness, accountability, and transparency, pages 230–239, 2019
> - J. Perdomo, T. Zrnic, C. Mendler-Dünner, and M. Hardt. Performative prediction. In International Conference on Machine Learning, pages 7599–7609. PMLR, 2020
> - C. Mendler-Dünner, J. Perdomo, T. Zrnic, and M. Hardt. Stochastic optimization for performative prediction. Advances in Neural Information Processing Systems, 33:4929–4939, 2020
> - M. Jagadeesan, T. Zrnic, and C. Mendler-Dünner. Regret minimization with performative feedback. In International Conference on Machine Learning, pages 9760–9785. PMLR, 2022

---

> > ### Comment · Reviewer_MXSY · 2025-08-05
> >
> > I thank the authors for their thorough rebuttal and additional analysis, which has resolved the majority of my concerns. Accordingly, I decide to keep my rating as borderline accept.

---

> ### Author Response · Authors · 2025-08-06
>
> Dear Reviewer MXSY,
>
> We sincerely thank you for taking the time to thoroughly evaluate our work and provide such detailed and constructive feedback. Your insightful comments, questions, and suggestions have significantly helped us improve both the technical content and the presentation of our paper.
>
> Thank you again for your time and effort in reviewing our submission.
>
> Sincerely,
>
> The Authors

---

### Note · Authors · 2025-08-16

We thank all reviewers for their thoughtful evaluations and constructive feedback. We are encouraged by recognition that our work introduces a novel and rigorous framework for distributionally robust performative optimization (DRPO), establishes new convergence guarantees, and demonstrates consistent empirical improvements across diverse applications. Our framework is designed to encompass a broad class of problems in performative optimization, including strategic classification, strategic regression, and decision-dependent problems in revenue management and portfolio optimization.

Several concerns focused on assumptions in Models 1–2. We clarify that the quadratic and Lipschitz-type structures are technical devices enabling tractable reformulations and do not limit generality. Model 3 (Appendix F) handles general decision-dependent distributions. We will highlight this explicitly and clarify how our assumptions compare favorably with prior work, which often relies on stronger smoothness or sensitivity conditions.

Reviewers appreciated the breadth of our experiments. While ablations and sensitivity studies were limited by space, we will expand them and clarify solver runtimes. Unlike classical DRO or ML benchmarks, performative decision-making (e.g., pricing, hiring, portfolio design) lacks established baselines for distributionally robust methods; our experiments demonstrate DRPO’s generality.

We also note some misunderstandings about our framework. We will revise the presentation to emphasize that DRPO addresses robustness in decision-dependent distributions, a more challenging setting than traditional DRO.

Finally, we thank reviewers for recognizing the theoretical novelty, particularly the use of Danskin’s theorem and the exponential smoothing algorithm. These tools enable principled handling of nonsmooth objectives beyond smooth convex problems. We will provide additional background on exponential smoothing (e.g., in the appendix) to improve accessibility, which contributed to the improved assessment.

Overall, our paper establishes the first convergence guarantees for distributionally robust performative optimization under Wasserstein ambiguity, broadens the scope beyond predictive tasks, and provides empirically validated, data-efficient methods for decision-dependent settings. We believe this work provides a strong foundation for future research at the intersection of DRO and performative optimization.

---

### Decision · Program_Chairs · 2025-09-17

**Decision:**

Accept (poster)

**Comment:**

The paper studies the performative optimization under a distributionally robust setting. The paper is well-written, and it addresses technical challenges arising in this new formulation. All the reviewers are leaning positive.

On the positive side, I am in support of this paper based on the new setting studied, the efforts made by the authors to overcome the technical challenge, and the writing. While I support accepting the paper, I still want to point out my reservations:
- Studying a setting "different" or "hasn't been studied before by other papers", or "resolving some technical challenges that don't appear in the previous setting", all of these don't mean that it is a setting worth working on. In other words, a niche setting that hasn't been studied before doesn't justify its usefulness. In the past few years, we have seen tons of papers on the topic of distributionally robust optimization; hardly any have been applied to a real application context. Many papers motivate the considered setting or Wasserstein distance from ML applications, such as Wasserstein GANs or the usage of distributional robustness in improving the ML model performance. While I think it is inappropriate to say that these imaginary robustness motivations become largely obsolete by the new methods, such as CLIP (OpenAI 2021) and others, I sincerely hope the researchers and the community working on DRO/RO to look for new problems and learn new techniques, rather than spending further time filling the gaps in the existing scope.